# Millennial-timescale thermogenic $CO_2$ release preceding the Paleocene-Eocene Thermal Maximum

Shijun Jiang ®[1,2] ✉, Ying Cui ®[3] ✉, Yasu Wang[1], Maurizia De Palma[3], B. David A. Naafs ®[4], Jingxin Jiang[5], Xiumian Hu[5], Huaichun Wu ®[6], Runjian Chu[6], Yangguang Gu ®[7], Jiuyuan Wang ®[8], Yizhou Huang ®[4], Miquela Ingalls ®[9], Timothy J. Bralower ®[9], Shiling Yang[10], James C. Zachos[11] & Andy Ridgwell ®[12]

Geologic records support a short-lived carbon release, known as the pre-onset excursion (POE), shortly before the Paleocene-Eocene Thermal Maximum (PETM; ~ 56 Ma). However, the source and pace of the POE carbon release and its relationship to the PETM remain unresolved. Here we show a high-temporal-resolution stratigraphic record spanning the POE and PETM from the eastern Tethys Ocean that documents the evolution of surface ocean carbon cycle, redox and eutrophication, confirming the global nature of the POE. Biomarkers extracted from the sedimentary record indicate a smaller environmental perturbation during the POE than that during the PETM in the eastern Tethys Ocean. Earth system modeling constrained by observed $\delta^{13}C$ and pH data indicates that the POE was driven by a largely thermogenic $CO_2$ source, likely associated with sill intrusions prior to the main eruption phase of the North Atlantic Igneous Province and possibly biogeochemical feedbacks involving the release of biogenic methane.

A holistic understanding of the carbon-climate dynamics of past warming events has important implications for $CO_2$-induced anthropogenic climate change. The Paleocene-Eocene Thermal Maximum (PETM; ~56 Ma[1–3]) represents the largest disruption of the global carbon cycle in the Cenozoic[4], which led to 5–6 °C global warming[5,6], ocean acidification[7], ocean deoxygenation[8–11], and intensified tropical cyclones[12,13]. The prominent 3–6‰ negative carbon isotope excursion (CIE) registered in both terrestrial and marine sections is consistent with major emissions ( ~ 2000 to >13,000 Pg C) of $^{13}C$-depleted carbon

to the atmosphere and/or ocean and on a timescale of a few to no more than ~ 20 kyr[14,15]. Recent work suggests that the North Atlantic Igneous Province (NAIP) and associated $CO_2$ emissions may have triggered the PETM[16–18], followed by carbon sequestration through organic carbon burial[19] and silicate weathering[20]. The PETM was proceeded by a transient warming accompanied by a smaller CIE[21]−known as the pre-onset excursion (POE) and which is recorded in terrestrial records from the Wyoming Bighorn Basin[21] together with only a few shallow marine sections (Atlantic coastal plain, southwest Pacific Ocean, the

[1]State Key Laboratory of Marine Resource Utilization in South China Sea, Hainan University, Haikou, China. [2]Southern Marine Science and Engineering Guangdong Laboratory (Zhuhai), Zhuhai, China. [3]Department of Earth and Environmental Studies, Montclair State University, Montclair, NJ, USA. [4]Organic Chemistry Unit, School of Chemistry and School of Earth Sciences, University of Bristol, Bristol, UK. [5]State Key Laboratory of Critical Earth Material Cycling and Mineral Deposits, School of Earth Sciences and Engineering, Nanjing University, Nanjing, China. [6]School of Ocean Sciences, China University of Geosciences (Beijing), Beijing, China. [7]South China Sea Fisheries Research Institute, Chinese Academy of Fishery Sciences, Guangzhou, China. [8]SKLab-DeepMinE, MOEKLab-OBCE, School of Earth and Space Sciences, Peking University, Beijing, China. [9]Department of Geosciences, The Pennsylvania State University, University Park, PA, USA. [10]State Key Laboratory of Lithospheric and Environmental Coevolution, Institute of Geology and Geophysics, Chinese Academy of Sciences, Beijing, China. [11]Department of Earth and Planetary Sciences, University of California, Santa Cruz, CA, USA. [12]Department of Earth and Planetary Sciences, University of California, Riverside, CA, USA. ✉e-mail: sjiang@hainanu.edu.cn; cuiy@montclair.edu

North Sea and the Pyrenean foreland basins)[22–26]. The POE is a short-lived warming event that occurred about 38 kyr to >100 kyr[27] prior to the PETM onset with an estimated duration of no more than a few centuries[22] to millennia[27]. As an environmental precursor to the PETM, the POE is absent in deep-sea sedimentary records because its short duration may have limited its preservation to surface and shallow water records[22]. Resolving a global POE signal could be further complicated by bioturbation, sediment mixing, and chemical burndown of deep-sea carbonates[7,28], which could only be understood by studying shallow marine and terrestrial sections. The POE warming may represent an early warning signal on the instability of carbon reservoirs and set the stage for a climatic threshold crossing occurred during the PETM. Previous studies suggest that the PETM is modulated by astronomical forcing[29–31], and linked with the POE via repeated, catastrophic $CO_2$ release[27], such as methane hydrate dissociation[22], either as a direct response of the warming or via positive feedback mechanisms. Furthermore, the close timing between the initial stage of the NAIP and the POE suggests that volcanism and magmatism may also serve as a viable trigger[32]. However, the global extent of the POE,

its relationship with the PETM and exact mechanisms that triggered the POE—whether from methane hydrate release, volcanic activity, or orbital drivers—remain debated.

Here we report ultra-high-resolution biogeochemical records from a recently discovered coastal shallow marine section in the eastern Tethys that span both the POE and the PETM (Fig. 1). The Kuzigongsu section (39°45'10" N, 75°17'29" E) is located in the western Xinjiang Uygur Autonomous Region of China, which was covered by the Turan Sea—an arm of the Tethys Ocean during the early Paleogene (Fig. 1). The eastern Tethys was a restricted shallow-water carbonate platform environment[33], and a critical site for the formation of warm and saline intermediate water and the burial of organic matter[34]. Abundant calcareous nannofossils[35] and well-preserved organic matter and oyster shells (Fig. S1) allow for an integrated sedimentological, biogeochemical, isotopic, organic geochemical, and global carbon cycle modeling approach to unravel the paleoenvironmental evolution of the eastern Tethys during the POE and PETM, thus filling a critical spatial data gap and advancing knowledge on forcing and recovery mechanisms of ancient hyperthermals.

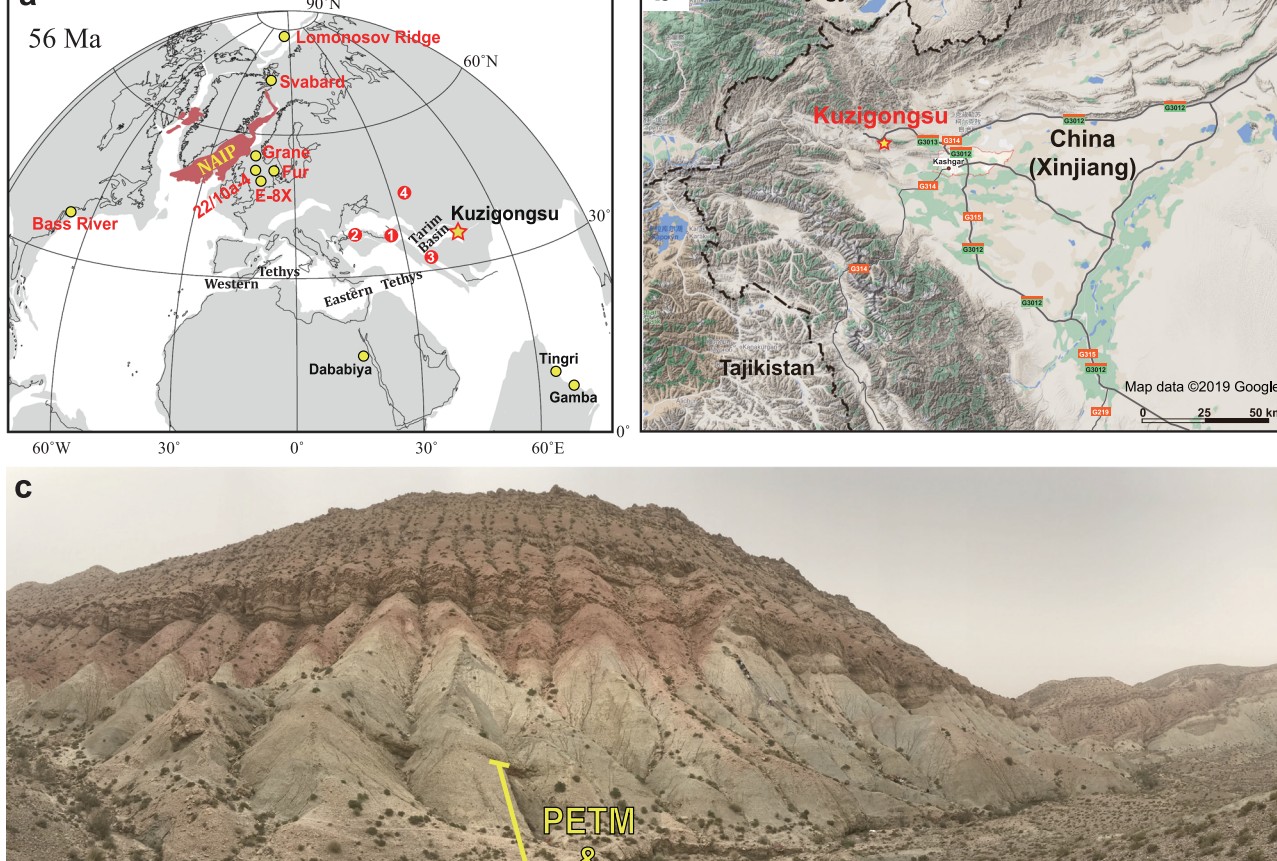

**Fig. 1 | Maps showing the paleogeography, location and outcrop image of the study site at the Kuzigongsu section. (a)**, paleogeographic map of the study area during the early Paleogene[106]. The base map was created from https://www.odsn.de using the reconstruction data from ref. 107 and edited using Adobe Illustrator, **(b)**, present location of the study site generated using Map data ©2019 Google, and **(c)**, a photo of the outcrop. Panel **(a)** also shows other shallow water Paleocene-Eocene Thermal Maximum (PETM) and pre-onset excursion (POE) records in

Aktumsuk[108] (1), Kheu River and Guru-Fatima[109] (2-3), West Siberian Sea[110] (4), southern Tibet (Tingri and Gamba)[111,112], Tarim Basin[35], Denmark (E-8X, 22/10a-4, Grane, and Fur)[18,64,113,114], Svalbard[14], Arctic (Lomonosov Ridge)[115], and Mid-Atlantic Coastal Plain Sites (Ancora, Wilson Lake, Clayton, and Millville located in the New Jersey, and South Dover Bridge or SDB and Cambridge-Dorchester Airport located in the Salisbury Embayment in Maryland)[116–121].

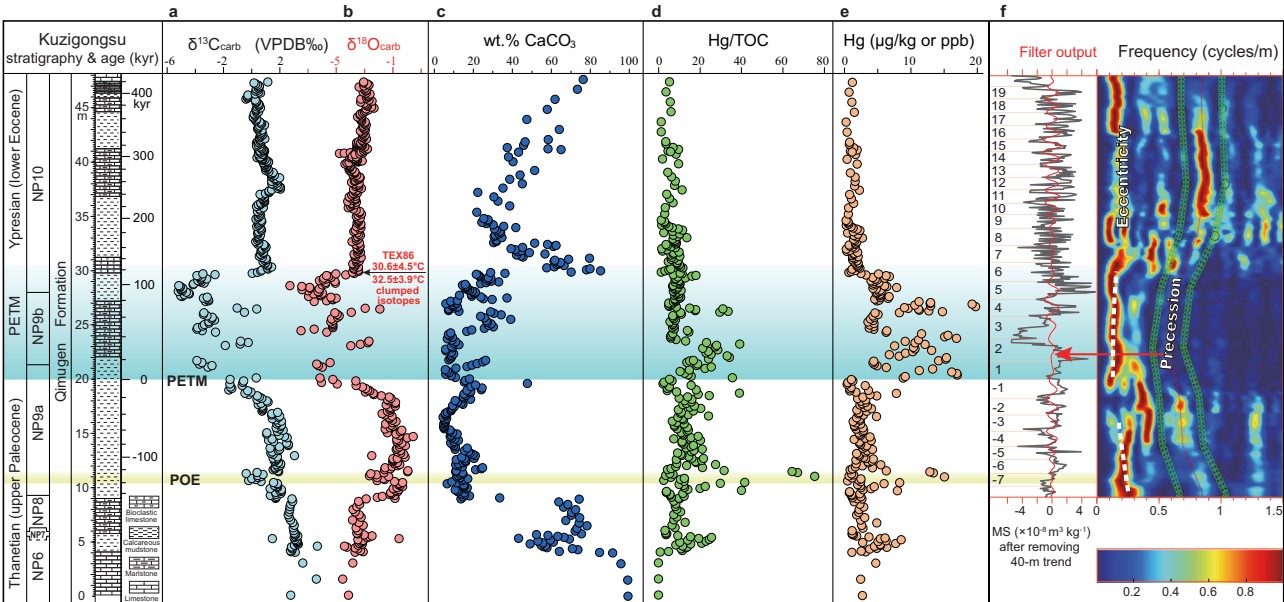

**Fig. 2 | Characteristics of the Paleocene-Eocene Thermal Maximum (PETM) and pre-onset excursion (POE) records at the Kuzigongsu section.** (**a**, **b**), $\delta^{13}C_{carb}$ and $\delta^{18}O_{carb}$ from Wang et al. (2022)[35]. Note the two novel sea surface temperature estimates based on oyster fossil $\Delta_{47}$ and $TEX^{H}_{86}$ at 29.8 m depth. (**c**), wt.% CaCO₃. (**d**), Mercury to total organic carbon content (Hg/TOC) ratio. (**e**), Hg concentration.

(**f**), astronomically tuned age model based on magnetic susceptibility (MS) across the POE and PETM. The color bar represents spectral power and the green band represents the bandwidth of the precession cycles. Numbers in (**f**) indicate precession cycles assignments.

## Results and discussion

### Astronomically tuned high-resolution PETM and POE records from the understudied eastern Tethys

The PETM has been identified at the Kuzigongsu section through calcareous nannofossil biostratigraphy (the NP9/NP10 boundary) and carbon isotope stratigraphy[35]. It occurs at 19.9 m (on a depth scale of 0 to 48 meters in Fig. 2) and corresponds to a ~ 6–8‰ negative carbon isotope excursion (CIE)—among the largest CIEs observed in shallow marine sites[4]. The CIE magnitude is ~ 6.3‰ in carbonate[35], ~ 6.0‰ in organic matter, and somewhat amplified in long-chain *n*-alkanes (~ 7.8‰), which is likely a result of an enhanced hydrological cycle[36] and elevated $pCO_2$[22,37]. The primary $\delta^{13}C_{carb}$ signal is likely well preserved, based on: (1) the strong covariation between $\delta^{13}C_{carb}$ and $\delta^{13}C_{org}$ ($r^2 = 0.75$, $p < 0.001$; Fig. S3); (2) the fact that most $\delta^{13}C_{carb}$ and $\delta^{18}O_{carb}$ data plot within the area of primary carbonates[38] (Fig. S3), and (3) the presence of only a weak correlation ($r^2 = 0.18$, $p < 0.001$, Fig. S3) between $\delta^{13}C_{carb}$ and Mn/Sr, as a strong correlation would indicate diagenetic alteration[39].

The POE is found at ~ 8.4 m below the PETM onset within lower nannofossil Zone NP9a[35] and occurs in a 1.2-meter-thick interval (10.3 to 11.5 m) characterized by a −1 to −2.5‰ CIE (Fig. 2). Specifically, we observed CIEs of −2.5‰ in carbonate and −2.1‰ in organic matter, but in contrast to the PETM, only ~ −1‰ in long-chain *n*-alkanes. The relatively smaller recorded magnitude in the *n*-alkane record is likely due to the lack of data at 10.8 m depth where $\delta^{13}C_{carb}$ and $\delta^{13}C_{org}$ values reach their minima (Fig. 2).

Power spectrum analysis of the detrended magnetic susceptibility (MS) data series shows significant peaks in wavelength at 0.8, 1.2, 1.9, 3, 5, 6.5, and 9.8 m (see Methods, SI and Figs. S3–S5), with the filtered 1.2–1.9 m cycles interpreted as precession signal with an assumed 21 kyr duration and the filtered 5 to 9.8 m cycles as short eccentricity (~ 100 kyr). Spectral analysis revealed sedimentation rates averaging between 6.0 and 8.3 cm kyr⁻¹ (Fig. S5) and suggests that the durations of the PETM and the POE at our study site are ~ 127 kyr and ~ 21 kyr, respectively (age model option 1; see SI and Supplementary Data 2 for details). The PETM and POE are separated

by ~ 144 kyr (± 21 kyr). The estimated PETM duration of 127 kyr is shorter than inferred from the deep sea sites (e.g., ~ 170 kyr from Röhl et al.[2] and Zeebe and Lourens[3]), likely due to incomplete preservation of the entire PETM at Kuzigongsu with a change in lithology that truncates the recovery phase. The POE onset duration of ~ 7.0 kyr (age model option 1) is similar to, but slightly longer than the 2 to 5.5 kyr estimated by Bowen et al.[21]. An alternative age model (Option 2) that accounts for the significant drop in wt.% CaCO₃ and a likely truncation assumes the filtered 6–10 m cycles represent ~ 20 kyr precessional signal. This age model option provides a duration of ~ 39 kyr for the PETM, ~ 4 kyr for the POE and ~ 54 kyr between the PETM and POE, which suggests the study site only preserves the PETM onset and the plateau, rather than the recovery (see SI for more discussion). However, due to the uncertainty in the astronomically tuned age model, we assume that the POE onset duration ranges from 500 to 7000 years to cover the full range of reported values in the literature[22,27].

### Paleoenvironment of the eastern Tethys during the POE and PETM

We use a multi-proxy approach to reconstruct the paleoenvironmental evolution of the eastern Tethys during the POE and PETM (Supplementary Data 1). Our records (Figs. 2, 3) include C/N ratios as indicators of organic matter source, weight percent (wt.%) CaCO₃ as a proxy for ocean acidification and detrital dilution, trace element geochemistry for marine nutrient and chemical weathering proxies, organic biomarkers as proxies for marine microbial communities, and mercury content as a possible indicator of the NAIP activity. Together, our new data suggest that the shallow eastern Tethys experienced profound environmental changes, including extreme warmth, eutrophication, and biological turnover. Furthermore, the moderately high sedimentation rates (optimal sedimentation rate fluctuates between ~ 6–8 cm kyr⁻¹; Fig. S3, S4) at this shallow site (estimated water depth is ~ 30–50 m based on microfacies analysis and foraminifera indicators[33]) yield highly expanded records that provide unique details on the relationship between the PETM and the POE. Such details are generally

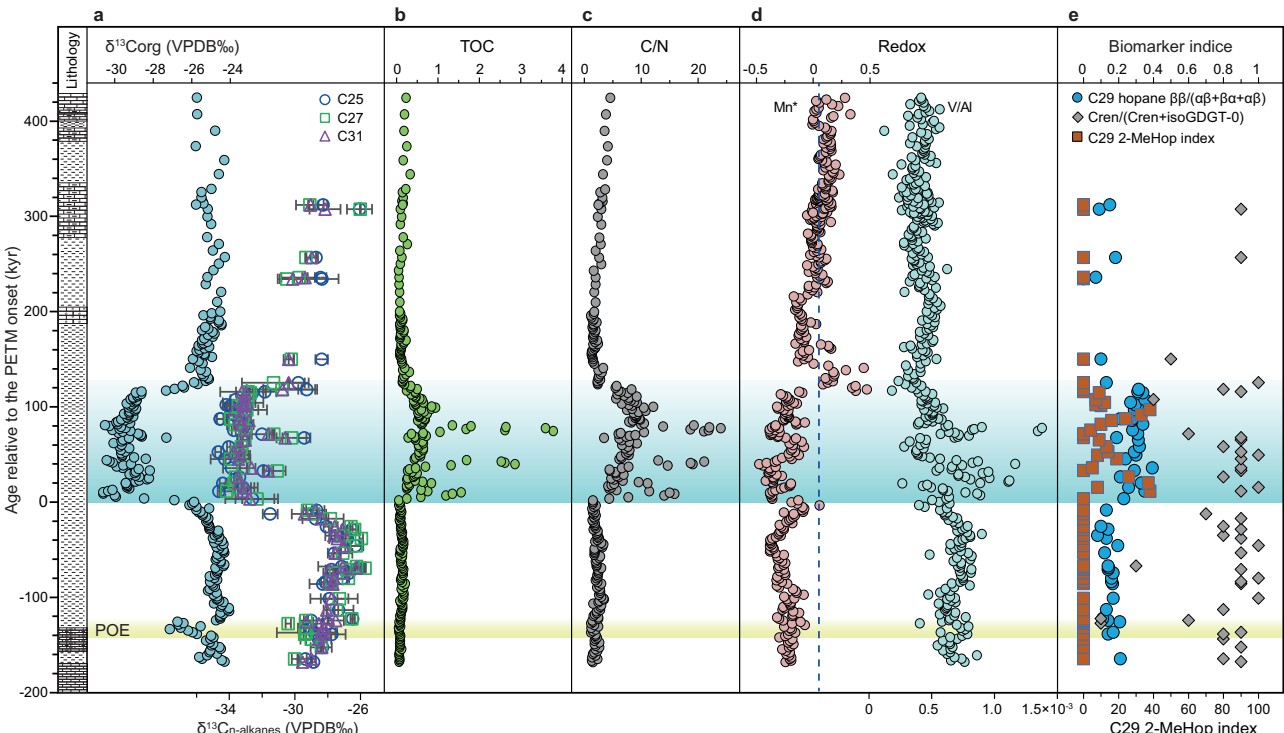

**Fig. 3 | Proxy-based reconstruction of environmental changes across the Paleocene-Eocene Thermal Maximum (PETM) and pre-onset excursion (POE) at the Kuzigongsu section, eastern Tethys.** Relative age from the onset of the PETM is based on an astronomically tuned age model described in Methods and Supplementary Information. (**a**), $\delta^{13}C_{org}$ from bulk organic matter and $\delta^{13}C_{n\text{-alkanes}}$ from long-chain $n$-alkanes ($nC_{25}$ in blue circles, $nC_{27}$ in green squares, and $nC_{31}$ in purple triangles); (**b**), Total organic carbon content (TOC); (**c**), organic carbon to nitrogen ratio (C/N); (**d**), Mn* (pink circles)[63] and V/Al ratio (orange circles) as redox proxies; (**e**), Biomarker indices based on $C_{29}$ hopane $\beta\beta/(\alpha\beta + \beta\alpha + \alpha\beta)$ (blue circles), Crenarchaeol/(Crenarchaeol+isoGDGT-0) or Cren/(Cren+isoGDGT-0) (dark blue diamond), and $C_{29}$ 2-Methylhopane index (2-MeHop) (red squares).

obscured in deep-sea sites because of lower sedimentation rates, dissolution, and bioturbation[7].

The section is characterized by a rapid decrease in wt.% CaCO₃ from >80 wt.% to near 0 wt.% at ~ 9 m—a shift which precedes the POE and PETM and may be attributed to significant reduction of carbonate production, detrital dilution, and/or shallow ocean acidification[7,40]. The sharp decrease (−2.5‰) in oxygen isotopes of marine carbonate values ($\delta^{18}O_{carb}$) is consistent with an abrupt and significant warming during the POE and PETM (Fig. 2b), though the magnitude is likely an artifact of diagenetic overprinting. Additionally, a portion of the $\delta^{18}O_{carb}$ decrease may represent a decline in local salinity as the $\delta^{18}O_{sw}$ as epeiric sites can be strongly influenced by freshwater input from surrounding continents[5]. Clumped isotope data from a well-preserved oyster specimen (at 29.8 m; Fig. S1) indicate that the eastern Tethys surface water temperature was around $32.5 \pm 1.5\,°C$ (1σ) at the recovery phase of the PETM (Fig. 2). This estimate is similar to our independent temperature estimate of $30.6 \pm 4.5\,°C$ (1σ) based on the $TEX_{86}^H$ proxy[41] for the sample at the same depth. However, the thermal maturity is relatively high for this section and the cyclized isoGDGTs abundance is low, preventing us from obtaining a high-resolution and precise $TEX_{86}^H$ temperature record at the site (Fig. 2).

In the organic matter fraction, peak TOC and C/N ratios coincide with the lowest $\delta^{13}C_{org}$ values during the PETM, suggesting increased terrestrial organic matter input at the study site, a likely consequence of intensified continental weathering and/or higher terrestrial primary production[42]. The inferred increase in terrestrial weathering is supported by the higher values of Ti/Al and K/Al ratios[11]. Elevated $C_{29}$ hopane $\beta\beta/(\alpha\beta + \beta\alpha + \alpha\beta)$ ratios (average = 0.3) during the PETM indicate increased input of fresh organic matter either due to higher primary productivity or increased flux of fresh terrestrial organic matter into the basin (Fig. 3e). Lower $C_{29}$ hopane $\beta\beta/(\alpha\beta + \beta\alpha + \alpha\beta)$ ratios (average = 0.1) in the pre- and post-PETM samples suggest relatively low primary production in the surface waters with background input of reworked and more mature organic matter from the surrounding continents[42]. Similarly, Crenarchaeol/(Crenarchaeol +isoGDGT-0) ratios range from 0.1 to 1.0, with a significant decrease during the POE and PETM. Crenarchaeol (with four cyclopentane rings and one cyclohexane ring) is considered as a biomarker for Thaumarchaeota[43]. The lower Cren/(Cren+isoGDGT-0) ratios during the POE and PETM therefore likely reflect a reduction in marine Thaumarchaeota, which may be attributed to warmer surface ocean temperature and lower dissolved oxygen concentration[44]. The occurrence of 2-methylhopanes (2-MeHop) in the PETM interval indicates a transient perturbation of surface ocean characteristics (Fig. 3). The $C_{29}$ 2-MeHop Index, calculated as $100 \times (C_{29}$ 2-MeHop)/($C_{29}$ 2-MeHop + $C_{29}$ Hop)[45], ranges from ~ 0–38% with two prominent peaks, at 20.9 m and 28.5 m respectively, corresponding to the peak values of TOC and C/N ratios. Several studies reported that the occurrence of 2-MeHop in the sedimentary record can be viewed as indicators of stress responses to the capacity of microbial respiration under hypoxia[46], nitrogen fixation[47], increased productivity[48], and changes in pH[49], corroborating the interpretations of elevated primary productivity discussed above. Furthermore, the anomalously high $C_{29}$ 2-MeHop Index during the PETM may be attributed to marine nitrogen cycle perturbation as a result of biogeochemical changes. This is similar to observations of other major carbon cycle perturbations of the Phanerozoic, such as the end-Permian mass extinction event[50], the end-Triassic extinction event[51], and the Mesozoic Oceanic Anoxic Events[52,53].

Ocean deoxygenation may have been enhanced by increased primary productivity from elevated nutrient input due to enhanced

terrestrial weathering. This suggestion is supported by negative Mn* values (Eq. 1) from the POE to the PETM (Fig. 3), which are associated with more reducing conditions due to significant redox-related changes in the solubility of Fe and Mn[54].

$$Mn* = \log[(Mn_{sample}/Mn_{shales})/(Fe_{sample}/Fe_{shales})] \quad (1)$$

The values used for the $Mn_{shales}$ and $Fe_{shales}$ are 600 and 46,150 ppm, respectively[55]. Furthermore, the inferred surface ocean deoxygenation is consistent with elevated V/Al ratios over the same interval (Fig. 3) because V ions ( + 4 and +5 valence) are closely coupled with the redox cycle of Mn[56]. Widespread deoxygenation is well documented in many ocean basins across the globe during the PETM[8,57], including the North Sea[58], the Arctic Ocean[59], the Atlantic and Caribbean[60,61], and the northwestern Tethyan margins[62]. However, no significant changes in these redox indicators were observed across the POE[63], suggesting relatively stable redox conditions in the eastern Tethys at this time.

Mercury content (or Hg concentration normalized as a ratio to organic carbon content−Hg/TOC) has been used as a signal of NAIP activity by several previous studies[18,64]. Our site exhibits two prominent Hg/TOC peaks that preceded the onset of the POE (˜ 11 kyr) and the PETM (˜ 26 kyr) (Fig. 2), supporting a possible link between Hg source and the $^{13}$C-depleted carbon source. However, because of the overall low Hg concentrations at the study site, establishing a direct link between the NAIP and the Hg peaks is not straightforward. Low Hg is likely due to dilution by carbonate and detrital input, the long distance of the site relative to Hg source, and/or Hg transport via oceanic waters rather than global atmospheric transport[18]. Increased Hg concentrations across the POE and PETM compared to background values suggest that multiple possible sources and processes may have been at play in addition to the NAIP activity. For example, variations in Hg concentrations in the sedimentary records can be caused by changes in river runoff, weathering, transport of terrestrial materials, primary productivity, source of organic matter, and post-depositional processes (e.g., diagenesis and dissolution)[18], which could become more important at the study site because of its restricted carbonate platform setting[65]. Deoxygenation and changes in organic matter preservation and transport cannot fully account for the excess Hg as shown by the steeper Hg gradient to TOC within the PETM and POE interval at our site (Fig. S6). Moreover, Hg fluxes associated with wildfire (e.g., Arctic region[66], northeastern US margin[67], and England[68]) may have been far less than the Hg fluxes associated with a large igneous province event[69], and therefore cannot provide sufficient Hg into the study site. Principal component analysis (PCA) suggests that Hg is most closely related to C/N ratios (higher C/N ratio indicates more terrestrial organic source) and $\delta^{13}C_{org}$ during the PETM, which reflect changes in source of organic matter and $^{13}$C-depleted $CO_2$ emissions (Fig. S6). The C/N ratio exhibits no significant change across the POE, suggesting the increase in Hg and Hg/TOC ratio is unrelated to changes in source of organic matter. On the other hand, C/N ratio shows a large increase across the PETM, which indicates that changes in source of organic matter may have contributed to the increased Hg concentrations. These potential processes do not preclude volcanic involvement, however, especially via more complex pathways than simple atmospheric loading and deposition[70]. Despite these potential complex sources of Hg, we cannot completely exclude direct and indirect involvement of the NAIP in driving the Hg changes in the study section[71]. For example, the NAIP was active as early as 62 Ma[71], and its peak activity may have encompassed both the POE and the PETM[72–75]. A negative shift in $^{187}$Os/$^{188}$Os ratios has been observed prior to the PETM in several sites globally[32,71,76,77], lending support to the occurrence of large igneous province activity prior to the PETM. Furthermore, hydrothermal vent complexes in the northeast Atlantic region[78,79]

further support that the NAIP activity can at least partially explain the observed Hg records.

## Thermogenic $CO_2$ emissions associated with NAIP activity during the POE

The PETM carbon emission history has been extensively modeled in the past, with estimated carbon emission rates ranging from 0.3 to 1.7 Pg C yr$^{-1}$ for a CIE onset duration from ˜ 3000 to ˜ 20,000 years and cumulative amount of carbon added ranging from ˜ 2500 to ˜ 13,000 Pg C[14,15,21,79–81]. Because the carbon emission history preceding the PETM has not been systematically quantified in an Earth system model and very little is yet known about the $CO_2$ source during this time[22], we then focused our model analysis on the POE (Table S1 and Fig. 4). Our new high-resolution geochemical data, together with an orbitally tuned astronomical age model, provide a unique opportunity to assess the effects of $CO_2$ emissions during the POE.

We quantify carbon emissions over the POE using a data assimilation approach that considers paired $\delta^{13}C_{DIC}$-pH variation across the POE within an Earth system model of intermediate complexity cGENIE, following the approach detailed in Gutjahr et al.[15]. In this, changes with time in annual global mean surface ocean pH (derived from $\delta^{11}B$ proxy data from the Mid-Atlantic Coastal Plain with a change of ˜ −0.1 to −0.3 pH units[22]) constrain the emission rate of $CO_2$ to the atmosphere. Similarly, the change with time in observed $\delta^{13}C$ of annual global mean surface ocean DIC ($\delta^{13}C_{DIC}$) (reconstructed by applying an anomaly derived from the $\delta^{13}C$ data of a global compilation; Fig. S7) refines the $\delta^{13}C$ value of the (pH-constrained) $CO_2$ emissions. The novelty of this approach is that it offers a unique solution of the mean $\delta^{13}C_{source}$ without having to make a specific assumption about the carbon source (e.g., compare with Cui et al.[14]; see Methods and SI for detailed model results and sensitivity tests). To account for the uncertainty in the POE onset duration, we place our records on four different age models, including age model options 1 and 2 from this study, an age model from the Bighorn Basin based on Bowen et al.[21] and an assumed age of 500 years based on Babila et al.[22] (a summary of our model results and sensitivity analyses for the POE is listed in Table S1). (Fig. 4).

The flux-weighted $\delta^{13}C_{source}$ values across the entire emission duration vary between −30.8 and −44.5‰ for the four age models used in our simulations with the minimum change in pH suggested by Babila et al.[22], consistent with a thermogenic $CO_2$ source[82] (−30 to −65‰; Fig. 4a–d and Table S1). Longer POE duration (e.g., Age 1 associated with ˜ 7000 year POE onset) necessitates lower flux-weighted $\delta^{13}C_{source}$ values (−44.5‰) over the entire emission interval (Fig. 4a) at slower emission rate (˜ 0.2 Pg C yr$^{-1}$). We note that the $\delta^{13}C_{source}$ values become progressively lower from the POE onset, likely resulting from a faster rate of change toward its minimum values in the $\delta^{13}C$ forcing. This may represent a shift from thermogenic methane to biogenic methane (˜ −34 to <−70‰[83,84]) emissions during the development of the POE. The average carbon emission rate over the entire emission period ranges from 0.2 to 1.3 Pg C yr$^{-1}$ (Fig. 4e–h), comparable to those estimated for the PETM from sill-degassed $CO_2$ and thermogenic methane (0.2 to 0.5 Pg C yr$^{-1}$ from Jones et al.[16]; 0.6 Pg C yr$^{-1}$ from Frieling et al.[79]). Larger magnitude of pH changes (e.g., ΔpH = ˜ 0.2 to 0.3) yield overall larger average peak $CO_2$ emission flux (2.9 Pg C yr$^{-1}$) and higher average $\delta^{13}C_{source}$ values (−19.5‰) (Table S1), still consistent with largely thermogenic methane source. The pH change for the POE has been documented at only a single location, using a novel approach to measuring boron isotopes ($\delta^{11}B$), and therefore has a high degree of uncertainty[22]. Considering the smaller magnitude of $\delta^{13}C$ excursion, the smaller degree of warming, its shorter duration, and the minor ecological responses, the changes in pH during the POE are unlikely to exceed those during the PETM (ΔpH = ˜ 0.3)[85]. A higher average carbon emission rate is associated with shorter POE onset duration (Fig. 4e), which represents a combined impact of the imposed ΔpH forcing and age models used (Fig. S7).

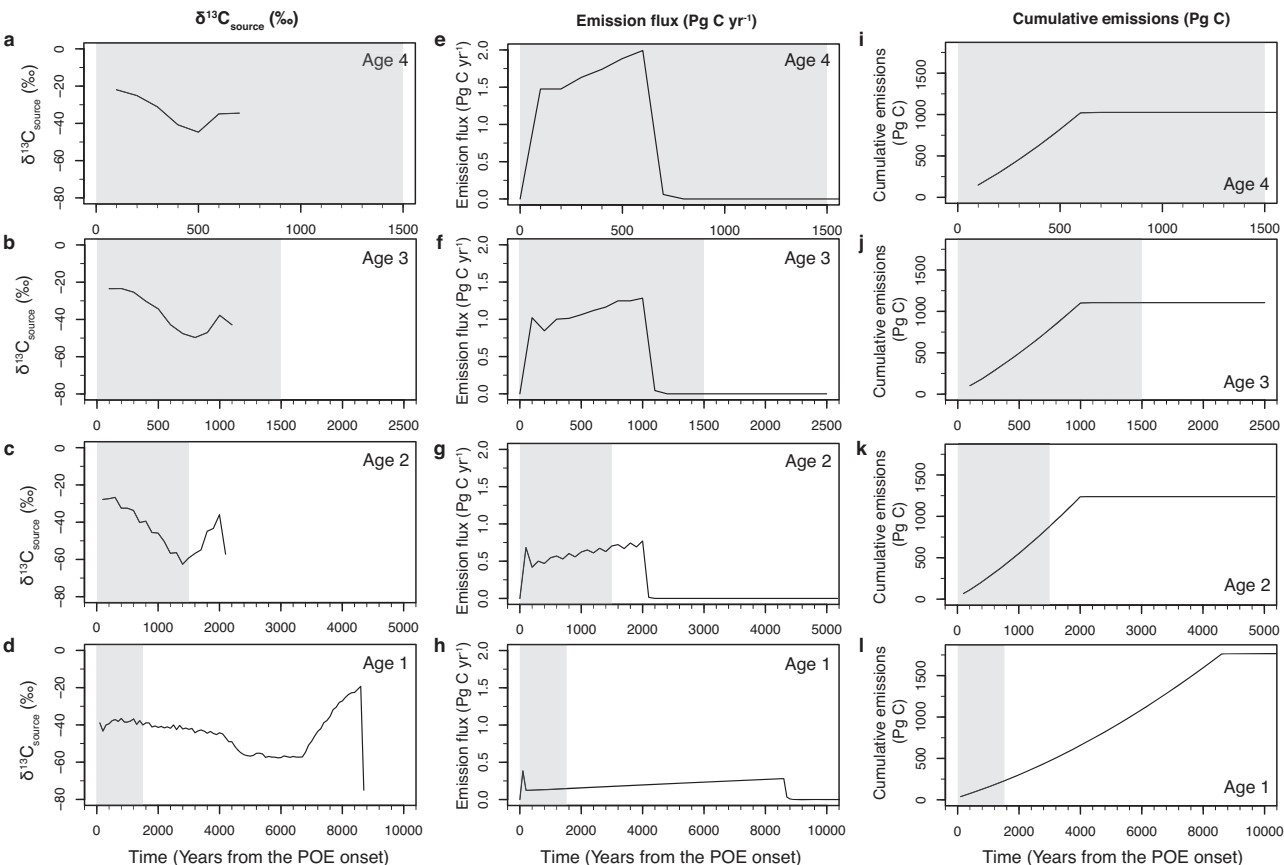

**Fig. 4 | Data assimilation results from our cGENIE Earth system modeling based on the pH·$\delta^{13}C_{DIC}$ double inversion of four scenarios based on different assumptions of POE onset duration.** (**a–d**), $\delta^{13}C_{source}$ values of the diagnosed carbon source for the four age models (see age model interpretation in the main text). (**e–h**), Model-diagnosed rates of $CO_2$ emission for the four age models. (**i–l**), Cumulative amount of $CO_2$ emitted for the four age models. The gray shaded area represents 1500 years.

The cumulative $CO_2$ emission during the POE ranges from ~ 1030 to 1765 Pg C (Fig. 4i–l), with peak $pCO_2$ reaching ~ 1180 to ~ 1220 ppm—a rise of ~ 350 to ~ 390 ppm above ~ 830 ppm (Fig. S8). The modeled cumulative carbon emitted during the POE falls within the range of the 400 to 1600 Pg C suggested by Babila et al.[22] using similar $\Delta$pH. However, if the actual $\Delta$pH was at the lower end (lower than ~ 0.1), it is more likely that the carbon source was primarily biogenic methane. Associated with the diagnosed carbon emissions is a modelled global sea surface temperature rise ($\Delta$T) of ~ 1.1 to ~ 1.3 °C (Fig. S8). Although the paleotemperature history of the POE is currently poorly known, existing Mg/Ca ratios of planktonic foraminifera from the mid-Atlantic coastal plain suggest that the surface ocean temperature increase was ~ 2 °C with an uncertainty of ±1°C due to salinity variations[22], consistent with our modeled temperature changes within uncertainty. The POE warming may also help explain the observed increase in warm-water coccolithophore taxa in the eastern Tethys[35].

Thermogenic $CO_2$ related to the NAIP activities may have been the dominant carbon sources during the POE via contact metamorphism by intrusive activity through hydrothermal vent complexes[78]. It should be noted that mantle convection models suggest that a peak NAIP carbon emission flux at ~ 0.5 Pg C yr$^{-1}$ could occur between 1 and 20 kyr[16], comparable to those simulated in our inversion experiments, despite the geochronology of the NAIP continental flood basalt sequences being not very well constrained[71]. It is also important to note that a caveat of cGENIE in interpreting our results is the lack of terrestrial biosphere and potential changes in orbital forcing, which could impact the climate responses and lead to uncertainties in carbon emission estimates. Although this study provides a range of estimates

on the carbon source and emission flux during the POE, more precise $\delta^{11}$B-based global surface pH records, detailed history of the sill intrusion of the NAIP, sea surface temperature records from across different latitudes, and better-constrained geochronology of the NAIP activity are clearly needed to reduce the uncertainty of the estimated thermogenic carbon emission fluxes from the NAIP.

The evolution of mean core-top carbonate ($CaCO_3$) with time in the model exhibits a smaller magnitude of $\delta^{13}C$ decrease for simulations with bioturbation turned on compared to those without bioturbation (Fig. S9). Similarly, core-top wt.% $CaCO_3$ also exhibits smaller degree of dissolution for experiments with bioturbation on (Fig. S9). Longer experiment duration allows for a larger CIE magnitude regardless of whether bioturbation is on. This is due to the combined effects of bioturbation and dissolution as a result of the cumulative carbon emission (Fig. S9 and Fig. 4i–l), supported by a comparable Eocene hyperthermal event[86]. These experiments support the inference that short POE onset duration (less than millennial timescale) and bioturbation are the main causes of the lack of POE signal in the deep-sea sedimentary records.

In conclusion, we report astronomically tuned, ultrahigh-resolution PETM and POE stratigraphic records from a recently discovered site in the eastern Tethys. Geochemical proxies based on carbonate, bulk organic matter, and biomarkers suggest that the eastern Tethys experienced profound carbon cycle perturbations during the POE and PETM. Our integrated stratigraphic data and Earth system modeling together suggest that the millennial time-scale POE may be attributed to mainly thermogenic $CO_2$ emission associated with sill intrusion prior to the main eruption phase of the NAIP, with

contributions from amplifying feedbacks such as biogenic methane release. Furthermore, our findings predict substantial carbon fluxes driving the POE (averaging 0.2 to 1.3 Pg C yr$^{-1}$), which could be tested by refined geochronological investigations of potential sources such as the NAIP. The POE may have set the stage for the ecosystem threshold crossing and the extreme carbon cycle disruption occurred during the PETM.

## Methods

### Cyclostratigraphy and astronomically tuned age model based on magnetic susceptibility measurements and time series analysis

A total of 480 samples at 10 cm intervals spanning both the POE and the PETM weighing 4 to 8 grams were measured for bulk mass-normalized magnetic susceptibility (MS or χ) using KLY-4S Kappabridge after being crushed in a copper rock hammer and placed in a $2 \times 2 \times 2$ cm$^3$ cubic plastic holder. The MS measurements were conducted at the Paleomagnetism and Environmental Magnetism Laboratory at the China University of Geoscience (Beijing). Measurements were made at room temperature with an applied field amplitude of 200 A/m and frequency of 976 Hz. Each measurement is corrected for the contribution of the plastic sample holder. Each sample was measured three times, with the average value corrected by mass to obtain χ in units of m$^3$ kg$^{-1}$. Relative standard deviations between the three runs were smaller than 0.5%.

Time-series analysis was conducted using MS data with the open-source software Acycle V2.4[87] because MS measures the magnetic mineral concentration, and is considered as a proxy for detrital fluxes from land to the ocean[88]. The MS data series was first detrended by subtracting a 40 m "loess" trend (locally estimated scatterplot smoothing, a non-parametric method for a series of data smoothing with a default window size of 35%) to remove non-periodic or high-amplitude long-term trends following the procedures described in Li et al.[89]. The multi-taper method (MTM)[90] with 2π tapers was used to estimate the spectrum for the detrended MS series and confidence levels (mean, 90%, 95%, and 99%) were provided to test against robust first-order autoregressive model AR(1) red noise in order to reveal the MS series' dominant wavelength. The evolutionary power spectra were calculated with "Evolutionary Spectral Analysis" function in Acycle with a sliding window of 10 m and a step of 0.1 m to identify any secular trend in dominant frequencies, which may be attributed to variations in sedimentation rates. The time scale optimization (TimeOpt; Meyers[91]) and correlation coefficient (COCO; Li et al.[87]) methods were used to identify the optimal sedimentation rate using Acycle's "COCO" and "TimeOpt" functions, which use 2000 Monte Carlo statistical simulations to test the null hypothesis of no orbital forcing. The evolutionary versions of COCO and TimeOpt functions (i.e., eCOCO and eTimeOpt) were used to track changes in sedimentation rates. In addition, the "Spectral Moments" function was used to estimate variable sedimentation rates based on a periodogram with two spectral moments: evolutionary mean frequency (μf) and evolutionary bandwidth (B) (Fig. S3, S4). Subsequently, "Dynamic Filtering" function was used to apply dynamic filtering and isolate interpreted precession cycles from the MS data series. Since the power of long-term cycles (i.e., short eccentricity cycles) may have muted the manifestation of precession cycles in the evolutive harmonic analysis (EHA), we remove the > 4 m cycles that may be associated with eccentricity cycles to reveal precession-related cycles as the most prominent signal in the EHA spectrogram (Fig. S5). The significant power of the interpreted precession cycles in the EHA spectrogram allows us to effectively isolate this signal from EHA (Fig. S5). We then use the precession cycles to construct an astrochronological timescale for the study interval. Analyses of TimeOpt and COCO indicate alternation of optimal sedimentation rates (i.e., 6.0 cm kyr$^{-1}$ and 8.3 cm kyr$^{-1}$) (Fig. S3). Spectral Moments, eTimeOpt and eCOCO together suggest the estimated

sedimentation rate ranges from 4.2 to 10.6 cm kyr$^{-1}$ with increased sedimentation rate during the PETM body (Fig. S3–S5).

### Stable carbon isotopes of bulk organic matter and wt.% CaCO$_3$

HCl-treated carbonate-free powders were measured for total organic carbon (TOC) and total nitrogen (TN) concentrations on a Vario EL-III elemental analyzer, and the δ$^{13}$C$_{org}$ analyses were made using a thermo DELTA plus XL mass spectrometer at State Key Laboratory of Organic Geochemistry, Guangzhou Institute of Geochemistry, Chinese Academy of Sciences. Three reference materials were used to monitor the measurement of carbon isotopic ratio of bulk organic carbon, which included black carbon (−22.43‰), Urea#1 (−34.13‰), and Urea#2 (−8.02‰). Precision based on repeated measurement of these three standards were 0.12‰, 0.08‰, and 0.09‰, respectively. δ$^{13}$C$_{org}$ values were reported in VPDB and analytical precision was better than ±0.1‰ based on replicate analyses of the standards processed with each batch of samples. Weight percent (wt.%) CaCO$_3$ was measured using a modified acid soluble weight-loss method[92].

### Carbonate clumped isotope geochemistry

The carbonate clumped isotope thermometer is based on the thermodynamic stability of C–O bonds at varying temperature, in which "clumping" of the rare, heavy isotopes of carbon and oxygen ($^{13}$C and $^{18}$O) occurs more frequently at lower temperatures[93]. The excess occurrence of the $^{13}$C$^{18}$O$^{16}$O isotopologue of CO$_2$ relative to a stochastic distribution of the heavy isotopes among all CO$_2$ molecules is referred to as the mass 47 anomaly and notated as $\Delta_{47}$, in which $\Delta_{47} = \left( \frac{^{47}R}{^{47}R^*} \right) \times 1000$ where $^{47}R = [^{13}C^{16}O^{18}O + ^{12}C^{17}O^{18}O + ^{13}C^{17}O_2]/[^{12}C^{16}O_2]$ and * denotes a stochastic distribution of isotopes. Clumped isotope thermometry presents a significant innovation over oxygen isotope-based thermometry because the temperature estimate is independent of the bulk isotopic composition, and thus requires no assumptions about δ$^{18}$O$_{carb}$ or δ$^{18}$O$_{water}$. This mineral formation temperature can be used to calculate δ$^{18}$O of ancient waters when paired with δ$^{18}$O$_{carb}$ values of the same sample, which is measured concurrently with $\Delta_{47}$.

Carbonate clumped isotope measurements of one Eocene fossil oyster (*Crassostrea* sp.) and one modern oyster specimen (*Crassostrea hongkongensis*) collected from northern South China Sea (21°42'7.89" N, 111°55'44.61" E) in 2022 were made at the Pennsylvania State University in April 2022 (see SI). Approximately 8 mg of pure carbonate powder was digested in a 105% phosphoric acid common acid bath at 90 °C to yield CO$_2$. Evolved CO$_2$ was passed through a Protium Isotope Batch Extraction (IBEX) carbonate preparation line to purify the sample gas. The gas is passed through a cryogenic trap to separate CO$_2$ from water, a silver wool-packed borosilicate column to trap sulfides, and a gas chromatography column packed with Poropak to separate CO$_2$ from other compounds with a He carrier gas. The purified CO$_2$ gas is once more frozen into a cryogenic trap before being frozen into a microvolume, and passed through a polished nickel capillary to the MAT 253 Plus bellows. Purified CO$_2$ sample gas was analyzed on a Thermo MAT253 Plus dual inlet IRMS relative to an Oztech working gas.

$\Delta_{47}$ values versus the working gas were projected to the Intercarb-Carbon Dioxide Equilibrium Scale[94] (I-CDES) using a carbonate standard-based empirical transfer function. ETH 1, 2, 3, and 4 were measured to build the reference frame and for interlaboratory comparison, and IAEA-C2 and Carrara Marble were treated as unknowns. Individual replicates were averaged to create final sample $\Delta_{47}$ values and reported with a 95% confidence interval. Temperatures were calculated using the T-$\Delta_{47}$ calibration of Anderson et al.[95]. The average measured $\Delta_{47}$ value for the oyster fossil is 0.573 ± 0.011 (2σ), while the $\Delta_{47}$ value for the modern oyster specimen is 0.604 ± 0.028 (2σ). The calculated sea surface temperature in the eastern Tethys based on early Eocene oyster fossil is 32.5 ± 3.9 °C (2σ). The calculated modern sea surface temperature based on modern oyster specimen is

$21.6 \pm 8.7\,°C$ ($2\sigma$), falling in the range of the observed average annual sea surface temperature ($24.1 \pm 5.6\,°C$) in northern South China Sea in 2022.

## Biomarker and stable carbon isotopes of long-chain $n$-alkanes

Around 11 grams of dried and powdered sample were extracted for their biomarker content using a microwave system (Milestone Ethos EX) and using 20 ml of a dichloromethane and methanol mixture (9:1). The total lipid extract was separated using silica flash chromatography and elution with hexane:DCM (9:1) for the apolar and DCM:MeOH (2:1) for the polar fraction. The apolar fractions were characterized on a Thermo Scientific ISQ single quadrupole mass spectrometer (MS) coupled to a gas chromatograph (GC). Compounds were separated using a fused silica column (50 m × 0.32 mm) with a ZB1 stationary phase and helium as the carrier gas. The GC was programmed for: injection at $70\,°C$ (1 min hold), ramp to $130\,°C$ at $20\,°C$/min, followed by a ramp to $300\,°C$ at $4\,°C$/min (20 min hold). The MS continuously scanned between $m/z$ 650-50. The apolar fractions were subsequently analyzed using an Isoprime 100 combustion isotope ratio mass spectrometer, coupled to an Agilent GC, to determine the $\delta^{13}C$ of the long-chain $n$-alkanes. We used the same type of column and temperature program as used for the GC-MS analyses. Samples were measured in duplicate on the GC-C-IRMS, and the average is reported here. An in-house $CO_2$ reference gas was used to calculate compound specific $\delta^{13}C$ values relative to Vienna Pee Dee Belemnite (VPDB). $\delta^{13}C$ values of the $C_{29}$ $n$-alkane are not reported here due to possible co-elution with other lipids. All biomarker and stable carbon isotopes of long-chain $n$-alkane analyses were performed at the University of Bristol.

## Methods for GDGTs

Polar fractions were filtered through a 0.45 μm filter at the university of Bristol. The filtered polar fractions were redissolved in hexane: iso-propanol (99:1) and analyzed using a high-pressure liquid chromatography atmospheric pressure chemical ionization mass spectrometer for their GDGT distribution. We used two ultra-high performance liquid chromatography silica columns to separate compounds, following Hopmans et al. (2016)[96], and analyses were performed in selective ion monitoring (SIM) mode.

The thermal maturity of the organic matter in this section was estimated using the hopane isomerisation index: $C_{29}$ $\beta\beta/(\alpha\beta + \beta\alpha + \beta\beta)$[97]. The results indicate that the thermal maturity changes across the section, but the $C_{29}$ $\beta\beta/(\alpha\beta + \beta\alpha + \beta\beta)$ ratio is consistently below 0.4 (Fig. 3). This is indicative for an elevated thermal maturity, but well below the oil window. Although this level of thermal maturity will not affect apolar compounds like hopanoids or $n$-alkanes, it is likely to impact more labile biomarkers such as glycerol dialkyl glycerol tetraethers (GDGTs)[98]. We determined the GDGT distribution in all samples. As expected with this level of thermal maturity, GDGT concentrations were low and, in most samples, branched (br)GDGTs were absent, as were isoprenoidal (iso)GDGTs containing cyclopentane rings. However, a few samples did have isoprenoidal (iso)GDGTs with cyclopentane rings. This includes the sample at depth 29.8 m that hosts the well-preserved oyster shell fossil. This sample has a $TEX_{86}$ value of 0.76, which results in an SST of $30.6 \pm 4.5\,°C$ using the $TEX_{86}^{H}$ calibration[99]. Although we treat this estimate cautious as thermal maturity might have impacted the GDGTs distribution, this $TEX_{86}$-based SST is consistent with the clumped isotope data from well-preserved oyster shell fossils from the same sample, adding confidence that we are able to constrain the SSTs at this site during the recovery phase of the PETM.

## Earth system modeling

The carbon-centric Grid Enabled Integrated Earth system model (cGENIE) is an intermediate complexity climate model that couples a 3D ocean (36 × 36 grid, 16 levels) with a 2D atmosphere that has the capability to track biogeochemical cycling of elements, stable carbon isotopes, marine sediments, and continental weathering[15,100]. Bathymetry, paleogeography, planetary albedo, and wind fields are configured for the late Paleocene-early Eocene with the same initial and boundary conditions as Gutjahr et al.[15]. For example, the $\delta^{13}C$ value of late Paleocene-early Eocene atmospheric $CO_2$ ($\delta^{13}C_{CO_2}$) is set as ~ –5‰, and the atmospheric $pCO_2$ is set as ~ 830 ppmv. The moderately high $pCO_2$ allows for a small buildup of sea ice (0.5%) in the northern polar regions. We then run a number of 'double inversion' experiments in which $\delta^{13}C$ of surface ocean dissolved inorganic carbon ($\delta^{13}C_{DIC}$) and surface ocean pH[22] are used as the two data assimilation constraints for the POE. The $\delta^{13}C_{DIC}$ forcing is based on the high-temporal-resolution $\delta^{13}C_{carb}$ data from the shallow Tethys Kuzigongsu section using astronomically tuned age models. For our inversion experiments, the model was first spun up for 20 kyr to establish the basic ocean circulation and climatic state under published late Paleocene-early Eocene boundary conditions, including paleogeography and paleobathymetry[101,102]. This is followed by an open-system spin-up of 200 kyr to allow the long-term $\delta^{13}C$ cycle to reach balance. A range of inversion experiments were carried out (Table S1; Fig. S7–9). Although uncertainty exists for pre-PETM $\delta^{11}B$, the surface ocean pH at the end of the open-system spinup is 7.75, same as those used in Gutjahr et al. (2017)[15], which is adapted as the initial surface ocean pH forcing in the "double inversion" experiment.

First, the "double-inversion" modeling takes the observed pH data, which constrains the flux and magnitude of $CO_2$ emissions, and the observed $\delta^{13}C$ values of the dissolved inorganic carbon of the surface ocean, which simultaneously determines the source of the emitted carbon by computing the $\delta^{13}C$ values of the carbon source. At each model time step, a pulse of $CO_2$ is emitted to the atmosphere at a given rate if the $\delta^{13}C$ value is lower than the previous time step, and the modeled surface DIC $\delta^{13}C$ values and the observed $\delta^{13}C$ values at the Kuzigongsu section are compared. If the current modeled surface DIC $\delta^{13}C$ value is higher than the data value, the $\delta^{13}C$ value of the emitted $CO_2$ is assigned a value of –100‰. In contrast, if the current modeled surface DIC $\delta^{13}C$ value is lower than the data value, the $\delta^{13}C$ value of the emitted $CO_2$ is assigned a value of 0‰. $\delta^{13}C$ values of the emitted $CO_2$ between –100‰ and 0‰ can be achieved by binning the emission fluxes in time and averaging flux-weighted $\delta^{13}C$ values. Justification for the choice of these end-member $\delta^{13}C$ values of the emitted $CO_2$ is provided in Gutjahr et al.[15]. During the experiments, cGENIE continually adjusts the rate and $\delta^{13}C$ value of emitted $CO_2$ into the atmosphere in order to simultaneously reproduce the two proxy records as a function of time. In these experiments, we assume that the POE onset occurred as a linear decline in both $\delta^{13}C$ and pH simultaneously (Fig. 4a, b). We use the same "double-inversion" methodology in both the main experiments and the sensitivity experiments, both starting from the same open-system spin-up state (Table S1).

## Sensitivity experiments and analyses

We carried out sensitivity experiments to explore the importance of the duration of the POE onset (~ 7000, ~ 1600, ~ 850, and ~ 500 years based on age model option 1, age model option 2, Bowen et al. (2016), and Babila et al. (2022), respectively) using a global compilation of marine carbonate $\delta^{13}C_{carb}$ records (Table S1; Fig. S7). We also tested the effect of larger pH decrease (i.e., –0.24 and –0.32 pH unit) in combination with each of the four assumed age model (Table S1). Additionally, we test the role of bioturbation on the carbon isotope excursion magnitude of core-top carbonates (Fig. S9).

## Data availability

The geochemical and age data generated in this study are provided in the Supplementary Information. These data are also archived in Figshare[103].

## Code availability

The code for the version of the 'muffin' release of the cGENIE Earth system model used in this paper, is tagged as v0.9.33, and is assigned a https://doi.org/10.5281/zenodo.7268917[104]. Configuration files for the specific experiments presented in the paper can be found in Figshare[103]. Details of the experiments, plus the command line needed to run each one, are given in the readme.txt file in that directory. All other configuration files and boundary conditions are provided as part of the code release. A manual detailing code installation, basic model configuration, tutorials covering various aspects of model configuration, experimental design, and output, plus the processing of results, is assigned a https://doi.org/10.5281/zenodo.7545814[105].

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

## Acknowledgements

We thank Zhilin Yang for field assistance, and Hong Su for providing technical support and Isabel M. Fendley for discussions on mercury data interpretation. S.J. and Y.W. are supported by the National Key R&D Program of China (2022YFF0800800) and NSFC grants 41888101 & 41976045 & 42206047, Y.C. is supported by NSF Award 2002370. J.W. thanks the Agouron Institute for support. B.D.A.N. thanks the NERC (contract no. NE/V003917/1) and funding from the European Research Council under the European Union's Seventh Framework Programme (FP/2007-2013) and European Research Council Grant Agreement number 340923 for partial funding of the National Environmental Isotope Facility and GC-MS, GC-C-IRMS, and HPLC-MS capabilities at the OGU in Bristol. B.D.A.N. was funded through a Royal Society Tata University Research Fellowship. A.R. acknowledges support from NSF (EAR 2121165 and MG&G 2244897). This manuscript is a contribution to IGCP 739 project.

## Author contributions

S.J. and Y.C. designed the study and interpreted the data. Y.C., M.D.P. and A.R. performed and analyzed cGENIE modeling experiments. B.D.A.N. and Y.H. led the biomarker data acquisition. Y.W., S.J. and Y.G. interpreted the calcareous nannofossil biostratigraphy and performed XRF analysis, H.W. and R.C. performed age model, J.J. and X.H. conducted sedimentology. M.I. performed clumped isotope analysis. S.J. and Y.C. wrote the manuscript with inputs from T.B., A.R., J.Z., B.D.A.N., J.W., H.W., R.C., S.Y., and M.I.

## Competing interests

The authors declare no competing interests.
