## [Transparent Peer Review file · Nature Communications]

Millennial-timescale thermogenic CO₂ release preceding the Paleocene-Eocene Thermal Maximum

Corresponding Author: Dr Ying Cui

Version 0:

Reviewer comments:

Reviewer #1

(Remarks to the Author)

The paper of Jiang et al. presented new data from a section in Xinjiang province in China across the POE and PETM. The dataset is extensive, new and really exciting, so is therefore worthy of publication in Nature Communications. However, there are a few areas where the text and discussion should be improved prior to publication, so I recommend moderate revisions.

With best regards, Morgan Jones

Important Comments:

Mercury signal.

The main issue I have with the paper is the interpretation of the mercury signal. All of the samples presented here have Hg <20 ppb and similarly low TOC contents, which means that analytical errors are going to be enhanced. On the machines I've used, Hg contents below 10 ppb have errors comparable to the concentrations, and TOC <0.2 wt% has been shown as the broad cut off for using Hg/TOC as a proxy (e.g. Grasby et al., 2019). Therefore, a lot more care needs to be taken before we can arrive at "originating from the NAIP" (lines 40.42). Hg/TOC values are almost all <30 ppb/wt%, which is well below the average for shales. Even the enriched sections during the POE and PETM are below this mean value from current data sets (Grasby et al., 2019). The authors need to address other possible avenues for disturbances to the Hg cycle. The authors mention and rule out wildfires as a possible source, but the main one that jumps out to me from their dataset is a possible change in organic matter source.

There are several indicators of enhanced terrestrial runoff in this dataset (e.g. line 124), including a negative $\delta^{18}\text{O}$ excursion and elevated C/N ratios (what does the BIT index of the GDGTs look like?). Mercury uptake into terrestrial plant biomass is inherently different to marine uptake, leading to different initial Hg/TOC ratios in each organic matter pool. Given the low Hg and TOC contents, how can the authors be sure that the signal they are observing is indeed due to changes in NAIP activity? There seem to be two distinct Hg/TOC populations in Figure S7, do the population with 10-16 ppb Hg and 0.5-1.5 wt% TOC also have high C/N ratios?

Moreover, the authors interpret the POE as being the main warming driven by intrusive activity. Volcanic and thermogenic emissions are near impossible to differentiate in the far field, but the limited near-field dataset that we do have suggests that the submarine emissions from contact metamorphism around intrusions leads to much of the Hg being scrubbed into the local water bodies, rather than released to the atmosphere (e.g. Jones et al., 2019a). Why then would the POE have the larger relative Hg/TOC excursion at Kuzigongsu? I'm not saying that the signal here isn't due to NAIP activity, but more nuanced discussion is needed in the interpretation of this dataset.

NAIP Emplacement.

North Atlantic Igneous Province (NAIP) activity has been oversimplified in this manuscript. The province was active as early as 62 Ma, and while the main acme of activity did encompass the PETM, it is not (yet) constrained to as fine a window as 56-55.6 Ma (lines 162-163). This interval is based on the start of the crystallisation of the Skaergaard intrusion as it was buried by 5-6 km of lava (Larsen & Tegner, 2006) in East Greenland. The upper constraint is the eruption of the Gronau tuff, which is linked to Ash -17 in Denmark (Storey et al., 2007a). Once corrected to the most recent Fish Canyon Tuff age estimate, this layer is dated to 55.48 ± 0.12 Ma, which gives us a ~0.5 Myr window where much of the East Greenland basalts were emplaced. However, this is just one part of a much bigger province, and the Skaergaard magma chamber was intruded into older basalts. Other parts of the province, particularly places like the Faroe Islands and the Vøring plateau, are not nearly so

well constrained, and it is unclear at the moment whether this spike in activity in East Greenland is indicative of the province as a whole, or more localised activity.

Furthermore, I strongly disagree with argument in lines 237-238. The study of Tian and Buck (2022) does not mention the NAIP once, and focuses on how the Deccan Traps and the Colombia river basalts were emplaced. LIP emplacements and the subsequent climatic impacts are different between individual LIPs (see Jones et al., 2016), and other emplacement models exist for other LIPs (e.g. Burgess et al., 2017 suggesting extrusive-intrusive-extrusive for the Siberian Traps). We know from field evidence that there is considerable NAIP volcanism prior to 56.5 Ma and after 55.6 Ma. The intrusive activity is less well temporally constrained, but the only two hydrothermal vent complexes that have been sampled and analysed contain the PETM onset within the vent infill (Frieling et al., 2016; Planke et al., 2022; Berndt et al., in review). There may have been a pulse in intrusive activity at the POE, but almost certainly also at the PETM. Many studies argue, quite convincingly, that magmatic carbon fluxes are insufficient to induce global warming on their own (e.g. Jones et al., 2019b). This section needs to be rewritten to be more nuanced in terms of what constitutes NAIP activity, how (if at all) it is possible to distinguish between volcanic and thermogenic emissions at such distance, and whether such definitive conclusions can be derived from a distal locality.

Minor comments:

Title: The title can be improved, as the PETM is by definition a period of climate instability. The strength of this paper is the well preserved POE and its relation to the PETM, so I suggest rewording to draw attention to what is new and exciting here.

Abstract: Mercury concentrations are not elevated, <20 ppb throughout the entire section is Hg-starved.

Line 41: The NAIP was not one eruption. Rephrase to “originating from North Atlantic Igneous Province activity” or similar.

Line 72 (& Line 27 in SI): The coordinates for the Kuzigongsu section point to a warehouse in Wuqia county on Google Earth. Please check the coordinates.

Lines 87-88: “The onset of the PETM is defined as the most rapid decline in $\delta^{13}\text{C}$ values in all studied substrates at 19.9 m” reads like this is the biggest excursion in this section, where I suspect you mean it is the largest of all PETM sections. I suggest you reword to “the onset of the PETM at 19.9 m displays a rapid decline in $\delta^{13}\text{C}$ values, among the largest CIEs in studied PETM substrates (McInerney & Wing, 2011)”.

Lines 94-95: Please explain why these are deemed “more likely” and “less likely”? Is it to do with the way the age models are set up, or is it that age model 2 predicting a 39 kyr PETM duration goes against a mountain of evidence from other localities?

Lines 116-117: Please add Oxford commas.

Lines 162-163: 9 million km^3 is the total volume of the NAIP emplaced between 62-54 Ma. The percentage that was emplaced between 56.0 and 55.6 Ma is going to be less than this. Suggest rewording.

Lines 185-186: The current best estimates for the onset of the PETM are either 55.93 Ma (Westerhold et al., 2017) or 56.01 Ma (Zeebe and Lourens, 2019). The Charles et al. (2011) paper provides an important ash marker horizon of 55.785 Ma, but this is during the recovery of the CIE. The shape of the CIE in Svalbard also makes it slightly difficult to pin down where in the PETM this marker horizon sits. Citations are needed for this sentence (e.g. Storey et al. 2007a; 2007b; Wilkinson et al., 2017).

Line 226: How do these model results compare to the findings of Gutjahr et al.?

Lines 235 & 284: Activity is only classified volcanic at the surface. Intrusive activity is magmatism.

Line 266: Also cite Papadomanolaki et al. (2022) here.

Line 284: Rephrase to “as a result of intrusive and/or eruptive activity from a major large igneous province.”

Author contributions: It is not clear what the last four authors contributed to the manuscript.

Figure 5: The colour palette used for this figure is not very friendly towards the colour blind, suggest changing to colours with greater contrast (see Crameri et al., 2020, for useful tips).

Supplementary lines 139-154: What were the BIT on analysed samples? Does this tell you anything about the amount of terrestrial organic matter input?

Supplementary lines 248-251: A value of 2600 kg m^3 seems a very high estimate for a density of shale. Are your samples that metamorphosed? We measured the density of the PETM clay in Denmark (paper in prep.) and got a value of 1400 kg m^3 . Admittedly this is an un lithified section, but if the value used here is correct then the section may have undergone significant compaction post-deposition. Some text addressing this, or better yet a measurement of the actual shale density,

would be useful here.

Supplementary line 254: Table S4 leaves out several recently studied sections that have much thicker complete sections than this, including Fur, Svalbard, the North Sea, and the Norwegian continental margin (e.g. Jin et al., 2022; Jones et al., 2019a; Planke et al., 2022). Perhaps tone down the language a little to say it is one of the thicker PETM sequences, or that it is one of the thickest carbonate-rich sections?

Supplementary line 287: Correct typo to “performed”

References mentioned:

Burgess, S.D., Muirhead, J.D., Bowring, S.A., 2017. Initial pulse of Siberian Traps sills as the trigger of the end-Permian mass extinction. *Nature Communications* 8, 164. doi:10.1038/s41467-017-00083-9.

Cramer, F., Shephard, G.E., Heron, P.J., 2020. The misuse of colour in science communication. *Nature communications* 11, 5444.

Frieling, J., Svensen, H.H., Planke, S., Cramwinckel, M.J., Selnes, H., Sluijs, A., 2016. Thermogenic methane release as a cause for the long duration of the PETM. *Proceedings of the National Academy of Sciences USA* 113, 12059-12064.

Grasby, S.E., Them II, T.R., Chen, Z., Yin, R., Ardakani, O.H., 2019. Mercury as a proxy for volcanic emissions in the geologic record. *Earth Science Reviews* 196, 102880.

Jin, S., Kemp, D.B., Jollet, D.W., Vieira, M., Zachos, J.C., Huang, C., Li, M., Chen, W., 2022. Large-scale, astronomically paced sediment input to the North Sea Basin during the Paleocene Eocene Thermal Maximum. *Earth and Planetary Science Letters* 579, 117340.

Jones, M.T., Jerram, D.A., Svensen, H.H., Grove, C., 2016. The effects of large igneous provinces on the global carbon and sulphur cycles. *Palaeogeography Palaeoclimatology Palaeoecology* 441, 4-21.

Jones, M.T., Percival, L.M.E., Stokke, E.W., Frieling, J., Mather, T.A., Riber, L., Schubert, B.A., Schultz, B., Tegner, C., Planke, S., Svensen, H.H., 2019a. Mercury anomalies across the Palaeocene–Eocene Thermal Maximum. *Climate of the Past* 15, 217-236.

Jones, S.M., Hoggett, M., Greene, S.E., Dunkley Jones, T., 2019b. Large Igneous Province thermogenic greenhouse gas flux could have initiated Paleocene-Eocene Thermal Maximum climate change. *Nature Communications* 10, 5547.

Larsen, R.B., Tegner, C., 2006. Pressure conditions for the solidification of the Skaergaard intrusion: Eruption of East Greenland flood basalts in less than 300,000 years. *Lithos* 92, 181-197.

Papadomanolaki, N.M., Sluijs, A., Slomp, C.P., 2022. Eutrophication and Deoxygenation Forcing of Marginal Marine Organic Carbon Burial During the PETM. *Paleoceanography and Paleoclimatology* 37, e2021PA004232.

Planke, S., Berndt, C., Alvarez Zarikian, C.A., Expedition 396 scientists, 2022. Expedition 396 Preliminary Report: Mid-Norwegian Continental Margin Magmatism. *International Ocean Discovery Program*.

Storey, M., Duncan, R., Swisher III, C., 2007a. Paleocene-Eocene Thermal Maximum and the opening of the Northeast Atlantic. *Science* 316, 587-589.

Storey, M., Duncan, R., Tegner, C., 2007b. Timing and duration of volcanism in the North Atlantic Igneous Province: Implications for geodynamics and links to the Iceland hotspot. *Chemical Geology* 241, 264-281.

Westerhold, T., Röhl, U., Frederichs, T., Agnini, C., Raffi, I., Zachos, J.C., Wilkens, R.H., 2017. Astronomical calibration of the Ypresian timescale: implications for seafloor spreading rates and the chaotic behavior of the solar system? *Climate of the Past* 13, 1129-1152.

Wilkinson, C., Ganerød, M., Hendriks, B., Eide, E., 2017. Compilation and appraisal of geochronological data from the North Atlantic Igneous Province (NAIP), in: Péron-Pinvidic, G., Hopper, J.R., Stoker, M.S., Gaina, C., Doornenbal, J.C., Funck, T., Ártung, U.E. (Eds.), *The NE Atlantic Region: A Reappraisal of Crustal Structure, Tectonostratigraphy and Magmatic Evolution*. Geological Society, London, Special Publications.

Zeebe, R.E., Lourens, L.J., 2019. Solar System chaos and the Paleocene–Eocene boundary age constrained by geology and astronomy. *Science* 365, 926-929.

Reviewer #2

(Remarks to the Author)

Jiang et al presents new records of the PETM and the precursor event named the POE from the eastern Tethys. There is a

lot of new data presented in this manuscript - multiple proxies of carbon cycling and volcanism, ocean redox, and nutrient conditions (as the authors summarize in the abstract) as well as ocean temperature (clumped isotope measurements). In addition, the authors present results from simulations of the carbon injection across the PETM and POE using the cGENIE Earth system model.

While the data are interesting, the new records raise many more questions than are addressed with this manuscript. My overall impression is that the manuscript fails to integrate the numerous data types and modeling in an effective or original manner. The content of the manuscript does not convincingly address the topic suggested by the title, which in any case is also not articulated in the text of the abstract that follows.

The title presents a very interesting question: is the climate state, presumably on millennial timescales, more or less stable than the 'background' Paleocene and Eocene climate? I say presumably because this is a key aspect of the question that is not elaborated - how do you define 'climate instability'? Over what timescale? I find only a few sentences in the main text that address the topic of the manuscript title (Lines 244-248) and argue that the use of a 'sequential regime shift detection algorithm' identifies 'multiple regime shifts in the mean and variance of the time-series.' None of this is sufficiently explained - neither the method nor the interpretation in either the main text or supplement. Out of all the records, why is this method only applied to the d13C and d18O records? Why use d18O at all if, as the authors caution, it may be 'at least partially influenced by groundwater leaching and/or increased freshwater input' (Line 123-124). How might this analysis be impacted by big changes in preservation across the section - in other words, is it valid to interpret noisier records in the body of the PETM as evidence for real climatic instability? Later in the Conclusions (Line 283) the authors suggest that 'carbon cycle feedbacks...maintained the higher pCO₂ during the body of the PETM, highlighting the climate instability.' First, it isn't clear that this study demonstrated the occurrence of carbon cycle feedbacks, and I'm not sure what evidence was meant to support this claim. Second, how is high CO₂ alone evidence for climate instability?

However, upon reading the abstract, the purpose of the manuscript instead seems to be to constrain the carbon source across the PETM and POE. The argument appears to be that volcanism was responsible for both events and organic carbon burial was an important recovery mechanism for the PETM. Neither of these arguments are novel, though much less is constrained about the POE compared to the PETM. However, the description of the findings is rather misleading because this study does not present any boron isotope data, which also has implications for the modeling conducted, as I describe below.

The authors use the double-inversion methodology of Gutjahr et al., 2017, in the cGENIE model. The main text inadequately describes what the authors have done here - they have taken d11B records from different sites and publications compared to their new d13C data (from Site 401 in the North Atlantic for the PETM and from a Maryland site for the POE). This has many unexplored consequences. For instance, have the authors considered the uncertainty in correlating these records? They've had to do this point-by-point to use records from two locations simultaneously as inversion targets. This correlation has direct impacts on the d13C of the source recovered in the model. Next, have they looked to see how well inversions of d13C from their new sites reproduce d13C curves in other locations, like equivalent to Site 401 from the Gutjahr paper? Each site is clearly impacted differently by dissolution and mixing and by combining the records used in the inversion the authors have inherited all these sources of uncertainty without doing anything to explore the implications for their results.

Based on the modeling, the authors write, 'shorter duration of the POE and less severe pH responses led us to conclude that the NAIP volcanic CO₂ emissions may have led to the occurrence of the POE.' Why does short duration and smaller pH response suggest volcanism? Instead, the diagnosed d13C source values from the double inversion are lower than for the PETM, and the authors appear to recognize this, writing that the results support 'different triggering mechanisms for these two events (Line 234-235).' I don't understand what the authors mean by suggesting that both the POE and PETM are driven by NAIP volcanism but triggered by different mechanisms. Figure 5 summarizing the model results doesn't provide any description in the caption for panels g through i for the POE inversion. There is no way that the records shown on the top in panel (g) were used to drive the entirety of the POE inversion given that no carbon emissions are diagnosed after 7k years.

Overall, there is too much data and not enough rigorous interpretation in this manuscript. I'm left with many questions and without any clear takeaways. The title, abstract, and text of the manuscript are not well matched. I envision a study that could effectively use some (perhaps not all) of the presented data to drive model simulations of the PETM and POE and compare and contrast the results in terms of the diagnosed emissions, d13C, impact on CO₂, and differential recovery mechanisms. Or, I can envision a study specifically focused on climate instability (as in the title), though this seems to require better constraints on temperature specifically.

Version 1:

Reviewer comments:

Reviewer #1

(Remarks to the Author)

The revised paper by Jiang and co-authors is a significant improvement on the first submission and is an excellent and exciting paper overall. The only area where I believe significant further investigation is required is the modelling results and the implications for a plausible source. After addressing this one moderate comment and a few minor revisions, it is worth publishing in Nature Communications. With best regards, Morgan Jones

Main comment:

Lines 266 to 272: I am a little worried about the validity of the modelling results that ascribe a purely NAIP volcanic carbon source for the POE, as 1 to 10 Pg C/yr is an enormous flux, and up to 12,300 Pg C is an enormous volume. To put these numbers into perspective, Stephen Jones et al. (2019)'s paper estimates that the total mantle-derived carbon reservoir for the entire NAIP is 14,100 Pg C, while our recent estimate puts it between 21,000 and 35,000 Pg C (M. Jones et al., 2023). These are estimates for the entire 62 to 54 Ma interval, and while much of this volume is believed to have erupted between 56 and 55.5 Ma, it would still need a phenomenal amount of magma/lava to degas in an extremely short amount of time to solely account for the POE or the PETM. Taking the approximation of 3.5 Tg C per km³ of magma as a potential degassing source (see Jones et al., 2016), a yearly flux of 1 to 10 Pg C/yr would require an eruptive / shallow intrusive flux of 286 to 2860 km³/yr of magma, which is absolutely colossal. The geochronology of the NAIP continental flood basalt sequences is not very well constrained, but the field evidence in localities such as East Greenland and the Faroe Islands shows individual lavas with evidence of hiatuses between that seem to contradict the hypothesis that large volumes of these lavas were erupted in a very short (e.g. 1–10 kyr) time periods. Therefore, the model results appear to generate unrealistic values for magmatic gas fluxes, and this discrepancy between the model findings and the NAIP magmatic carbon reservoir potential needs to be discussed in greater detail in the manuscript.

I don't think that the model needs to be rerun, but some discussion about the potential sources and differences between the model and field observations is warranted. Potential avenues include how the model setup may vary from natural conditions, and what NAIP sources may be active. Does the model cope well with an enhanced ¹²C sink (e.g. organic matter burial) that could be masking the degassing signature in atmospheric CO₂ values, perhaps? The cGENIE model is not my expertise, but I think that the paper would be significantly enhanced with some discussion around what the limitations of the model are, what potential reasons can there be between the calculated fluxes from modelling and field observations, and how future studies could constrain these uncertainties.

Minor comments:

The focus on the NAIP as a purely volcanic source is a little misleading, as numerous studies (e.g. Berndt et al., 2023) highlight the importance of sub-volcanic intrusions as a significant carbon flux across this key interval. The explosive ejection of thermogenic gases through these hydrothermal vent complexes would be a mix of magmatic and sedimentary sources. Stephen Jones et al. (2019) estimate that the partition would be 90% contact metamorphic and 10% mantle signature, and that peak emission fluxes of 0.2–0.5 Pg C/yr from thermogenic NAIP sources could have initiated the PETM. I personally think the 90:10 ratio is a little high, as they assume a thick metamorphic aureole in their calculations and do not account for magmatic degassing from deeper in the plumbing system. Therefore, the intrusive part of the NAIP can potentially be a substantial source of magmatic carbon, so I would err on the side of caution and recommend referring to it as "NAIP activity" or "emplacement of the NAIP" rather than explicitly "volcanism" throughout the text.

Line 124: Add 'a' to "and mercury content as a possible indicator of volcanism"

Line 128: Add 'evidence of' to "...represent evidence of an abrupt and significant warming..."

Lines 180–181: This explanation of Mn* would look better as an equation.

Line 192: Elements at the start of sentences should be written out in full (i.e. change Hg to Mercury here)

Line 203: For reasons stated above, I think it would be better to change "volcanism" to "activity" here.

Line 216: As above. Either change to "The C/N ratio" or "Carbon/nitrogen ratio"

Lines 229–230: The locations of hydrothermal vent complexes here is wrong. The hydrothermal vent complexes are exclusively found on the continental margins around the NAIP (modern day Northeast Atlantic Ocean), caused by igneous intrusions into sedimentary sequences. See the recent paper by Berndt et al. (2023).

Line 234: I would probably recommend not stating the estimated PETM onset age of 56.01 Ma (Zeebe and Lourens, 2019) here, as the U-Pb age of 55.785 ± 0.034 Ma from a bentonite layer that is within the PETM CIE body in Svalbard (Charles et al., 2011) does not match with the 127 kyr PETM duration presented in this paper. You can perhaps mention this discrepancy and the alternate 55.93 Ma PETM onset age (Westerhold et al., 2017) if you want to dig into what the cyclostratigraphy presented in this paper suggests for the validity of the various age models.

Line 554 (Figure 1): Does the "CIE" marked on panel (d) refer to the PETM? Please clarify

Lines 635–636 and 756–758: References numbered 12 and 51 are a repeat, the latter is the correct version.

References mentioned:

Berndt, C., Planke, S., Alvarez Zarkian, C. A., Frieling, J., Jones, M. T., Millett, J. M., Brinkhuis, H., Bünz, S., Svensen, H. H., Longman, J., Scherer, R. P., Karstens, J., Manton, B., Nelissen, M., Reed, B., Faleide, J. I., Huismans, R. S., Agarwal, A., Andrews, G. D. M., Betlem, P., Bhattacharya, J., Chatterjee, S., Christopoulou, M., Clementi, V. J., Ferré, E. C., Filina, I. Y., Guo, P., Harper, D. T., Lambart, S., Mohn, G., Nakaoka R., Tegner, C., Varela, N., Wang, M., Xu, W., and Yager, S. L.: Shallow-water hydrothermal venting linked to the Palaeocene–Eocene Thermal Maximum, *Nat. Geosci.*, <https://doi.org/10.1038/s41561-023-01246-8>, 2023.

Jones, S. M., Hoggett, M., Greene, S. E., and Dunkley Jones, T.: Large Igneous Province thermogenic greenhouse gas flux could have initiated Paleocene–Eocene Thermal Maximum climate change, *Nat. Commun.*, 10, 5547, <https://doi.org/10.1038/s41467-019-12957-1>, 2019.
(Currently reference No. 13 in this manuscript)

Jones, M. T., Jerram, D. A., Svensen, H. H., and Grove, C.: The effects of large igneous provinces on the global carbon and sulphur cycles, *Palaeogeogr. Palaeoclimatol.*, 441, 4–21, 2016.
(Ref. 64)

Jones, M.T., Stokke, E.W., Rooney, A.D., Frieling, J., Pogge von Strandmann, P.A.E., Wilson, D.J., Svensen, H.H., Planke, S., Adate, T., Thibault, N., Vickers, M.L., Mather, T.A., Tegner, C., Zuchuat, V., Schultz, B.P.: Tracing North Atlantic volcanism and seaway connectivity across the Paleocene–Eocene Thermal Maximum (PETM). *Clim. Past* 19, 1623-1652, 2023.
(Ref. 55, updated from the preprint version now it is published)

Westerhold, T., Röhl, U., Frederichs, T., Agnini, C., Raffi, I., Zachos, J. C., and Wilkens, R. H.: Astronomical calibration of the Ypresian timescale: implications for seafloor spreading rates and the chaotic behavior of the solar system?, *Clim. Past*, 13, 1129–1152, <https://doi.org/10.5194/cp-13-1129-2017>, 2017.

Zeebe, R. E. and Lourens, L. J.: Solar System chaos and the Paleocene–Eocene boundary age constrained by geology and astronomy, *Science*, 365, 926–929, 2019.
(Ref. 63)

Reviewer #2

(Remarks to the Author)

I reviewed a previous version of this manuscript, and I agree with the authors' decision to focus on the diagnosis of the likely carbon source for the POE. The additional experiments conducted as sensitivity analysis are welcome, but unfortunately I have many more unanswered questions raised by this version of the manuscript.

Overall, I don't think that a clear and succinct summary of the double inversion experiments has been presented. I do not think that the authors have clearly demonstrated that volcanic emissions can explain the POE without generating a $\delta^{13}\text{C}$ excursion that should have been observable in deep sea records. It is very challenging to interpret all experimental results by trying to combine the description of each experiment provided in a supplemental table with multiple figures with lines labeled 'Experiment 1, 2, etc.' and captions that are not re-written for each figure. It was difficult to remember what I was meant to be evaluating from each plot contrasting Figure 4 with the figures in SI. Key sensitivity analyses seem to be 1) the significance of varying the age model 2) importance of different assumptions about the pH constraint, and 3) importance of the alignment of $\delta^{13}\text{C}$ and pH targets. I guess the goal is to summarize the importance of 1) and 2) in Figure 4 and leave 3) to the SI. Overall, it would help for each subplot in each figure to identify the plotted lines in a way that clearly explains what is being varied (not 'Experiment ID ='). Importantly, comparison of the significance of different age model assumptions is hindered a bit by the use of different axis limits for the right and left-side subplots showing carbon emissions. The reported diagnosed $\delta^{13}\text{C}$ of the source (-4 to -13 per mil) (Line 267-268) is not what it looks like from Figure 4 (which is also difficult to see because the axis extends to -60 per mil).

A significant concern is the misrepresentation of the benthic $\delta^{13}\text{C}$ excursion generated in the experiments. In the main text, it is reported that the POE double inversions yield 'no obvious CIE' in the modeled deep ocean (Line 275). I don't agree with that interpretation based on what is shown in Figure S11. The benthic CIE in Fig S11a is roughly 1.6 per mil and roughly 1 per mil in Fig S11b. In panels c-f benthic $\delta^{13}\text{C}$ is still decreasing because the duration of the experiment is so short that the deep ocean has either barely seen or has not seen the full magnitude of the excursion in the surface. It is very hard to argue that one would not expect to see a 1 to 1.6 per mil excursion in the deep ocean as predicted by these experiments - how do you explain the abundance of 'hyperthermals' identified of approximately this magnitude across the Early Eocene? It is a really interesting question whether inversions of the POE CIE and pH are compatible with the supposed absence of this event from the deep sea record. Viewing, for instance, the high resolution benthic foram $\delta^{13}\text{C}$ record from ODP Site 1262 (Littler et al., 2014) there does not appear to be evidence for a -1 per mil excursion ~100 kyr prior to the PETM. Does this suggest something about the amount of time between the two events and hence the likelihood that the POE could be erased by burn-down? It is not the duration alone of an event that limits its appearance in the deep sea - it is the total size of the event that matters. (If only duration mattered, that would erroneously suggest that the impact of modern carbon emissions wouldn't be recorded in the geologic record). Some of the inversions presented here show a really size-able event - CO_2 more than doubling and masses of carbon predicted that are similar to some previously published estimates for the PETM itself - how are these 'relatively small' emissions estimates?

There are additional aspects of the analysis that are confusing to me (itemized by line below). The figures that present the age model development and the model results are difficult to read and have very small text. It is strange to leave the cyclostratigraphy to SI given the centrality of age model constraints to the interpretation of the POE via Earth system modeling. Text is duplicated in both Methods and SI but sometimes with slight differences. The SI should not repeat the Methods but provide additional explanation/discussion (for instance, an analysis of the sensitivity experiments).

Moreover, the Methods (particularly regarding the cyclostratigraphy and modeling) are incomplete. Cyclostratigraphy uses magnetic susceptibility, but MS measurements are never described. What is the resolution of MS measurements? Exactly how is the record interpolated, detrended, smoothed or filtered? 'Dynamical filter to isolate precession cycles' needs elaboration - where is the evidence that there is significant precession power in the timeseries prior to filtering the record at this frequency and hence adding power? How is a 'reasonable range of sedimentation rate' set? What are the overall constraints? Multiple software packages are described (MATLAB scripts, AnalySeries) but then the final sentence states that all analyses are conducted with Astrochron unless otherwise noted. My overall impression is that this description of cyclostratigraphic methods doesn't give the reader the ability to reproduce the results, which should be the goal.

Regarding the description of Earth system modeling, the Methods first states that the isotopic inversion target is CO₂ d¹³C (Line 503) and then later that the target is surface ocean DIC d¹³C (Line 511). What was the d¹³C record from the KZGS adjusted to in absolute values - CO₂ d¹³C or DIC d¹³C? Line 496 is duplicated in Line 523 (details of the SPINUP) with configuration variously described as late Paleocene and early Eocene. In the SI (Line 315) there is mention of different pH scenarios for both the PETM and POE inversions but these appear to have been handled differently. I think the different pH scenarios for the PETM double inversion actually have adjusted pH targets in terms of absolute value whereas the different pH scenarios for the POE adjust only the magnitude of the pH change but this is really unclear. For the PETM double inversion, what do you mean by pH is 'set to'? From Fig S8e, there is a difference in initial CO₂ across the three different pH values (suggesting the absolute pH was offset initially in each inversion from the SPINUP). Why were these different approaches taken for the PETM and POE double inversions? (In Figure 4, the different pH options for the POE inversions don't impact the initial value of CO₂, but the data shown in grey versus red dots do have differences in the absolute values of pH in addition to the magnitude of the change in pH).

Additional comments below:

Line 92 - provide the breakdown of CIE size across source for the POE as done above for the PETM?

Line 95 - in the main text, could briefly summarize what the cycle assumption is for each of the two tuned age models?

Line 144 - conceivably there could also be a significant decline in carbonate production (rather than reduced preservation alone)?

Line 151 v Line 162 - duplicate description of elevated C₂₉ hopane ratios during the PETM

Line 174 - should be 'perturbing'

Line 172 - 177 - the wording of this sentence is unclear; why specifically related 2-MeHop to nitrogen cycle perturbation in this sentence when the previous sentence referred to a host of interpretations of which changes in N fixation was one option?

Line 188 - 'global extent' is misleading; anoxia is documented in relatively restricted basins

Line 195 - if there were a direct link between Hg source and ¹³C-depleted source, why would there be a lag of more than 10 kyr?

Line 211-213 - is this meant to argue that wildfires in general cannot explain the Hg record or wildfires in the Arctic in particular?

Line 305 - didn't the double inversion provide constraints on the necessary magnitude of organic carbon burial during the POE? What is the rate and d¹³C of carbon removal in the simulated POE recovery?

Line 309 - what does it mean for the system to 'transition into the PETM'? If the argument is that the PETM was driven also by NAIP emissions, then does it matter whether or not organic carbon burial accelerated the recovery from the emissions that drove the POE? A more interesting question is to what extent CO₂ has recovered after POE emissions but before the PETM and whether that requires additional drawdown fluxes.

Methods - location of analyses is not mentioned for n-alkanes only

Figure 2 - how is the choice made where to identify the PETM onset? In the supplement, the shading to indicate the PETM covers a different interval, with the onset aligned to the initial decline in d¹³C of CaCO₃ rather than d¹³C of Corg. The overall description of the wt% CaCO₃ record in the text doesn't reflect the figure. The major decline in CaCO₃ doesn't appear to have anything to do with the timing of the POE or the PETM CIE, though an increase in CaCO₃ is aligned with the end of the PETM.

Reference:

Littler, K., Röhl, U., Westerhold, T. and Zachos, J.C., 2014. A high-resolution benthic stable-isotope record for the South Atlantic: Implications for orbital-scale changes in Late Paleocene–Early Eocene climate and carbon cycling. *Earth and Planetary Science Letters*, 401, pp.18-30.

Version 2:

Reviewer comments:

Reviewer #1

(Remarks to the Author)

This is my third review of Jiang et al. for submission to Nature Communications. The manuscript is an improvement on the previous iteration and details an exciting new POE and PETM locality. I want to be supportive of this paper because of the high-resolution nature of the Kuzigongsu outcrop and the coverage of the POE. Moreover, I appreciate the efforts that the authors have put in to advance the paper and address my previous comments and suggestions. Sadly, I still find that the modelling part of this paper is its Achilles heel and not publishable in its current state. I am not a modeller by trade, so I cannot offer advice on where the issue lies, only that the output number of 230 km³ magma /year from the NAIP for 8000 years defies all our current understanding of LIP emplacements, mantle convection, and climate feedbacks. This magma flux estimate is probably around two orders of magnitude too high for the late Thanetian. I will detail a few of the reasons why here:

1) Timing. If the POE is ~144 kyr before the PETM onset, then it is around 56.07 Ma (using Westerhold et al), 56.15 Ma using Zeebe and Lourens (2019). This predates much of the known NAIP lavas, even within uncertainty. In East Greenland, we have the precise age of the Skaergaard intrusion at 56.02 Ma (Wotzlaw et al., 2012), which is the anchor for the emplacement age of the flood basalts that buried the magma chamber as it crystallised (Larsen and Tegner, 2006). Skaergaard was intruded into the lower basalt sequence, but almost all the subsequent flood basalt activity postdates the intrusion. A similar picture emerges on the Norwegian margin. The lower series encountered on IODP 642 (see http://www-odp.tamu.edu/publications/104_SR/VOLUME/CHAPTERS/sr104_51.pdf) likely contains the PETM (IODP 396 scientists are working on this right now), but that means that the main volcanism is either syn- or post-PETM in age. The geochronology of the Faroes is less well constrained, but it is probably similar to these other localities. There are a few lava flows that might possibly be the right sort of age to coincide with the POE (see site U1566 for example; Planke et al., 2023), but these are volumetrically a very small component of the NAIP. It is certainly nowhere near “19% to 38% of the total NAIP volume” as stated in line 356. Even 0.1% of the total NAP volume in 8000 years would be an extreme scenario. Which brings us on to:

2) Volume. 230 km³(magma)/year is an insanely high flux for a single eruption, let alone a flux from a LIP that continued for 8000 years. That is the equivalent of a Yellowstone sized-eruption’s worth of material emplaced in a decade, 800 times in a row. For comparison, the 1783-84 eruption Laki that is one of the largest known basaltic eruptions was 14 km³ volume. It is also very difficult to envision a mantle convection model that can emplace such a large volume in such a short space of time, go into almost a complete period of quiescence, then restart at similar flux levels for the PETM. Therefore, these estimated fluxes are well beyond the realms of possibility.

3) Contemporaneous effects. Volcanoes don’t just emit carbon, and some co-emitted volatiles would have a significant climate impact. Emplacing 1.8 x 10⁶ km³ magma in 8000 years would release considerable volumes of such volatiles. Take sulfur for instance, using the 2014-15 eruption of Holuhraun in Iceland as an analogue for some quick back of the envelope calculations to illustrate this point. The Holuhraun eruption was 1.6 km³ magma volume, with estimated emissions of 6.7 Tg SO₂ (Carboni et al., 2019). This is roughly 3.35 Tg S, and 2.09 Tg S / km³ magma. Scaling this up to the proposed flux (230 km³/yr) gives a sulfur flux of 482 Tg S/yr. Comparing this to pre-industrial cycle would increase the total atmospheric S flux by more than an order of magnitude (sea spray = 13-36 Mt S/yr; volcanoes = 6.5 – 10.5 Mt S/yr; 1Mt = 1 Tg), and nearly a 50x increase on volcanic fluxes (see Jones et al., 2016 for discussion). Experiencing these conditions for 8000 years would have catastrophic effects on terrestrial and freshwater ecosystems in particular, through a sustained volcanic winter and increase in acid rain. To reuse the Laki example, that eruption caused crop failures in the Americas and Europe and mass famine in Iceland from one year of volcanic winter. There just isn’t the evidence in the geological record for this type of extinction-level disturbance during the POE, nor really during the PETM.

This leaves us with the conclusion that the model presented cannot be representative of the POE. This could be due to incorrect input parameters, issues with the simplified nature of the model itself, unidentified sources/sinks, or some combination thereof. This also calls into question the assertion that the PETM is largely driven by volcanic CO₂, if Gutjahr et al. (2017) used similar model parameters as presented here. The main issue is that the only NAIP emissions that are likely to be capable of producing such large fluxes on sub-millennial timescales are thermogenic sources (cf. S. Jones et al., 2019). The only way that can be reconciled with the presented model is if the volatile flux from hydrothermal vents was magma-dominated over contact metamorphic-dominated and that many of the vents were active ~100 kyr before the first known vent activity. Even then, the fluxes required are colossal, which suggests we are missing other potential sources or processes that are not accounted for in the model set up. Unfortunately I don’t know what the best way forward is, as I do not have the expertise to suggest where the model is falling short. However, unless the model can produce a POE that is grounded in realistic NAIP fluxes, I don’t think it should be included in this paper.

With best regards, Morgan Jones

Minor comments:

Line 90 (and 119, ++): I don’t see the benefit of shortening Kuzigongsu to KZGS, it doesn’t significantly shorten the text and adds a barrier to readability. I suggest just using Kuzigongsu to describe the locality.

Line 142: Is the ‘≥’ symbol correct? If the water depth is from around 30 m up to 50 m, then the symbol should be ‘≤’ instead.

Lines 158-162: This is a long sentence. I suggest you break in two to improve readability (I would break it into “...same

depth. However, the thermal maturity...”

Line 189: Suggest splitting this sentence too and fixing a slight grammar typo in doing so: “...as a result of biogeochemical changes. This is similar to observations of other major carbon cycle perturbations...”

Line 244: Change “receive” to “received”.

References mentioned:

- Carboni, E., Mather, T.A., Schmidt, A., Grainger, R.G., Pfeffer, M.A., Ialongo, I., Theys, N. (2019). Satellite-derived sulfur dioxide (SO₂) emissions from the 2014–2015 Holuhraun eruption (Iceland). *Atmospheric Chemistry and Physics* 19 (7), 4851–4862.
- Gutjahr, M., Ridgwell, A., Sexton, P. F., Anagnostou, E., Pearson, P. N., Pälike, H., Norris, R. D., Thomas, E., and Foster, G. L. (2017). Very large release of mostly volcanic carbon during the Palaeocene–Eocene Thermal Maximum, *Nature*, 548, 573–577.
- Jones, S.M., Hoggett, M., Greene, S.E., and Dunkley Jones, T. (2019) Large Igneous Province thermogenic greenhouse gas flux could have initiated Paleocene-Eocene Thermal Maximum climate change, *Nat. Commun.*, 10, 5547, <https://doi.org/10.1038/s41467-019-12957-1>.
- Larsen, R. B. and Tegner, C. (2006). Pressure conditions for the solidification of the Skaergaard intrusion: Eruption of East Greenland flood basalts in less than 300,000 years, *Lithos*, 92, 181–197.
- Planke, S., Berndt, C., Alvarez Zarikian, C.A., Agarwal, A., Andrews, G.D.M., Betlem, P., Bhattacharya, J., Brinkhuis, H., Chatterjee, S., Christopoulou, M., Clementi, V.J., Ferré, E.C., Filina, I.Y., Frieling, J., Guo, P., Harper, D.T., Jones, M.T., Lambart, S., Longman, J., Millett, J.M., Mohn, G., Nakaoka, R., Scherer, R.P., Tegner, C., Varela, N., Wang, M., Xu, W., and Yager, S.L. (2023). Site U1566. In Planke, S., Berndt, C., Alvarez Zarikian, C.A., and the Expedition 396 Scientists, *Mid-Norwegian Margin Magmatism and Paleoclimate Implications. Proceedings of the International Ocean Discovery Program, 396: College Station, TX (International Ocean Discovery Program)*. <https://doi.org/10.14379/iodp.proc.396.104.2023>.
- Wotzlaw, J., Bindeman, I., Schaltegger, U., Brooks, C., and Naslund, H. (2012). High-resolution insights into episodes of crystallization, hydrothermal alteration and remelting in the Skaergaard intrusive complex, *Earth Planet. Sc. Lett.*, 355, 199–212.

Reviewer #2

(Remarks to the Author)

I reviewed a previous version of this manuscript and I appreciate the authors responses to my comments, particularly the addition of more information about the age model construction. New high-resolution PETM records, including expression of the POE, are a welcome contribution, and additional age information about the duration of the POE and between the POE and PETM is especially significant. The problem is that I am ultimately not convinced that the way the double inversion modeling approach is applied and interpreted, and the central conclusions of the manuscript about carbon forcing that result, are valid for the POE. Central to the concept of the double inversion, as applied by Gutjahr et al., 2017, across the PETM, is the understanding that the PETM d13C excursion represents a whole ocean perturbation driven by carbon addition from a source external to the ocean-atmosphere system and that the record of d11B change across the PETM in the North Atlantic, due to its consistency with the existing Pacific record, is a good estimate of the magnitude of global surface ocean acidification. Many PETM records collected over decades substantiate the argument about the whole ocean d13C perturbation, and model-data comparison of simulated pH changes at the locations corresponding to the d11B records substantiate the application of those pH changes as global constraints. I don't think that the evidence for the POE reaches the same bar, so I think too much weight is being given to interpreting the results from an experimental framework that I'm not convinced is valid. The POE has not been conclusively identified in deep-sea records, and despite the authors comparison to features in the Site 1209 and Site 1263 benthic foraminiferal records of the late Paleocene (not shown and no correlation attempted), it's not even clear that the POE is identifiable in the deep sea. This was the basis of my previous major criticism of this paper, and I don't think the revision has adequately addressed this problem. A modeled <1‰ d13C excursion is significant, and a -1.5‰ excursion doesn't compare well to a -0.4‰ benthic excursion, with no quantification of how bioturbation should impact the modeled core-top d13C excursion magnitude (moreover, the issue of experiment duration in recording the core-top excursion magnitude has not been addressed). The authors modeling also imposes a pH change greater than that reconstructed across the PETM and consequently results in a CO₂ increase that is larger than independent estimates for the PETM CO₂ change (not those from boron isotopes, see Tierney et al., 2022).

The authors are using the d11B record from Babila et al. in their inversion, but that paper provides significantly more caveats in the interpretation of d11B as pH across the POE. They write, “the determination of the absolute magnitude of the pH excursion at the POE is limited by both the uncertainty of the initial environmental conditions and the larger uncertainty associated with the laser ablation analyses generated in this study...which makes determining the absolute magnitude of the pH excursion at the POE rather uncertain.” And also, “a full propagation of uncertainty...allows us to determine that the POE pH excursion at the Maryland sites was greater than -0.08, with an upper limit that is poorly constrained but overlaps with the magnitude estimated for the main CIE observed elsewhere.” Two important points I want to emphasize from these quotes: first, the authors here chose to use something close to the poorly constrained upper limit for their pH inversion target (-0.4) in the main results presented in this text but do not provide a justification for this choice. Second, the Babila paper correctly differentiates the pH change observed in the coastal North Atlantic to the pH decline for the CIE observed elsewhere (emphasis mine). I can think of a handful of reasons that the changes in pH near coastal sites and/or more restricted areas (as in this study location) would differ from the open ocean pH change. Changes in freshwater input, for instance, could account for some amount of local pH change. Hence using a -0.4 pH change in the double inversion seems to me as though the authors are choosing their forcing in order to arrive at a particular result. I appreciate that the authors have performed sensitivity analyses, so I think this entire discussion could be reworked largely using experiments already

conducted, even though the authors did not test the minimum pH change estimate of Babila et al. (-0.08). I don't think the "preferred scenario" is well-justified. Moreover, shouldn't the discussion acknowledge and explain contrasts between their results and the modeling results presented by Babila et al., who evaluated a large range of pH changes and POE duration estimates, including much shorter durations than the minimum 1000 years tested here? Also, could the authors provide more discussion of why these age models differ from the age model developed for the Bighorn Basin terrestrial record from Bowen et al. 18? (The authors describe their estimated duration of 7 kyr as similar but slightly longer than" the Bowen estimate, but in reality their estimate is more than double the Bowen estimate, of "fewer than 2,000 years" (Bowen et al., 2018). Also, a more clear acknowledgement that Babila et al also suggested that "it is plausible that a short-lived C emission pulse preceded the main pulse of volcanic emissions" is warranted.

Also, regarding the modeled d13C target: could the authors include a plot in the supplement that shows a high-resolution image of all the d13C records across the POE alone for comparison to the target used in the inversion? Also, were the yellow dots in figure 4 actually the inversion target? Or was the inversion target a smoothed version of this curve?

Reference cited: Tierney, J.E., Zhu, J., Li, M., Ridgwell, A., Hakim, G.J., Poulsen, C.J., Whiteford, R.D., Rae, J.W. and Kump, L.R., 2022. Spatial patterns of climate change across the Paleocene–Eocene Thermal Maximum. *Proceedings of the National Academy of Sciences*, 119(42), p.e2205326119.

Line 57 - add reference to Tierney et al., 2022.

Line 96 – 'The' primary d13Ccarb signal...

Line 113 – 'average' rather than 'are averaging'

Line 109: more about the age model – the 1.2 – 1.9 m peaks show the strongest power? These are well-identified throughout each of the POE and PETM? There is error on the duration between the POE and PETM of +/- 1 precession cycle but no error within the PETM? Presumably the 21-kyr duration estimates are consistent with identification of a single precession cycle, so please indicate that precession has an assumed 21 kyr duration on Line 112 (instead of ~20 kyr).

Line 119 – how can you tell that it is the recovery phase in particular that is truncated? What about alignment based on features in the d13C records?

Age model options 1 and 2 seem like extremes – what about 3 and 5 m cycles? What are these?

Line 151 – to support the possibility of diagenetic overprinting or freshwater influence, what is the magnitude of the d18O excursion?

Line 178 – say something about the significance of a reduction in Thaumarchaeota?

Line 255 – perhaps add a sentence here specifying how the carbon emissions rates and sources are re-assessed? As in, substituting the new KZGS d13C record but using existing d11B constraints from Gutjahr?

Line 263 – is this really a data assimilation approach?

Line 274 – the POE change in pH is 0.4 units? As in, larger than the PETM pH change of 0.3?

Line 279 – this could use a bit more explanation; specifically that you forced the model to follow the d13C excursion in DIC to the minimum but then not through the recovery, such that it was the modeled recovery mechanisms that controlled the mismatch between the modeled d13C of DIC during the recovery from the target curve...

Line 302 – please provide the size of the simulated benthic d13C excursion

Line 304 – so why not look at the downcore record?

Line 379 – is this an argument against terrestrial organic carbon burial for the POE? Why would the PETM argument hold for the POE?

Line 405-406 – are these two transient warming events? Based on the authors discussion, it seems like they are interpreting the entirety of the record as phases of NAIP carbon release.

Line 412 – why round up the organic carbon burial estimate; use the same value given above (2500-2700 Pg C)

Version 3:

Reviewer comments:

Reviewer #1

(Remarks to the Author)

The authors have done an excellent job of addressing previous concerns I had about the presentation and interpretation of their exciting data set. The revised manuscript is much more polished and well presented. I only have a couple of small comments and a few grammar suggestions, otherwise I believe it is now ready for publication.

Response to reviewers: The authors did a great job of addressing my previous comments. One slight error is that they continue to advocate for a potential PETM age of 55.53 Ma from Westerhold et al. (2007) as possible evidence for the POE to be after the current dates we have for the big upswing in NAIP activity. Multiple lines of evidence, including an ash layer within the PETM dated to 55.785 Ma (Charles et al., 2011), rule out a PETM onset age of 55.5 Ma. This doesn't make much difference to the manuscript as the focus has now shifted away from the precise timing of the POE, but I thought it was worth mentioning again.

The final piece of the puzzle that I think the manuscript needs prior to acceptance is that there is now a testable hypothesis from their findings. 0.2 to 1.3 Pg C/yr (line 304) is a sizeable flux of carbon to the ocean-atmosphere system, and if the NAIP is the dominant source, this should be discernible in the geological record. With more accurate dating of NAIP rocks, this may confirm the NAIP as a possible source. The discussion and/or conclusion could be improved with a sentence or two

along the lines of “our findings predict substantial carbon fluxes driving the POE, and refining geochronological investigations of potential sources such as the NAIP could corroborate or refute this hypothesis.”

Grammar edits (suggested additions in caps):

Line 40: Add “(NAIP)”

Line 45: Add “confirming THE global...”

Line 50: Add “driven by A largely...”

Line 51: Add “associated with sill intrusionS prior...”

Line 52: Add “involving THE release...”

Line 62: Suggest replacing “twenty millenia” with “20 kyr”

Line 70: Suggest replacing “preceding” with “prior to”

Line 71: Add “PETM ONSET with...”

Line 73: Add “preservation to surface AND SHALLOW WATER records...”

Line 74: Add “Resolving A global POE signal...”

Line 83: Add “that volcanism AND MAGMATISM may also...”

Line 84: Add “the global extent OF THE POE...”

Line 104-105: Describe what the depth scale of the stratigraphic log is relative to when first referred to in the text.

Line 181: Add “weathering AND/or higher terrestrial...”

Lines 260-264: This needs rewording. Mantle inputs of osmium have low values of 187Os/188Os due to less rhenium in the source material that has decayed to 187Os. Low Os isotopes are therefore more unradiogenic, not the other way around.

Line 345: Specify here whether you are referring to volcanic, thermogenic, or a mixture of these sources from the NAIP here.

Response to referees

Reviewer #1 (Remarks to the Author):

The paper of Jiang et al. presented new data from a section in Xinjiang province in China across the POE and PETM. The dataset is extensive, new and really exciting, so is therefore worthy of publication in Nature Communications. However, there are a few areas where the text and discussion should be improved prior to publication, so I recommend moderate revisions.

With best regards, Morgan Jones

We appreciate reviewer #1's positive feedback and thank you for your in-depth comments and suggestions, which have greatly improved the quality of the manuscript.

Important Comments:

Mercury signal.

The main issue I have with the paper is the interpretation of the mercury signal. All of the samples presented here have Hg <20 ppb and similarly low TOC contents, which means that analytical errors are going to be enhanced. On the machines I've used, Hg contents below 10 ppb have errors comparable to the concentrations, and TOC <0.2 wt% has been shown as the broad cut off for using Hg/TOC as a proxy (e.g. Grasby et al., 2019). Therefore, a lot more care needs to be taken before we can arrive at "originating from the NAIP" (lines 40.42). Hg/TOC values are almost all <30 ppb/wt%, which is well below the average for shales. Even the enriched sections during the POE and PETM are below this mean value from current data sets (Grasby et al., 2019). The authors need to address other possible avenues for disturbances to the Hg cycle. The authors mention and rule out wildfires as a possible source, but the main one that jumps out to me from their dataset is a possible change in organic matter source.

The TOC contents for our study section range from 0.05% to 3.8%, and therefore, we follow the reviewer's suggestion to exert more caution when interpreting Hg/TOC as a proxy for volcanism. We add new discussions on the various processes that may have led to changes in the Hg/TOC ratios across the POE and the PETM: "However, because of the overall low Hg concentrations at the KZGS section, establishing a link between NAIP and the Hg peaks is not straightforward. The low Hg from the KZGS site is likely due to dilution by carbonate and detrital inputs, being more distant from the Hg source, and/or Hg transport via oceanic waters rather than global atmospheric transport¹² (Fig. 3). Increased Hg concentrations across the POE and the PETM compared to background values suggest that multiple possible sources and processes may have been at play in addition to the NAIP volcanism. For example, changes in Hg concentrations in the sedimentary records can be caused by changes in river runoff, weathering, transport of terrestrial materials, primary productivity, source of organic matter, and post-depositional processes (e.g., diagenesis and dissolution)¹², which could become more important at the KZGS section because of its restricted carbonate platform setting⁴⁵. Ocean anoxia and changes in organic matter preservation and transport cannot fully account for the excess Hg as shown by the steeper Hg gradient to TOC within the PETM and POE interval at our site (Fig. 3).

Moreover, Hg fluxes associated with wildfire in the Arctic region⁴⁶ are far less than the Hg fluxes associated with a LIP event⁴⁷, and therefore cannot provide sufficient Hg into the study site. Principal component analysis (PCA) suggests that Hg is most closely related to C/N ratio (higher C/N ratio indicates more terrestrial organic source) and $\delta^{13}\text{C}_{\text{org}}$, which reflect changes in source of organic matter and ^{13}C -depleted CO_2 emissions (Fig. 3). C/N ratio exhibits no change across the POE, suggesting the increase in Hg and Hg/TOC ratio is unrelated to changes in source of organic matter. On the other hand, C/N ratio shows large increase across the PETM, which indicates that changes in source of organic matter may have contributed to the increased Hg concentrations. These potential processes do not preclude volcanic involvement, however, especially via more complex pathways than simple atmospheric loading and deposition⁴⁷. Despite these potential complex sources of Hg, we cannot completely exclude direct and indirect involvement of the NAIP in driving the Hg changes in the study section⁴⁸. For example, the NAIP was active as early as 62 Ma, and its peak activity may have encompassed the POE and the PETM⁴⁹⁻⁵². Enrichment in radiogenic Os isotopes suggests that the sediments from the Arctic and Peri-Tethys receive radiogenic materials from volcanic source⁵³. Furthermore, hydrothermal vent complexes in the circum-Arctic and the Tethyan region suggest large emissions of Hg ^{51,54,55}, further supporting that the NAIP volcanic activities that encompassed the POE and the PETM can at least partially explain the observed Hg records. Geological evidence also supports increased activity of the NAIP, which encompassed both the POE and the PETM⁴⁹⁻⁵¹ (PETM age: 56.01 Ma⁵⁶).”

Revised Figure S6. a) Cross plot of Hg concentration (ppb) with TOC (%) (b) PCA results that show the relative associations of sedimentary Hg with C/N ratios, $\delta^{13}\text{C}_{\text{org}}$ and other geochemical proxies.

There are several indicators of enhanced terrestrial runoff in this dataset (e.g. line 124), including a negative $\delta^{18}\text{O}$ excursion and elevated C/N ratios (what does the BIT index of the GDGTs look like?). Mercury uptake into terrestrial plant biomass is inherently different to marine uptake, leading to different initial Hg/TOC ratios in each organic matter pool. Given the low Hg and TOC contents, how can the authors be sure that the signal they are observing is indeed due to changes in NAIP activity? There seem to be two distinct Hg/TOC populations in Figure S7, do the population with 10-16 ppb Hg and 0.5-1.5 wt% TOC also have high C/N ratios? Moreover, the authors interpret the POE as being the main warming driven by

intrusive activity. Volcanic and thermogenic emissions are near impossible to differentiate in the far field, but the limited near-field dataset that we do have suggests that the submarine emissions from contact metamorphism around intrusions leads to much of the Hg being scrubbed into the local water bodies, rather than released to the atmosphere (e.g. Jones et al., 2019a). Why then would the POE have the larger relative Hg/TOC excursion at Kuzigongsu? I'm not saying that the signal here isn't due to NAIP activity, but more nuanced discussion is needed in the interpretation of this dataset.

Thank you for these insightful suggestions. The GDGTs in this section are weak and sometimes completely absent, brGDGTs especially are absent in most samples, hence we focus on the two main isoGDGTs only: isoGDGT-0 and crenarchaeol. Where brGDGTs are present, BIT is low with an average value of 0.05 ± 0.18 across the section. Overall the brGDGTs are too weak in abundance to make any robust interpretations about changes in terrestrial run-off and hence we do not discuss this data in our manuscript. We added new discussions on why the POE have larger relative Hg/TOC excursion at KZGS (see above). We also revised Fig. 3 to show the relationship between Hg, TOC and C/N ratio (an indicator of sources of organic matter), which show that most data associated with population with 10-16 ppb Hg and 0.5-1.5 wt% TOC also have high C/N ratios ($C/N > 10$), in particular for those with wt% TOC values greater than 1. PCA results also support C/N ratio and $\delta^{13}C_{org}$, indicators of sources of organic matter and ^{13}C -depleted CO_2 emissions, exert primary controls on the Hg concentrations.

NAIP Emplacement.

North Atlantic Igneous Province (NAIP) activity has been oversimplified in this manuscript. The province was active as early as 62 Ma, and while the main acme of activity did encompass the PETM, it is not (yet) constrained to as fine a window as 56-55.6 Ma (lines 162-163). This interval is based on the start of the crystallisation of the Skaergaard intrusion as it was buried by 5-6 km of lava (Larsen & Tegner, 2006) in East Greenland. The upper constraint is the eruption of the Gronau tuff, which is linked to Ash -17 in Denmark (Storey et al., 2007a). Once corrected to the most recent Fish Canyon Tuff age estimate, this layer is dated to 55.48 ± 0.12 Ma, which gives us a ~0.5 Myr window where much of the East Greenland basalts were emplaced. However, this is just one part of a much bigger province, and the Skaergaard magma chamber was intruded into older basalts. Other parts of the province, particularly places like the Faroe Islands and the Vøring plateau, are not nearly so well constrained, and it is unclear at the moment whether this spike in activity in East Greenland is indicative of the province as a whole, or more localised activity.

We now follow the reviewer's suggestion to add more discussions and related references on the NAIP activity. The revised text now reads as the following: "Despite these potential complex sources of Hg, we cannot completely exclude direct and indirect involvement of the NAIP in driving the Hg changes in the study section⁴⁸. For example, the NAIP was active as early as 62 Ma, and its peak activity may have encompassed the POE and the PETM⁴⁹⁻⁵². Enrichment in radiogenic Os isotopes suggests that the sediments from the Arctic and Peri-Tethys receive radiogenic materials from volcanic source⁵³. Furthermore, hydrothermal vent complexes in the circum-Arctic and the Tethyan region suggest large emissions of Hg^{51,54,55}, further supporting that the NAIP volcanic activities that encompassed the

POE and the PETM can at least partially explain the observed Hg records. Geological evidence also supports increased activity of the NAIP, which encompassed both the POE and the PETM⁴⁹⁻⁵¹ (PETM age: 56.01 Ma⁵⁶).”

Furthermore, I strongly disagree with argument in lines 237-238. The study of Tian and Buck (2022) does not mention the NAIP once, and focuses on how the Deccan Traps and the Colombia river basalts were emplaced.

Thank you for pointing this out, the discussion related to Tian and Buck (2022) is now removed.

LIP emplacements and the subsequent climatic impacts are different between individual LIPs (see Jones et al., 2016), and other emplacement models exist for other LIPs (e.g. Burgess et al., 2017 suggesting extrusive-intrusive-extrusive for the Siberian Traps). We know from field evidence that there is considerable NAIP volcanism prior to 56.5 Ma and after 55.6 Ma. The intrusive activity is less well temporally constrained, but the only two hydrothermal vent complexes that have been sampled and analysed contain the PETM onset within the vent infill (Frieling et al., 2016; Planke et al., 2022; Berndt et al., in review). There may have been a pulse in intrusive activity at the POE, but almost certainly also at the PETM. Many studies argue, quite convincingly, that magmatic carbon fluxes are insufficient to induce global warming on their own (e.g. Jones et al., 2019b). This section needs to be rewritten to be more nuanced in terms of what constitutes NAIP activity, how (if at all) it is possible to distinguish between volcanic and thermogenic emissions at such distance, and whether such definitive conclusions can be derived from a distal locality.

Thank you for these great suggestions. This section is now rewritten to include more discussions on what constitutes NAIP activity and how we estimate the likely contributions of volcanic versus thermogenic sources. The revised text now reads as the following:

“Despite these potential complex sources of Hg, we cannot completely exclude direct and indirect involvement of the NAIP in driving the Hg changes in the study section⁴⁸. For example, the NAIP was active as early as 62 Ma, and its peak activity may have encompassed the POE and the PETM⁴⁹⁻⁵². Enrichment in radiogenic Os isotopes suggests that the sediments from the Arctic and Peri-Tethys receive radiogenic materials from volcanic source⁵³. Furthermore, hydrothermal vent complexes in the circum-Arctic and the Tethyan region suggest large emissions of Hg^{51,54,55}, further supporting that the NAIP volcanic activities that encompassed the POE and the PETM can at least partially explain the observed Hg records. Geological evidence also supports increased activity of the NAIP, which encompassed both the POE and the PETM⁴⁹⁻⁵¹ (PETM age: 56.01 Ma⁵⁶).”

Minor comments:

Title: The title can be improved, as the PETM is by definition a period of climate instability. The strength of this paper is the well preserved POE and its relation to the PETM, so I suggest rewording to draw attention to what is new and exciting here.

The title is changed to better reflect the conclusion of this research. The revised title now reads: “Millennial-timescale volcanic CO₂ release prior to the PETM.”

Abstract: Mercury concentrations are not elevated, <20 ppb throughout the entire section is Hg-starved.

The relevant text is now revised to read: “Although the entire section is Hg-starved, peak Hg concentrations coincide with both the POE and the PETM, supporting direct and indirect involvement of the North Atlantic Igneous Province activity.”

Line 41: The NAIP was not one eruption. Reword to “originating from North Atlantic Igneous Province activity” or similar.

Changes are made following the reviewer’s suggestion: “... supporting direct and indirect involvement of the North Atlantic Igneous Province activity ...”

Line 72 (& Line 27 in SI): The coordinates for the Kuzigongsu section point to a warehouse in Wuqia county on Google Earth. Please check the coordinates.

The coordinates for the Kuzigongsu section are revised: “39°45'10" N, 75°17'29" E”.

Lines 87-88: “The onset of the PETM is defined as the most rapid decline in $\delta^{13}\text{C}$ values in all studied substrates at 19.9 m” reads like this is the biggest excursion in this section, where I suspect you mean it is the largest of all PETM sections. I suggest you reword to “the onset of the PETM at 19.9 m displays a rapid decline in $\delta^{13}\text{C}$ values, among the largest CIEs in studied PETM substrates (McInerney & Wing, 2011)”.

Changes are made following the reviewer’s suggestion: “...the onset of the PETM at 19.9 m displays a rapid decline in the $\delta^{13}\text{C}$ values, among the largest CIEs in studied PETM substrates¹”

Lines 94-95: Please explain why these are deemed “more likely” and “less likely”? Is it to do with the way the age models are set up, or is it that age model 2 predicting a 39 kyr PETM duration goes against a mountain of evidence from other localities?

Further discussion is provided to explain why age model 2 is less likely. The revised text now reads: “Age model 2 is less likely because the predicted 39 kyr PETM duration goes against several lines of evidence from other localities suggesting the PETM duration ranges from 170 to 220 kyr^{27,28}”

Lines 116-117: Please add oxford commas.

Oxford commas are added after “organic biomarker” and “nutrient”: “These proxies include C/N ratios as indications of organic matter source, wt.% CaCO₃ as a proxy for ocean acidification and detrital dilution, trace element geochemistry, organic biomarker, and mercury content as redox, nutrient, and volcanism indicators.”

Lines 162-163: 9 million km³ is the total volume of the NAIP emplaced between 62-54 Ma. The percentage that was emplaced between 56.0 and 55.6 Ma is going to be less than this. Suggest rewording.

This sentence is reworded following reviewer's suggestion. The revised text now reads: "initiated at *ca.* 62 Ma and ended at *ca.* 54 Ma⁴¹ with a volume of ~9 million cubic km⁴²"

Lines 185-186: The current best estimates for the onset of the PETM are either 55.93 Ma (Westerhold et al., 2017) or 56.01 Ma (Zeebe and Lourens, 2019). The Charles et al. (2011) paper provides an important ash marker horizon of 55.785 Ma, but this is during the recovery of the CIE. The shape of the CIE in Svalbard also makes it slightly difficult to pin down where in the PETM this marker horizon sits. Citations are needed for this sentence (e.g. Storey et al. 2007a; 2007b; Wilkinson et al., 2017).

We changed the age of the onset of the PETM to 56.01 Ma following Zeebe and Lourens (2019) and revised the text accordingly. The new text reads: "Geological evidence also supports increased activity of the NAIP, which encompassed both the POE and the PETM⁴⁹⁻⁵¹ (PETM age: 56.01 Ma⁵⁶)." These additional citations are now included in this sentence.

Line 226: How does these model results compare to the findings of Gutjahr et al.?

The PETM model results are summarized in the new Table S5, and it is broadly similar to the results of Gutjahr et al. (2017). Because the PETM has been extensively modeled, we shifted our focus to the POE, and provided detailed sensitivity analyses on the carbon source, emission rate and total amount of the emitted carbon (revised Fig. 4 and Table S5).

Lines 235 & 284: Activity is only classified volcanic at the surface. Intrusive activity is magmatism.

Thank you for pointing this out. We now changed "intrusive volcanism" as "intrusive activity".

Line 266: Also cite Papadomanolaki et al. (2022) here.

This reference is now cited.

Line 284: Reword to "as a result of intrusive and/or eruptive activity from a major large igneous province."

This sentence is now reworded following reviewer's suggestions: "highlighting the climate instability as a result of intrusive and/or eruptive activity from a major large igneous province"

Author contributions: It is not clear what the last four authors contributed to the manuscript.

We made clear what each author contributed to the manuscript: “S.J. and Y.C. designed the study and interpreted the data. Y.C. and M.D.P. performed cGENIE modeling experiments. D.N. and Y.H. performed organic geochemistry. Y.W., S.J. and Y.G. examined the calcareous nannofossil biostratigraphy and performed XRF analysis, H.W. and R.C. performed the cyclostratigraphy, J.J. and X.H. conducted microfacies analysis. M.I. performed clumped isotope analysis. S.J. and Y.C. wrote the manuscript with contributions from T.B., J.Z., J.W., and A.R.”

Figure 5: The colour palate used for this figure is not very friendly towards the colour blind, suggest changing to colours with greater contrast (see Crameri et al., 2020, for useful tips).

Thank you for this suggestion. The colour palate is now changed to be more friendly towards the colour blind following Crameri et al. (2020). The revised Fig. 5 (now Fig. 4) is shown below:

Supplementary lines 139-154: What were the BIT on analysed samples? Does this tell you anything about the amount of terrestrial organic matter input?

Similar to the reply to comment above: The GDGTs in this section are weak and sometimes completely absent, brGDGTs especially are absent in most samples, hence we focus on the two main isoGDGTs only: isoGDGT-0 and crenarchaeol. Where brGDGTs are present, BIT is low with an average value of 0.05 ± 0.18 across the section. Overall the brGDGTs are too weak in abundance to make any robust interpretations about changes in terrestrial run-off and hence we do not discuss this data in our manuscript.

Supplementary lines 248-251: A value of 2600 kg m^{-3} seems a very high estimate for a density of shale. Are your samples that metamorphosed? We measured the density of the PETM clay in Denmark (paper in prep.) and got a value of 1400 kg m^{-3}

$m^{>3}$. Admittedly this is an unlithified section, but if the value used here is correct then the section may have undergone significant compaction post-deposition. Some text addressing this, or better yet a measurement of the actual shale density, would be useful here.

We measured the density of the rock samples in the Kuzigongsu section, and got an average value of 2270 kg m^{-3} for mudstone samples ($n = 5$), which is higher than those of the PETM clay in Denmark as the reviewer mentioned. The calculation of organic carbon burial using the measured density is redone using these new density values.

Supplementary line 254: Table S4 leaves out several recently studied sections that have much thicker complete sections than this, including Fur, Svalbard, the North Sea, and the Norwegian continental margin (e.g. Jin et al., 2022; Jones et al., 2019a; Planke et al., 2022). Perhaps tone down the language a little to say it is one of the thicker PETM sequences, or that it is one of the thickest carbonate-rich sections?

We now tone down the language to say it is one of the thickest carbonate-rich sections. The revised sentence now reads: “Given that the POE and the PETM span 1.0 m and 10.6 m, respectively, at the studied section, the KZGS section is considered as one of the thickest carbonate-rich sections”. These recent references are also added in Table S4.

Supplementary line 287: Correct typo to “performed”

This typo is now corrected to “performed”.

References mentioned:

Burgess, S.D., Muirhead, J.D., Bowring, S.A., 2017. Initial pulse of Siberian Traps sills as the trigger of the end-Permian mass extinction. *Nature Communications* 8, 164. doi:10.1038/s41467-017-00083-9.

Cramer, F., Shephard, G.E., Heron, P.J., 2020. The misuse of colour in science communication. *Nature communications* 11, 5444.

Frieling, J., Svensen, H.H., Planke, S., Cramwinckel, M.J., Selnes, H., Sluijs, A., 2016. Thermogenic methane release as a cause for the long duration of the PETM. *Proceedings of the National Academy of Sciences USA* 113, 12059-12064.

Grasby, S.E., Them II, T.R., Chen, Z., Yin, R., Ardakani, O.H., 2019. Mercury as a proxy for volcanic emissions in the geologic record. *Earth Science Reviews* 196, 102880.

Jin, S., Kemp, D.B., Jollet, D.W., Vieira, M., Zachos, J.C., Huang, C., Li, M., Chen, W., 2022. Large-scale, astronomically paced sediment input to the North Sea Basin during the Paleocene Eocene Thermal Maximum. *Earth and Planetary Science Letters* 579, 117340.

- Jones, M.T., Jerram, D.A., Svensen, H.H., Grove, C., 2016. The effects of large igneous provinces on the global carbon and sulphur cycles. *Palaeogeography Palaeoclimatology Palaeoecology* 441, 4-21.
- Jones, M.T., Percival, L.M.E., Stokke, E.W., Frieling, J., Mather, T.A., Riber, L., Schubert, B.A., Schultz, B., Tegner, C., Planke, S., Svensen, H.H., 2019a. Mercury anomalies across the Palaeocene–Eocene Thermal Maximum. *Climate of the Past* 15, 217-236.
- Jones, S.M., Hoggett, M., Greene, S.E., Dunkley Jones, T., 2019b. Large Igneous Province thermogenic greenhouse gas flux could have initiated Paleocene-Eocene Thermal Maximum climate change. *Nature Communications* 10, 5547.
- Larsen, R.B., Tegner, C., 2006. Pressure conditions for the solidification of the Skaergaard intrusion: Eruption of East Greenland flood basalts in less than 300,000 years. *Lithos* 92, 181-197.
- Papadomanolaki, N.M., Sluijs, A., Slomp, C.P., 2022. Eutrophication and Deoxygenation Forcing of Marginal Marine Organic Carbon Burial During the PETM. *Paleoceanography and Paleoclimatology* 37, e2021PA004232.
- Planke, S., Berndt, C., Alvarez Zarikian, C.A., Expedition 396 scientists, 2022. Expedition 396 Preliminary Report: Mid-Norwegian Continental Margin Magmatism. International Ocean Discovery Program.
- Storey, M., Duncan, R., Swisher III, C., 2007a. Paleocene-Eocene Thermal Maximum and the opening of the Northeast Atlantic. *Science* 316, 587-589.
- Storey, M., Duncan, R., Tegner, C., 2007b. Timing and duration of volcanism in the North Atlantic Igneous Province: Implications for geodynamics and links to the Iceland hotspot. *Chemical Geology* 241, 264-281.
- Westerhold, T., Röhl, U., Frederichs, T., Agnini, C., Raffi, I., Zachos, J.C., Wilkens, R.H., 2017. Astronomical calibration of the Ypresian timescale: implications for seafloor spreading rates and the chaotic behavior of the solar system? *Climate of the Past* 13, 1129-1152.
- Wilkinson, C., Ganerød, M., Hendriks, B., Eide, E., 2017. Compilation and appraisal of geochronological data from the North Atlantic Igneous Province (NAIP), in: Péron-Pinvidic, G., Hopper, J.R., Stoker, M.S., Gaina, C., Doornenbal, J.C., Funck, T., Ártung, U.E. (Eds.), *The NE Atlantic Region: A Reappraisal of Crustal Structure, Tectonostratigraphy and Magmatic Evolution*. Geological Society, London, Special Publications.
- Zeebe, R.E., Lourens, L.J., 2019. Solar System chaos and the Paleocene–Eocene boundary age constrained by geology and astronomy. *Science* 365, 926-929.

Reviewer #2 (Remarks to the Author):

Jiang et al presents new records of the PETM and the precursor event named the POE from the eastern Tethys. There is a lot of new data presented in this manuscript - multiple proxies of carbon cycling and volcanism, ocean redox, and nutrient conditions (as the authors summarize in the abstract) as well as ocean temperature (clumped isotope measurements). In addition, the authors present results from simulations of the carbon injection across the PETM and POE using the cGENIE Earth system model.

While the data are interesting, the new records raise many more questions than are addressed with this manuscript. My overall impression is that the manuscript fails to integrate the numerous data types and modeling in an effective or original manner. The content of the manuscript does not convincingly address the topic suggested by the title, which in any case is also not articulated in the text of the abstract that follows.

We wish to express our appreciation for your in-depth comments and suggestions, which have greatly improved the quality of the manuscript. The content of the manuscript has been revised substantially to better reflect the new title: “Millennial-timescale volcanic CO₂ release prior to the PETM”, and articulated in the text of the abstract following the reviewer’s suggestions. We list these changes below.

The title presents a very interesting question: is the climate state, presumably on millennial timescales, more or less stable than the ‘background’ Paleocene and Eocene climate? I say presumably because this is a key aspect of the question that is not elaborated - how do you define ‘climate instability?’ Over what timescale? I find only a few sentences in the main text that address the topic of the manuscript title (Lines 244-248) and argue that the use of a ‘sequential regime shift detection algorithm’ identifies ‘multiple regime shifts in the mean and variance of the time-series.’ None of this is sufficiently explained - neither the method nor the interpretation in either the main text or supplement. Out of all the records, why is this method only applied to the d13C and d18O records? Why use d18O at all if, as the authors caution, it may be ‘at least partially influenced by groundwater leaching and/or increased freshwater input’ (Line 123-124). How might this analysis be impacted by big changes in preservation across the section - in other words, is it valid to interpret noisier records in the body of the PETM as evidence for real climatic instability? Later in the Conclusions (Line 283) the authors suggest that ‘carbon cycle feedbacks...maintained the higher pCO₂ during the body of the PETM, highlighting the climate instability.’ First, it isn’t clear that this study demonstrated the occurrence of carbon cycle feedbacks, and I’m not sure what evidence was meant to support this claim. Second, how is high CO₂ alone evidence for climate instability?

Thank you for the reviewer’s detailed comments. Based on the concerns raised by the reviewer, we shift the focus of this manuscript to the novelty of the POE and its relationship with the PETM (as reviewer #1 suggested and reviewer #2 hinted), and removed the discussions on climate instability. Our new focus is the Earth system modeling results on the POE carbon emissions, and the changes in nutrient conditions in the eastern Tethys supported by elemental proxies and biomarker data.

However, upon reading the abstract, the purpose of the manuscript instead seems to be to constrain the carbon source across the PETM and POE. The argument appears to be that volcanism was responsible for both events and organic carbon burial was an important recovery mechanism for the PETM. Neither of these arguments are novel, though much less is constrained about the POE compared to the PETM. However, the description of the findings is rather misleading because this study does not present any boron isotope data, which also has implications for the modeling conducted, as I describe below.

We apologize for the oversight in the confusion of the main message in the abstract. We now substantially revised the abstract to better reflect the main conclusions and the new title.

The authors use the double-inversion methodology of Gutjahr et al., 2017, in the cGENIE model. The main text inadequately describes what the authors have done here - they have taken $\delta^{11}\text{B}$ records from different sites and publications compared to their new $\delta^{13}\text{C}$ data (from Site 401 in the North Atlantic for the PETM and from a Maryland site for the POE). This has many unexplored consequences. For instance, have the authors considered the uncertainty in correlating these records? They've had to do this point-by-point to use records from two locations simultaneously as inversion targets. This correlation has direct impacts on the $\delta^{13}\text{C}$ of the source recovered in the model. Next, have they looked to see how well inversions of $\delta^{13}\text{C}$ from their new sites reproduce $\delta^{13}\text{C}$ curves in other locations, like equivalent to Site 401 from the Gutjahr paper? Each site is clearly impacted differently by dissolution and mixing and by combining the records used in the inversion the authors have inherited all these sources of uncertainty without doing anything to explore the implications for their results.

Thank you to the reviewer for raising these questions. The uncertainties of correlating records from different sites are now taken into consideration. New model sensitivity analyses focused on the POE are conducted to assess the effects of the uncertainties associated with the alignment of the $\delta^{13}\text{C}$ and pH records between different sites. The sensitivity results are shown in Figs. S9-12 (see below). The sensitivity test results are also highlighted in Table S5a, which suggests that the average $\delta^{13}\text{C}$ values of the source are consistent with mostly volcanic CO_2 signal.

Figure S11.

Based on the modeling, the authors write, ‘shorter duration of the POE and less severe pH responses led us to conclude that the NAIP volcanic CO_2 emissions may have led to the occurrence of the POE.’ Why does short duration and smaller pH response suggest volcanism? Instead, the diagnosed $\delta^{13}\text{C}$ source values from the double inversion are lower than for the PETM, and the authors appear to recognize this, writing that the results support ‘different triggering mechanisms for these two events (Line 234-235).’ I don’t understand what the authors mean by suggesting that both the POE and PETM are driven by NAIP volcanism but triggered by different mechanisms. Figure 5 summarizing the model results doesn’t provide any description in the caption for panels g through l for the POE inversion. There is no way that the records shown on the top in panel (g) were used to drive the entirety of the POE inversion given that no carbon emissions are diagnosed after 7k years.

Thank you to the reviewer for raising these questions. Since we shifted the focus of the study from the PETM to the POE, we now provided uncertainty analyses by shifting the POE age records of the pH and $\delta^{13}\text{C}$ from the two sites (i.e., pH from Atlantic coastal site; $\delta^{13}\text{C}$ from the Kuzigongsu site) (Fig. S12). The impacts on the $\delta^{13}\text{C}$ of the source from the uncertainties of such correlation are assessed in our newly carried-out sensitivity analyses. We agree that each site is impacted differently by dissolution and mixing and by combining the records used in the inversion, but these uncertainties do not affect our main conclusion that the POE is associated with mainly

volcanic CO₂ sources. We also expand the discussions on how the “double-inversion” is done for the POE:

First, the “double-inversion” modeling takes the observed pH data, which constrains the flux and magnitude of CO₂ emissions, and the observed $\delta^{13}\text{C}$ values of the dissolved inorganic carbon of the surface ocean, which simultaneously determines the source of the emitted carbon by computing the $\delta^{13}\text{C}$ values of the carbon source. At each model time step, a pulse of CO₂ is emitted to the atmosphere at a given rate if the $\delta^{13}\text{C}$ value is lower than the previous time step, and the modeled surface DIC $\delta^{13}\text{C}$ values and the observed $\delta^{13}\text{C}$ values at the KZGS section are compared. If the current modeled surface DIC $\delta^{13}\text{C}$ value is higher than the data value, the $\delta^{13}\text{C}$ value of the emitted CO₂ is assigned a value of -100‰. In contrast, if the current modeled surface DIC $\delta^{13}\text{C}$ value is lower than the data value, the $\delta^{13}\text{C}$ value of the emitted CO₂ is assigned a value of 0‰. $\delta^{13}\text{C}$ values of the emitted CO₂ between -100‰ and 0‰ can be achieved by binning the emission fluxes in time and averaging flux-weighted $\delta^{13}\text{C}$ values. Justification for the choice of these end-member $\delta^{13}\text{C}$ values of the emitted CO₂ is provided in Gutjahr et al.⁸.

For our inversion experiments, the model was first spun up for 20 kyr to establish the basic ocean circulation and climatic state under published late Paleocene boundary conditions, including paleogeography and paleobathymetry^{84,85}. This is followed by an open-system spin-up of 200 kyr to allow the long-term $\delta^{13}\text{C}$ cycle to reach balance. A range of inversion experiments were carried out (Table S5a; Figs. S8-11). We tested combinations of: age model option 1 (slower sedimentation rate) versus age model option 2 (precessional cycle or sub-Milankovitch cycle for the 6-10 m cycles shown by spectral analysis with faster sedimentation rate) based on orbital cyclostratigraphy; uncertainty associated with surface ocean pH reconstruction (average) vs. upper and lower limits; ‘upper’ and ‘lower, respectively).

In these experiments, we assume that the POE onset occurred as a linear decline in both $\delta^{13}\text{C}$ and pH simultaneously (Table S5a). We varied the duration of this decline from 1,000 to 6,959 years. Once the minima in $\delta^{13}\text{C}$ and pH were reached, these values were held constant up until the end of the experiment. We use the same “double-inversion” methodology in both the main experiments and the sensitivity experiments, both starting from the same open-system spin-up state (Table S5a).

Sensitivity experiments and analysis:

We carried out a range of sensitivity experiments to explore the importance of the age correlation between the KZGS site and the mid-Atlantic Coastal Plain site to explore whether there is a strong impact of the uncertainty in correlating these records to the diagnosed total carbon emissions. To address this issue, we use records from two locations simultaneously point-by-point as inversion targets. We align the two POE records from the KZGS and the Maryland Coastal Plain of the Salisbury Embayment according to the onset of the CIE, and use the orbitally tuned age model to estimate the entire duration of the CIE. The impacts on the $\delta^{13}\text{C}$ of the source from the

uncertainties of such correlation are assessed in sensitivity analyses (Table S5a; Figs S8-11).

Figure 4.

References:

- Bice KL, Barron EJ, Peterson WH. Reconstruction of realistic early Eocene paleobathymetry and ocean GCM sensitivity to specified basin configuration. In: Tectonic boundary conditions for climate reconstructions (eds Crowley T, Burke K). New York, Oxford University Press (1998).
- Ridgwell, A. et al. Marine geochemical data assimilation in an efficient Earth system model of global biogeochemical cycling. *Biogeosciences* 4, 87–104 (2007).

Ridgwell, A. Glacial–Interglacial Perturbations in the Global Carbon Cycle. PhD thesis, Univ. East Anglia (2001).

Kirtland Turner S, Ridgwell A. Development of a novel empirical framework for interpreting geological carbon isotope excursions, with implications for the rate of carbon injection across the PETM. *Earth and Planetary Science Letters* 435, 1-13 (2016).

Overall, there is too much data and not enough rigorous interpretation in this manuscript. I'm left with many questions and without any clear takeaways. The title, abstract, and text of the manuscript are not well matched. I envision a study that could effectively use some (perhaps not all) of the presented data to drive model simulations of the PETM and POE and compare and contrast the results in terms of the diagnosed emissions, $\delta^{13}\text{C}$, impact on CO_2 , and differential recovery mechanisms. Or, I can envision a study specifically focused on climate instability (as in the title), though this seems to require better constraints on temperature specifically.

We took the reviewer's suggestion to drop the climate instability due to the low resolution of temperature records from the study site, and instead focus our modeling of the carbon emissions during the POE and compare and contrast the results in terms of the diagnosed emissions. The new discussions provide uncertainty assessment by using pH record from coastal Atlantic Ocean, uncertainties in age model, and uncertainties in the $\delta^{13}\text{C}$ values from a single site (Table S5, Figs 9-12).

REVIEWER COMMENTS

Reviewer #1 (Remarks to the Author):

The revised paper by Jiang and co-authors is a significant improvement on the first submission and is an excellent and exciting paper overall. The only area where I believe significant further investigation is required is the modelling results and the implications for a plausible source. After addressing this one moderate comment and a few minor revisions, it is worth publishing in Nature Communications. With best regards, Morgan Jones

Reply: We wish to express our gratitude for reviewer's constructive suggestions for improving the quality of the manuscript, and have made revisions according to reviewer's comments. Line-by-line reply to reviewer's comments is provided below. We also further tighten up the text such that it exhibits a natural flow of speech and grammatical structure.

Main comment:

Lines 266 to 272: I am a little worried about the validity of the modelling results that ascribe a purely NAIP volcanic carbon source for the POE, as 1 to 10 Pg C/yr is an enormous flux, and up to 12,300 Pg C is an enormous volume. To put these numbers into perspective, Stephen Jones et al. (2019)'s paper estimates that the total mantle-derived carbon reservoir for the entire NAIP is 14,100 Pg C, while our recent estimate puts it between 21,000 and 35,000 Pg C (M. Jones et al., 2023). These are estimates for the entire 62 to 54 Ma interval, and while much of this volume is believed to have erupted between 56 and 55.5 Ma, it would still need a phenomenal amount of magma/lava to degas in an extremely short amount of time to solely account for the POE or the PETM. Taking the approximation of 3.5 Tg C per km³ of magma as a potential degassing source (see Jones et al., 2016), a yearly flux of 1 to 10 Pg C/yr would require an eruptive / shallow intrusive flux of 286 to 2860 km³/yr of magma, which is absolutely colossal. The geochronology of the NAIP continental flood basalt sequences is not very well constrained, but the field evidence in localities such as East Greenland and the Faroe Islands shows individual lavas with evidence of hiatuses between that seem to contradict the hypothesis that large volumes of these lavas were erupted in a very short (e.g. 1–10 kyr) time periods. Therefore, the model results appear to generate unrealistic values for magmatic gas fluxes, and this discrepancy between the model findings and the NAIP magmatic carbon reservoir potential needs to be discussed in greater detail in the manuscript.

Reply: We now provide new discussions to compare the modeled carbon emission and field-based evidence of intrusive and extrusive activities. First, we provide a clear and succinct summary of the double-inversion experiments, with a focus on our preferred model scenario associated with a median ΔpH and age option 1, which

provides an upper limit for the estimated carbon emission flux (lines 272-304). We also show model results of additional analyses turning on organic carbon burial feedback during the recovery phase of $\delta^{13}\text{C}$ (lines 284-286; Methods and Table S5a). Greater details on the model findings and the NAIP magmatic carbon reservoir potential are now provided in lines 342-365, which justify the simulated carbon emission as an upper limit.

I don't think that the model needs to be rerun, but some discussion about the potential sources and differences between the model and field observations is warranted. Potential avenues include how the model setup may vary from natural conditions, and what NAIP sources may be active. Does the model cope well with an enhanced ^{12}C sink (e.g. organic matter burial) that could be masking the degassing signature in atmospheric CO_2 values, perhaps? The cGENIE model is not my expertise, but I think that the paper would be significantly enhanced with some discussion around what the limitations of the model are, what potential reasons can there be between the calculated fluxes from modelling and field observations, and how future studies could constrain these uncertainties.

Reply: Regarding an enhanced ^{12}C sink associated with organic carbon burial in cGENIE, we ran additional model simulations turning on organic carbon burial flux from ~ 8 kyr onward when $\delta^{13}\text{C}$ starts to recover, which allows for the removal or excess organic carbon from the surface ocean (see Table S5a for model results with varying organic carbon burial scaling factor). This was done when the modelled mean global surface ocean $\delta^{13}\text{C}$ value becomes lower than the observed value at the KZGS site, ^{13}C -depleted carbon is removed from the system to force the $\delta^{13}\text{C}$ to become more positive via negative emissions (e.g., Gutjahr et al., 2017). The new modeling result suggests ~ 800 to $1,900$ Pg C cumulative removal of light carbon at average $\delta^{13}\text{C}$ value of -30‰ and a rate of ~ -0.1 to -0.3 Pg C yr^{-1} , which lead to a better match between modelled and observed recovery of both $\delta^{13}\text{C}_{\text{DIC}}$ and pH. It is also important to note that a caveat of cGENIE in interpreting our results is the lack of terrestrial biosphere and potential changes in orbital forcing, which could impact the climate responses and lead to uncertainties in carbon emission estimates. This caveat is acknowledged in lines 358-361. The revised text reads as the following:

“Indeed, we find that applying a time-varying organic carbon burial calculated to nudge model $\delta^{13}\text{C}_{\text{DIC}}$ closer to the data during recovery (~ 8 kyr onwards), we achieve not only a better match between the modelled and observed $\delta^{13}\text{C}_{\text{DIC}}$ recovery but also to the simulated pH recovery (Fig. 4a). The diagnosed organic carbon burial rate is approximately -0.1 to -0.3 Pg C yr^{-1} (Fig. 4c) with an average modelled $\delta^{13}\text{C}$ value of -30‰ , and leading to a cumulative C_{org} burial over the span of POE recovery of ~ 800 to $1,900$ Pg C (Fig. 4d). It is also important to note that a caveat of cGENIE in interpreting our results is the lack of terrestrial biosphere and potential changes in orbital forcing, which could impact the climate responses and lead to uncertainties in carbon emission estimates.”

Discussions around the reconciliation between model and data discrepancy are now provided in lines 342-365. Furthermore, we add new discussions on how future studies could constrain the uncertainties on modeled fluxes:

“Although this study provides an upper constraint on the carbon emission flux during the POE, more precise $\delta^{11}\text{B}$ -based global surface pH records, detailed history of the Iceland plume pulsing^{12, 77}, sea surface temperature records from across different latitudes, and better-constrained geochronology of the NAIP activity are clearly needed to reduce the uncertainty of the estimated carbon emission fluxes.”

Minor comments:

The focus on the NAIP as a purely volcanic source is a little misleading, as numerous studies (e.g. Berndt et al., 2023) highlight the importance of sub-volcanic intrusions as a significant carbon flux across this key interval. The explosive ejection of thermogenic gases through these hydrothermal vent complexes would be a mix of magmatic and sedimentary sources. Stephen Jones et al. (2019) estimate that the partition would be would be 90% contact metamorphic and 10% mantle signature, and that peak emission fluxes of 0.2–0.5 Pg C/yr from thermogenic NAIP sources could have initiated the PETM. I personally think the 90:10 ratio is a little high, as they assume a thick metamorphic aureole in their calculations and do not account for magmatic degassing from deeper in the plumbing system. Therefore, the intrusive part of the NAIP can potentially be a substantial source of magmatic carbon, so I would err on the side of caution and recommend referring to it as “NAIP activity” or “emplacement of the NAIP” rather than explicitly “volcanism” throughout the text.

Reply: Thank you for this great suggestion. We now reword “volcanism” as “NAIP activity” throughout the text.

Line 124: Add ‘a’ to “and mercury content as a possible indicator of volcanism”

Reply: Done.

Line 128: Add ‘evidence of’ to “...represent evidence of an abrupt and significant warming...”

Reply: Done.

Lines 180–181: This explanation of Mn* would look better as an equation.

Reply: Done.

Line 192: Elements at the start of sentences should be written out in full (i.e. change

Hg to Mercury here)

Reply: Done.

Line 203: For reasons stated above, I think it would be better to change “volcanism” to “activity” here.

Reply: Done.

Line 216: As above. Either change to “The C/N ratio” or “Carbon/nitrogen ratio”

Reply: Done.

Lines 229–230: The locations of hydrothermal vent complexes here is wrong. The hydrothermal vent complexes are exclusively found on the continental margins around the NAIP (modern day Northeast Atlantic Ocean), caused by igneous intrusions into sedimentary sequences. See the recent paper by Berndt et al. (2023).

Reply: We followed the reviewer’s suggestions to correct the locations of the hydrothermal vent complexes and cited relevant papers, including Berndt et al. (2023) and Frieling et al. (2016).

Line 234: I would probably recommend not stating the estimated PETM onset age of 56.01 Ma (Zeebe and Lourens, 2019) here, as the U-Pb age of 55.785 ± 0.034 Ma from a bentonite layer that is within the PETM CIE body in Svalbard (Charles et al., 2011) does not match with the 127 kyr PETM duration presented in this paper. You can perhaps mention this discrepancy and the alternate 55.93 Ma PETM onset age (Westerhold et al., 2017) if you want to dig into what the cyclostratigraphy presented in this paper suggests for the validity of the various age models.

Reply: We followed the reviewer’s suggestion and removed the estimated PETM onset age of 56.01 Ma.

Line 554 (Figure 1): Does the “CIE” marked on panel (d) refer to the PETM? Please clarify

Reply: The CIE marked on panel (d) does refer to the PETM. We now clarify this issue in the revised figure 1d.

Lines 635–636 and 756–758: References numbered 12 and 51 are a repeat, the latter is the correct version.

Reply: this is now corrected.

References mentioned:

Berndt, C., Planke, S., Alvarez Zarikian, C. A., Frieling, J., Jones, M. T., Millett, J. M., Brinkhuis, H., Bünz, S., Svensen, H. H., Longman, J., Scherer, R. P., Karstens, J., Manton, B., Nelissen, M., Reed, B., Faleide, J. I., Huismans, R. S., Agarwal, A., Andrews, G. D. M., Betlem, P., Bhattacharya, J., Chatterjee, S., Christopoulou, M., Clementi, V. J., Ferré, E. C., Filina, I. Y., Guo, P., Harper, D. T., Lambart, S., Mohn, G., Nakaoka R., Tegner, C., Varela, N., Wang, M., Xu, W., and Yager, S. L.: Shallow-water hydrothermal venting linked to the Palaeocene–Eocene Thermal Maximum, *Nat. Geosci.*, <https://doi.org/10.1038/s41561-023-01246-8>, 2023.

Jones, S. M., Hoggett, M., Greene, S. E., and Dunkley Jones, T.: Large Igneous Province thermogenic greenhouse gas flux could have initiated Paleocene-Eocene Thermal Maximum climate change, *Nat. Commun.*, 10, 5547, <https://doi.org/10.1038/s41467-019-12957-1>, 2019.
(Currently reference No. 13 in this manuscript)

Jones, M. T., Jerram, D. A., Svensen, H. H., and Grove, C.: The effects of large igneous provinces on the global carbon and sulphur cycles, *Palaeogeogr. Palaeoclimatol.*, 441, 4–21, 2016.
(Ref. 64)

Jones, M.T., Stokke, E.W., Rooney, A.D., Frieling, J., Pogge von Strandmann, P.A.E., Wilson, D.J., Svensen, H.H., Planke, S., Adatte, T., Thibault, N., Vickers, M.L., Mather, T.A., Tegner, C., Zuchuat, V., Schultz, B.P.: Tracing North Atlantic volcanism and seaway connectivity across the Paleocene–Eocene Thermal Maximum (PETM). *Clim. Past* 19, 1623-1652, 2023.
(Ref. 55, updated from the preprint version now it is published)

Westerhold, T., Röhl, U., Frederichs, T., Agnini, C., Raffi, I., Zachos, J. C., and Wilkens, R. H.: Astronomical calibration of the Ypresian timescale: implications for seafloor spreading rates and the chaotic behavior of the solar system?, *Clim. Past*, 13, 1129–1152, <https://doi.org/10.5194/cp-13-1129-2017>, 2017.

Zeebe, R. E. and Lourens, L. J.: Solar System chaos and the Paleocene–Eocene boundary age constrained by geology and astronomy, *Science*, 365, 926–929, 2019.
(Ref. 63)

Reviewer #2 (Remarks to the Author):

I reviewed a previous version of this manuscript, and I agree with the authors' decision to focus on the diagnosis of the likely carbon source for the POE. The additional experiments conducted as sensitivity analysis are welcome, but unfortunately I have many more unanswered questions raised by this version of the manuscript.

Reply: We wish to express our gratitude for reviewer's constructive suggestions for improving the quality of the manuscript, and have made revisions according to reviewer's comments. A line-by-line reply to reviewer's comments is provided below. We also further tightened up the text such that it exhibits a natural flow of speech and grammatical structure.

Overall, I don't think that a clear and succinct summary of the double inversion experiments has been presented. I do not think that the authors have clearly demonstrated that volcanic emissions can explain the POE without generating a $\delta^{13}\text{C}$ excursion that should have been observable in deep sea records. It is very challenging to interpret all experimental results by trying to combine the description of each experiment provided in a supplemental table with multiple figures with lines labeled 'Experiment 1, 2, etc.' and captions that are not re-written for each figure. It was difficult to remember what I was meant to be evaluating from each plot contrasting Figure 4 with the figures in SI. Key sensitivity analyses seem to be 1) the significance of varying the age model 2) importance of different assumptions about the pH constraint, and 3) importance of the alignment of $\delta^{13}\text{C}$ and pH targets. I guess the goal is to summarize the importance of 1) and 2) in Figure 4 and leave 3) to the SI. Overall, it would help for each subplot in each figure to identify the plotted lines in a way that clearly explains what is being varied (not 'Experiment ID ='). Importantly, comparison of the significance of different age model assumptions is hindered a bit by the use of different axis limits for the right and left-side subplots showing carbon emissions. The reported diagnosed $\delta^{13}\text{C}$ of the source (-4 to -13 per mil) (Line 267-268) is not what it looks like from Figure 4 (which is also difficult to see because the axis extends to -60 per mil).

Reply: We follow the reviewer's suggestion to provide a clear and succinct summary of the double-inversion experiments, with a focus on our preferred model scenario associated with a median ΔpH and age option 1, which provides an upper limit for the estimated carbon emission flux. We also show model results of an additional analysis turning on organic carbon burial feedback during the recovery phase of $\delta^{13}\text{C}$. Detailed discussions can be found in lines 248-365.

In order to allow for better visualization, we simplify Fig. 4 such that it only shows the preferred scenario. The results of remaining sensitivity experiments are now moved to Supplementary Figures S10-15. For the reported diagnosed $\delta^{13}\text{C}$ of the

source, we now report both the flux-averaged value (-11%) and the range of variability (-3 to -23%). To better reflect the range of variation in the diagnosed $\delta^{13}\text{C}_{\text{source}}$ and the $\delta^{13}\text{C}$ value of the buried organic matter, we now revise the y-axis limit to extend to -35% . These changes are reflected in the revised Fig. 4 copied below.

Revised Figure 4.

Furthermore, we provide a more detailed summary on the importance of (1) the duration of the POE onset (6,959, 1,596, and 1,000 years based on age model option 1, age model option 2, and an idealized age model, respectively); (2) uncertainty in the alignment age of $\delta^{13}\text{C}$ and pH (e.g., age of the minimum pH is offset by ± 200 years); (3) uncertainty of the ΔpH (0.13, 0.14, 0.32, 0.42, and 0.64 based on the reconstruction of Babila et al.¹⁸); (4) uncertainty of the $\delta^{13}\text{C}$ record (e.g., $\delta^{13}\text{C}_{\text{DIC}}$ reconstructed based on $\delta^{13}\text{C}_{\text{carb}}$ vs. $\delta^{13}\text{C}_{\text{org}}$ from the study site); and (5) different rate and amount of organic carbon burial in driving the recovery of $\delta^{13}\text{C}$ and pH. These new discussions are replicated below:

“For instance, a larger magnitude of ΔpH and a shorter POE duration would yield a more ^{13}C -enriched diagnosed $\delta^{13}\text{C}_{\text{source}}$ values and higher emission fluxes for the POE (Table S5a and Fig. S9-S13). A smaller magnitude of ΔpH (0.1 to 0.3) would lead to a lower diagnosed average $\delta^{13}\text{C}_{\text{source}}$ values (-14.2‰ to -31.9‰) and smaller emission fluxes (0.4 to 1 Pg C yr^{-1} with total cumulative emissions up to $\sim 4,500 \text{ Pg C}$; Table S5a; Fig. S15), but the first 3.4 kyr of the POE is characterized by a volcanic CO_2 signature (average $\delta^{13}\text{C}_{\text{source}}$ value ranges from -12.3‰ to -6.1‰), followed by more ^{13}C -depleted emissions (average $\delta^{13}\text{C}_{\text{source}}$ value ranges from -50.4‰ to -20.9‰) (Fig. S15). Assuming different alignment scenarios of $\delta^{13}\text{C}_{\text{DIC}}$ and pH targets do not seem to affect the average diagnosed $\delta^{13}\text{C}_{\text{source}}$ value (-10.3 to -12.6‰ ; Table S5a). Despite the uncertainty in the magnitude of pH change in particular, relatively isotopically heavy CO_2 emissions and hence a NAIP related carbon source appears to be a robust feature in our analysis.”.

A significant concern is the misrepresentation of the benthic $\delta^{13}\text{C}$ excursion generated in the experiments. In the main text, it is reported that the POE double inversions yield ‘no obvious CIE’ in the modeled deep ocean (Line 275). I don’t agree with that interpretation based on what is shown in Figure S11. The benthic CIE in Fig S11a is roughly 1.6 per mil and roughly 1 per mil in Fig S11b. In panels c-f benthic $\delta^{13}\text{C}$ is still decreasing because the duration of the experiment is so short that the deep ocean has either barely seen or has not seen the full magnitude of the excursion in the surface. It is very hard to argue that one would not expect to see a 1 to 1.6 per mil excursion in the deep ocean as predicted by these experiments - how do you explain the abundance of ‘hyperthermals’ identified of approximately this magnitude across the Early Eocene? It is a really interesting question whether inversions of the POE CIE and pH are compatible with the supposed absence of this event from the deep sea record. Viewing, for instance, the high resolution benthic foram $\delta^{13}\text{C}$ record from ODP Site 1262 (Littler et al., 2014) there does not appear to be evidence for a -1 per mil excursion ~ 100 kyr prior to the PETM. Does this suggest something about the amount of time between the two events and hence the likelihood that the POE could be erased by burndown? It is not the duration alone of an event that limits its appearance in the deep sea - it is the total size of the event that matters. (If only duration mattered, that would erroneously suggest that the impact of modern carbon emissions wouldn’t be recorded in the geologic record). Some of the inversions presented here show a really size-able event - CO_2 more than doubling and masses of carbon predicted that are similar to some previously published estimates for the PETM itself - how are these ‘relatively small’ emissions estimates?

Reply: The smaller magnitude of deep-sea carbonate POE is demonstrated in our revised Figure 4 (Fig. 4a; shown below) with only about -1.5‰ overall decrease over the entire modelled interval. The lack of organic carbon burial in our initial experiments led to the inability of $\delta^{13}\text{C}_{\text{DIC}}$ and $\delta^{13}\text{C}_{\text{carbonate}}$ to recover to their pre-perturbation values. To address this issue (and relevant issue raised in the comment below), we ran an additional experiment with organic carbon burial ($\sim -0.1 \text{ Pg C yr}^{-1}$

similar to Gutjahr et al. 2017), and show that $\delta^{13}\text{C}_{\text{DIC}}$ exhibits a recovery that closely follows the observation after 8.4 Kyr. Increasing the amount of organic carbon burial led to slightly better match between the modeled and observed recovery of $\delta^{13}\text{C}_{\text{DIC}}$ and pH data (Table S5a and Figure S15). We also provide a brief discussion on the role of bioturbation and sediment mixing on the smaller POE magnitude in core-top carbonates during the POE, and replicate the text below:

“The evolution of mean core-top carbonate (CaCO_3) with time in the model exhibits a delayed $\delta^{13}\text{C}$ response to CO_2 emissions compared to surface DIC, and is characterized by a smaller magnitude of $\delta^{13}\text{C}$ decrease ($\sim -1.5\text{‰}$) (Fig. 4a). This is due to combined effects of bioturbation and dissolution as a result of the cumulative carbon emission (Fig. 4d), as demonstrated in a comparable Eocene hyperthermal event³.”

Since we now focus our model results on our preferred scenario (median $\Delta\text{pH} = -0.4$ and age option 1), we remove the statement about ‘relatively small’ emissions, and state the emission history associated with this scenario and stress on the significance of our sensitivity experiments.

Reference:

Littler, K., Röhl, U., Westerhold, T. and Zachos, J.C., 2014. A high-resolution benthic stable-isotope record for the South Atlantic: Implications for orbital-scale changes in Late Paleocene–Early Eocene climate and carbon cycling. *Earth and Planetary Science Letters*, 401, pp.18-30.

There are additional aspects of the analysis that are confusing to me (itemized by line below). The figures that present the age model development and the model results are difficult to read and have very small text. It is strange to leave the cyclostratigraphy to SI given the centrality of age model constraints to the interpretation of the POE via Earth system modeling. Text is duplicated in both Methods and SI but sometimes with slight differences. The SI should not repeat the Methods but provide additional explanation/discussion (for instance, an analysis of the sensitivity experiments).

Reply: We apologize for the small text in our model results figure and now revise them accordingly for better visualization (see updated Fig. 4). We also remove duplicated text in SI, and provide additional explanation and discussion on the sensitivity experiments of the age model in SI. A brief introduction of the cyclostratigraphy method and main results is now included in the main text to provide a better context of the age model.

Moreover, the Methods (particularly regarding the cyclostratigraphy and modeling) are incomplete. Cyclostratigraphy uses magnetic susceptibility, but MS measurements are never described. What is the resolution of MS measurements? Exactly how is the record interpolated, detrended, smoothed or filtered? ‘Dynamical filter to isolate

precession cycles' needs elaboration - where is the evidence that there is significant precession power in the timeseries prior to filtering the record at this frequency and hence adding power? How is a 'reasonable range of sedimentation rate' set? What are the overall constraints? Multiple software packages are described (MATLAB scripts, AnalySeries) but then the final sentence states that all analyses are conducted with Astrochron unless otherwise noted. My overall impression is that this description of cyclostratigraphic methods doesn't give the reader the ability to reproduce the results, which should be the goal.

Reply: More details on MS data measurements (including number of samples measured) and how data are interpolated, detrended, smoothed or filtered are provided in Methods. Details on sensitivity analysis for cyclostratigraphy (Age model 2) is provided in the SI. Software packages are clarified in Methods.

Based on multi-taper method (MTM) spectral analysis, we find strong signals of both eccentricity and precession (Fig. 2f). However, we note that the power of long-term cycles (i.e., short eccentricity cycles) may have muted the manifestation of precession cycles in the evolutive harmonic analysis (EHA) (Fig. 2f). Therefore, we remove the > 4 m cycles that may be associated with eccentricity cycles to reveal precession-related cycles as the most prominent signal in the EHA spectrogram (Fig. S6). The significant power of the interpreted precession cycles in the EHA spectrogram allows us to effectively isolate this signal from EHA (Fig. S5). We then use the precession cycles to construct an astrochronological timescale for the study interval. The new text is shown below:

“Since the power of long-term cycles (i.e., short eccentricity cycles) may have muted the manifestation of precession cycles in the evolutive harmonic analysis (EHA), we remove the > 4 m cycles that may be associated with eccentricity cycles to reveal precession-related cycles as the most prominent signal in the EHA spectrogram (Fig. S7). The significant power of the interpreted precession cycles in the EHA spectrogram allows us to effectively isolate this signal from EHA (Fig. S5). We then use the precession cycles to construct an astrochronological timescale for the study interval. Analyses of TimeOpt and COCO indicate alternation of optimal sedimentation rates (i.e., 6.0 cm kyr^{-1} and 8.3 cm kyr^{-1}) (Fig. S5). Spectral Moments, eTimeOpt and eCOCO together suggest the estimated sedimentation rate ranges from 4.2 to 10.6 cm kyr^{-1} with increased sedimentation rate during the PETM body (Figs. S5, S6).”

The revised text regarding justification of a range of feasible sedimentation rates is copied below:

“A range of feasible sedimentation rates is required for TimeOpt and COCO analyses. The duration of the CIE during PETM was estimated to be 100 – 200 kyr based on astrochronology^{26, 29, 30, 31} and 3He age models^{27, 32}. Using a CIE duration of 100 – 200 kyr and a PETM thickness of 10.6 m at the study site, the calculated sedimentation

rate ranges from 5.3 to 10.6 cm kyr⁻¹, therefore, 4 to 12 cm kyr⁻¹ was set as the range of reasonable sedimentation rates for TimeOpt and COCO analyses (Fig. S6).”

Figure S7. (a) Detrended MS values to avoid possible/potential influence of eccentricity cycles. (b) EHA spectrogram of the detrended MS series. Sliding window is 8 m with a step of 0.1 m.

Regarding the description of Earth system modeling, the Methods first states that the isotopic inversion target is CO₂ d13C (Line 503) and then later that the target is surface ocean DIC d13C (Line 511). What was the d13C record from the KZGS adjusted to in absolute values - CO₂ d13C or DIC 13C? Line 496 is duplicated in Line 523 (details of the SPINUP) with configuration variously described as late Paleocene and early Eocene. In the SI (Line 315) there is mention of different pH scenarios for both the PETM and POE inversions but these appear to have been handled differently. I think the different pH scenarios for the PETM double inversion actually have adjusted pH targets in terms of absolute value whereas the different pH scenarios for the POE adjust only the magnitude of the pH change but this is really unclear. For the PETM double inversion, what do you mean by pH is ‘set to’? From Fig S8e, there is a difference in initial CO₂ across the three different pH values (suggesting the absolute

pH was offset initially in each inversion from the SPINUP). Why were these different approaches taken for the PETM and POE double inversions? (In Figure 4, the different pH options for the POE inversions don't impact the initial value of CO₂, but the data shown in grey versus red dots do have differences in the absolute values of pH in addition to the magnitude of the change in pH).

Reply: We apologize for the typo and it is now fixed. The reviewer is correct that the isotopic inversion target is $\delta^{13}\text{C}_{\text{DIC}}$, and we assume that the difference between $\delta^{13}\text{C}_{\text{carb}}$ and $\delta^{13}\text{C}_{\text{DIC}}$ is 0.8‰ such that the initial $\delta^{13}\text{C}_{\text{DIC}}$ value is 2.7‰, same as the model $\delta^{13}\text{C}_{\text{DIC}}$ value at the end of the open-system spin-up.

We also change the description of the model configuration to late Paleocene-early Eocene to be consistent with the literature that refers the model configuration as both late Paleocene and early Eocene (e.g., Bice et al., 1998; Cui et al., 2011; Kirtland Turner & Ridgwell, 2016; Gutjahr et al., 2017).

We clarify the treatment of the changes in pH values for the PETM and the POE in Methods. The new text now reads “Although uncertainty exists for pre-PETM $\delta^{11}\text{B}$, the surface ocean pH at the end of the open-system spinup is 7.75, same as those used in Gutjahr et al. (2017)¹⁰, which is adapted as the initial surface ocean pH forcing in the “double inversion” experiment”.

Additional comments below:

Line 92 - provide the breakdown of CIE size across source for the POE as done above for the PETM?

Reply: Done.

Line 95 - in the main text, could briefly summarize what the cycle assumption is for each of the two tuned age models?

Reply: A brief summary on the cycle assumption used for each of the two tuned age models is now provided in the main text (lines 107-127; lines 475-510) and Supplementary Information.

Line 144 - conceivably there could also be a significant decline in carbonate production (rather than reduced preservation alone)?

Reply: We revised the text to reflect the role of reduced carbonate production in decrease in wt.% CaCO₃. The new text reads as the following:
“...may be attributed to significant reduction of carbonate production and/or detrital dilution, or shallow ocean acidification.”

Line 151 v Line 162 - duplicate description of elevated C₂₉ hopane ratios during the PETM

Reply: The duplicated text is now removed from line 162.

Line 174 - should be 'perturbing'

Reply: Fixed.

Line 172 - 177 - the wording of this sentence is unclear; why specifically related 2-MeHop to nitrogen cycle perturbation in this sentence when the previous sentence referred to a host of interpretations of which changes in N fixation was one option?

Reply: We revise this sentence to add the latest understanding of the trigger for elevated C₂₉ 2-MeHop Index value due to nitrogen cycle perturbation during ancient hyperthermals (e.g., Naafs et al.¹¹). The revised sentence now reads:

“Furthermore, the anomalously high C₂₉ 2-MeHop Index during the PETM may be attributed to marine nitrogen cycle perturbation as a result of biogeochemical changes, similar as seen during other major carbon cycle perturbation events of the Phanerozoic, such as the end-Permian mass extinction event⁴⁰, the end-Triassic extinction event⁴¹, and the Mesozoic Oceanic Anoxic Events^{42, 43}.”

Line 188 - 'global extent' is misleading; anoxia is documented in relatively restricted basins

Reply: We removed “global extent” and used “widespread” instead.

Line 195 - if there were a direct link between Hg source and ¹³C-depleted source, why would there be a lag of more than 10 kyr?

Reply: The emission of ¹³C-depleted source did not reach its peak value immediately at the POE onset as revealed in Fig. 4, it takes a few kyr to reach its peak value, and the emission flux remains positive for about 8 kyr. This could explain the lag between Hg enrichment and the emission of ¹³C-depleted source(s). We also tone down this sentence to avoid the use of “direct link”. The revised sentence is shown below:

“Our site exhibits two prominent Hg/TOC peaks that show a small lead in time relative to the onset of the POE (~11 kyr) and the PETM (~26 kyr) (Fig. 2), supporting a pulsed Hg input and a possible link between Hg source and the ¹³C-depleted carbon source.”

Line 211-213 - is this meant to argue that wildfires in general cannot explain the Hg record or wildfires in the Arctic in particular?

Reply: We clarify this issue by stating that wildfire evidence from the Arctic region, northeastern US margin, and England appear to be associated with Hg flux much smaller than what the NAIP could afford. The revised text now reads as the following: “Moreover, Hg fluxes associated with wildfire (e.g., Arctic region¹⁵, northeastern US margin¹⁶, and England¹⁷) may have been far less than the Hg fluxes associated with a large igneous province event¹⁸, and therefore cannot provide sufficient Hg into the study site.”

Line 305 - didn't the double inversion provide constraints on the necessary magnitude of organic carbon burial during the POE? What is the rate and $\delta^{13}\text{C}$ of carbon removal in the simulated POE recovery?

Reply: Additional sensitivity experiment with organic carbon burial turned on is conducted to address this question (varying scaling factor from 0.01 to 0.05) (Table S5a; Fig. S15). For the experiments, the modeled rate and $\delta^{13}\text{C}$ of carbon removal in the simulated POE recovery is -0.1 to -0.3 Pg C yr⁻¹ with an average $\delta^{13}\text{C}$ value of $\sim -30\%$ for a cumulative amount of organic carbon burial of ~ 800 to 1,900 Pg C.

Line 309 - what does it mean for the system to ‘transition into the PETM’? If the argument is that the PETM was driven also by NAIP emissions, then does it matter whether or not organic carbon burial accelerated the recovery from the emissions that drove the POE? A more interesting question is to what extent CO₂ has recovered after POE emissions but before the PETM and whether that requires additional drawdown fluxes.

Reply: Our new modeling experiment suggests that ~ 800 to 1,900 Pg C organic carbon burial following the POE may have allowed for the system to recover gradually from the POE, but the $p\text{CO}_2$ did not completely return to the pre-POE level during the 21 kyr model simulation. Silicate weathering would have taken longer to drive the system to pre-PETM condition, but the main point here is that the POE has destabilized the Earth system long before the PETM and facilitated the transition from POE to the PETM. To avoid confusion, we now revise this sentence to provide a clearer summary of our finding in lines

Methods - location of analyses is not mentioned for n-alkanes only

Reply: The *n*-alkane analyses are performed at the University of Bristol and new text is provided in Methods.

Figure 2 - how is the choice made where to identify the PETM onset? In the

supplement, the shading to indicate the PETM covers a different interval, with the onset aligned to the initial decline in $\delta^{13}\text{C}$ of CaCO_3 rather than $\delta^{13}\text{C}$ of Corg. The overall description of the wt% CaCO_3 record in the text doesn't reflect the figure. The major decline in CaCO_3 doesn't appear to have anything to do with the timing of the POE or the PETM CIE, though an increase in CaCO_3 is aligned with the end of the PETM.

Reply: The choice for where to identify the PETM onset is based on the most rapid decline in $\delta^{13}\text{C}$ (depth = 18.9 m), not the decline of wt.% CaCO_3 .

Reviewers' comments:

Reviewer #1 (Remarks to the Author):

This is my third review of Jiang et al. for submission to Nature Communications. The manuscript is an improvement on the previous iteration and details an exciting new POE and PETM locality. I want to be supportive of this paper because of the high-resolution nature of the Kuzigongsu outcrop and the coverage of the POE. Moreover, I appreciate the efforts that the authors have put in to advance the paper and address my previous comments and suggestions. Sadly, I still find that the modelling part of this paper is its Achilles heel and not publishable in its current state. I am not a modeller by trade, so I cannot offer advice on where the issue lies, only that the output number of 230 km³ magma /year from the NAIP for 8000 years defies all our current understanding of LIP emplacements, mantle convection, and climate feedbacks. This magma flux estimate is probably around two orders of magnitude too high for the late Thanetian. I will detail a few of the reasons why here:

1) Timing. If the POE is ~144 kyr before the PETM onset, then it is around 56.07 Ma (using Westerhold et al), 56.15 Ma using Zeebe and Lourens (2019). This predates much of the known NAIP lavas, even within uncertainty. In East Greenland, we have the precise age of the Skaergaard intrusion at 56.02 Ma (Wotzlaw et al., 2012), which is the anchor for the emplacement age of the flood basalts that buried the magma chamber as it crystallised (Larsen and Tegner, 2006). Skaergaard was intruded into the lower basalt sequence, but almost all the subsequent flood basalt activity postdates the intrusion. A similar picture emerges on the Norwegian margin. The lower series encountered on IODP 642 (see http://www-odp.tamu.edu/publications/104_SR/VOLUME/CHAPTERS/sr104_51.pdf) likely contains the PETM (IODP 396 scientists are working on this right now), but that means that the main volcanism is either syn- or post-PETM in age. The geochronology of the Faroes is less well constrained, but it is probably similar to these other localities. There are a few lava flows that might possibly be the right sort of age to coincide with the POE (see site U1566 for example; Planke et al., 2023), but these are volumetrically a very small component of the NAIP. It is certainly nowhere near "19% to 38% of the total NAIP volume" as stated in line 356. Even 0.1% of the total NAP volume in 8000 years would be an extreme scenario. Which brings us on to:

Reply: Thank you for the suggestion regarding the timing of the POE and its relationship to the PETM.

In the current manuscript, we present two age model options, in which our age model option 1 suggests that the POE starts ~144 kyr before the PETM, and age model option 2 suggests that the POE starts ~59 kyr before the PETM. The 56.07 Ma POE age mentioned by the reviewer is based on Westerhold et al. (2007) age model option 2 (PETM age is 55.93 Ma). For our age model option 1, if we use the Westerhold et al. 2007 age model option 1 (PETM age is 55.53

Ma), the POE age is then 55.674 Ma which postdates the age of the Skaergaard intrusion. For our second age model, the POE starts \sim 55.988 (Westerhold et al. 2007 age option 2), and 55.588 ma (Westerhold et al. 2007 age option 1), both ages postdate the age of the Skaergaard intrusion at 56.02 Ma. Accounting for these uncertainties, the possible age of the POE overlaps with the age of Skaergaard intrusion in Greenland.

Considering both reviewers' comments, we take the model results that invert the minimum pH records (Δ pH \approx 0.1) in Babila et al. (2022) and a global compilation of $\delta^{13}\text{C}_{\text{carb}}$ records from four globally distributed sites (e.g., South Dover Bridge, Forada in Italy, Wyoming Bighorn Basin, and our study site in the eastern Tethys) as our main scenario to discuss the evolution of $\delta^{13}\text{C}_{\text{source}}$ and associated emission flux. The South Dover Bridge site contains $\delta^{13}\text{C}_{\text{carb}}$ measurements bulk carbonate, planktonic foraminifera (*Subbotina*), and benthic foraminifera (*Cibicidoides alleni*). All the other sites are associated with measurements on bulk carbonates and/or bulk organic matter or long-chain alkanes. But we choose to use carbonates, including foraminifera from all the sites for consistency. The global compiled $\delta^{13}\text{C}_{\text{carb}}$ results are shown in the revised Fig. S8. We also placed these $\delta^{13}\text{C}_{\text{carb}}$ records on four different age models that consider the wide range of varying published chronology of the POE from the literature and this study.

Figure S8. Paired “pH– $\delta^{13}\text{C}$ ” forcings used in our “double inversion” experiments using four independent age models. The $\delta^{13}\text{C}$ data are from a global compilation (blue circles are from the eastern Tethys (this study), green circles are from benthic foraminifera *Subbotina* at South Dover Bridge³⁵, red circles are from benthic foraminifera *Cibicidoids allenii* at South Dover Bridge³⁵, black cross represents bulk carbonate from South Dover Bridge³⁵, yellow upsidedown triangles are from pedogenic carbonate from Wyoming Bighorn Basin³⁶, and blue triangles are bulk carbonate from Forada, Italy³⁷). The pH data are from benthic foraminifera *Cibicidoids allenii* at South Dover Bridge³⁵. The magnitude of pH change used here is a conservative estimate of ~ 0.1 unit decrease from the onset of the POE.

The POE onset duration is defined by the initiation of $\delta^{13}\text{C}_{\text{carb}}$ decline to its minimum value and the alignment between all the records. The uncertainty associated with such temporal alignment can introduce potential errors in our estimates of the carbon sources, therefore, four age models were used to assess the degree to which these uncertainties can impact the source $\delta^{13}\text{C}$ values (see sensitivity tests). Here, we summarize the main model results based on our simulations using four varying age models, and list all our model results in the supplementary information. We note that the $\delta^{13}\text{C}_{\text{source}}$ varies from -19 to -75‰ , with a flux-weighted $\delta^{13}\text{C}$ value of -44.5‰ to -30.8‰ during the

entire emission duration. This signifies of thermogenic CO₂ emission (assuming thermogenic $\delta^{13}\text{C}_{\text{thermo}}$ value is -30‰ to -65‰, following Frieling et al., 2016 and Schoell, 1980). We also note a decrease in $\delta^{13}\text{C}_{\text{source}}$ values as time progressed toward the peak POE, suggesting that the declining $\delta^{13}\text{C}_{\text{source}}$ with time is consistent with the predicted $\delta^{13}\text{C}_{\text{source}}$ composition vs. time relationship based on coupled thermal-kinetic reaction model by Jones et al. (2019), i.e., as reaction temperature decreases, and thermal maturity declines, the $\delta^{13}\text{C}_{\text{source}}$ values become progressively lower.

Our new results suggest that mantle CO₂ source may have only played a minor role in driving the POE. During magmatic carbon emissions, mantle-derived CO₂ ($\delta^{13}\text{C} = -7\text{‰}$) can dissolve in the magma, exsolve before solidification and release via hydrothermal vents together with thermogenic methane (e.g., $\delta^{13}\text{C} = -30\text{‰}$; Jones et al., 2019). However, the rate at which mantle CO₂ is released is considered much slower compared to thermogenic methane (Jones et al., 2019). Bourdon and Sims (2003) and Peate and Hawkesworth (2005) proposed that the timescale at which basaltic magma travel upward to the surface is similar to or shorter than the PETM onset (~multi-millennial timescale). Out of the 11,000 to 18,000 sill-vent systems within the NAIP (Svensen et al., 2004), the average duration for the starting phase of the NAIP of 1-3 Myr (White and McKenzie, 1989; Eldholm and Grue, 1994), and average eruption rates ranging from 0.6 to 2.4 km³ yr⁻¹ (Eldholm and Grue, 1994). Jones et al. (2019) found that more than 90% of the carbon emitted from the sill province is associated with thermogenic methane, with less than 10% from mantle-derived CO₂ during the PETM. Our results are consistent with these studies.

Figure 4. Data assimilation results from our cGENIE Earth system modeling based on the pH- $\delta^{13}\text{C}_{\text{DIC}}$ double inversion of four scenarios based on different assumptions of POE onset duration. **(a-d)**, $\delta^{13}\text{C}_{\text{source}}$ values of the diagnosed carbon source for the four age models (see age model interpretation in the main text). **(e-h)**, Model-diagnosed rates of CO_2 emission for the four age models. **(i-l)**, Cumulative amount of CO_2 emitted for the four age models. The gray shaded area represents 1,500 years.

2) Volume. $230 \text{ km}^3(\text{magma})/\text{year}$ is an insanely high flux for a single eruption, let alone a flux from a LIP that continued for 8000 years. That is the equivalent of a Yellowstone sized-eruption's worth of material emplaced in a decade, 800 times in a row. For comparison, the 1783-84 eruption Laki that is one of the largest known basaltic eruptions was 14 km^3 volume. It is also very difficult to envision a mantle convection model that can emplace such a large volume in such a short space of time, go into almost a complete period of quiescence, then restart at similar flux levels for the PETM. Therefore, these estimated fluxes are well beyond the realms of possibility.

Reply: Considering the feasibility of these large fluxes, we shifted the focus of our main modelling results with a set of new simulations using a conservative ΔpH of 0.1 and a global carbon isotope data compilation (see Discussion in the revised manuscript). This is because the ΔpH value (0.4) used in our previous modelling led to an impossible carbon emission flux, suggesting this ΔpH may be an overestimate (see similar comments by reviewer #2: "The authors modeling also imposes a pH change greater than that reconstructed across the

PETM and consequently results in a CO₂ increase that is larger than independent estimates for the PETM CO₂ change (not those from boron isotopes, see Tierney et al., 2022).”). Our revised modelling that considers a large number of globally distributed sites and various age models suggest that the majority of ¹³C-depleted carbon emitted is associated with thermogenic source. Field observations, seismic imaging and thermal-kinetic reaction model suggest that the thickness of thermal contact aureole is greater than the thickness of the intruded sill, capable of generating large quantities of methane rapidly (Jones et al., 2019). Therefore, our estimated carbon emission flux (the $\delta^{13}\text{C}_{\text{source}}$ varies from -19 to -75‰, with a flux-weighted $\delta^{13}\text{C}$ value of -44.5‰ to -30.8‰) during the POE is consistent with the current understanding of the geological structures associated with the NAIP that control the carbon emissions.

3) Contemporaneous effects. Volcanoes don't just emit carbon, and some co-emitted volatiles would have a significant climate impact. Emplacing $1.8 \times 10^6 \text{ km}^3$ magma in 8000 years would release considerable volumes of such volatiles. Take sulfur for instance, using the 2014-15 eruption of Holuhraun in Iceland as an analogue for some quick back of the envelope calculations to illustrate this point. The Holuhraun eruption was 1.6 km^3 magma volume, with estimated emissions of 6.7 Tg SO₂ (Carboni et al., 2019). This is roughly 3.35 Tg S, and 2.09 Tg S / km³ magma. Scaling this up to the proposed flux (230 km³/yr) gives a sulfur flux of 482 Tg S/yr. Comparing this to pre-industrial cycle would increase the total atmospheric S flux by more than an order of magnitude (sea spray = 13-36 Mt S/yr; volcanoes = 6.5 – 10.5 Mt S/yr; 1Mt = 1 Tg), and nearly a 50x increase on volcanic fluxes (see Jones et al., 2016 for discussion). Experiencing these conditions for 8000 years would have catastrophic effects on terrestrial and freshwater ecosystems in particular, through a sustained volcanic winter and increase in acid rain. To reuse the Laki example, that eruption caused crop failures in the Americas and Europe and mass famine in Iceland from one year of volcanic winter. There just isn't the evidence in the geological record for this type of extinction-level disturbance during the POE, nor really during the PETM.

Reply: We agree that there is no geological evidence supporting extinction-level disturbance during the POE, and our new model results support this conclusion. Given that most of the carbon emitted in our revised modelling suggests thermogenic source, rather than volcanic CO₂, the contemporaneous SO₂ effect would be considered small.

This leaves us with the conclusion that the model presented cannot be representative of the POE. This could be due to incorrect input parameters, issues with the simplified nature of the model itself, unidentified sources/sinks, or some combination thereof. This also calls into question the assertion that the PETM is largely driven by volcanic CO₂, if Gutjahr et al. (2017) used similar model parameters as presented here. The main issue is that the only NAIP emissions that are likely to be capable of producing such large fluxes on sub-

millennial timescales are thermogenic sources (cf. S. Jones et al., 2019). The only way that can be reconciled with the presented model is if the volatile flux from hydrothermal vents was magma-dominated over contact metamorphic-dominated and that many of the vents were active ~ 100 kyr before the first known vent activity. Even then, the fluxes required are colossal, which suggests we are missing other potential sources or processes that are not accounted for in the model set up. Unfortunately I don't know what the best way forward is, as I do not have the expertise to suggest where the model is falling short. However, unless the model can produce a POE that is grounded in realistic NAIP fluxes, I don't think it should be included in this paper.

Reply: We follow the reviewer's suggestion to model the POE grounded in realistic NAIP fluxes (acknowledging that the ΔpH used in our prior simulations was too large). Our revised model results support thermogenic sources on sub-millennial timescales based on four independent age models.

With best regards, Morgan Jones

Minor comments:

Line 90 (and 119, ++): I don't see the benefit of shortening Kuzigongsu to KZGS, it doesn't significantly shorten the text and adds a barrier to readability. I suggest just using Kuzigongsu to describe the locality.

Reply: Thank you for this suggestion. We now remove KZGS throughout the text and use Kuzigongsu instead.

Line 142: Is the ' \geq ' symbol correct? If the water depth is from around 30 m up to 50 m, then the symbol should be ' \leq ' instead.

Reply: Thanks for catching this. Since the water depth is up to 50 m, we removed the ' \geq ' symbol to reflect this.

Lines 158-162: This is a long sentence. I suggest you break in two to improve readability (I would break it into "...same depth. However, the thermal maturity...")

Reply: We now break this sentence into two following the reviewer's suggestion.

Line 189: Suggest splitting this sentence too and fixing a slight grammar typo in doing so: "...as a result of biogeochemical changes. This is similar to observations of other major carbon cycle perturbations..."

Reply: We now split this sentence in two and the new sentence reads as the following:

“This is similar to observations of other major carbon cycle perturbations of the Phanerozoic, such as the end-Permian mass extinction event⁴², the end-Triassic extinction event⁴³, and the Mesozoic Oceanic Anoxic Events^{44, 45}.”

Line 244: Change “receive” to “received”.

Reply: “receive” is now changed to “received”.

References mentioned:

Carboni, E., Mather, T.A., Schmidt, A., Grainger, R.G., Pfeffer, M.A., Ialongo, I., Theys, N. (2019). Satellite-derived sulfur dioxide (SO₂) emissions from the 2014–2015 Holuhraun eruption (Iceland). *Atmospheric Chemistry and Physics* 19 (7), 4851–4862.

Gutjahr, M., Ridgwell, A., Sexton, P. F., Anagnostou, E., Pearson, P. N., Pälike, H., Norris, R. D., Thomas, E., and Foster, G. L. (2017). Very large release of mostly volcanic carbon during the Palaeocene–Eocene Thermal Maximum, *Nature*, 548, 573–577.

Jones, S.M., Hoggett, M., Greene, S.E., and Dunkley Jones, T. (2019) Large Igneous Province thermogenic greenhouse gas flux could have initiated Paleocene-Eocene Thermal Maximum climate change, *Nat. Commun.*, 10, 5547, <https://doi.org/10.1038/s41467-019-12957-1>.

Larsen, R. B. and Tegner, C. (2006). Pressure conditions for the solidification of the Skaergaard intrusion: Eruption of East Greenland flood basalts in less than 300,000 years, *Lithos*, 92, 181–197.

Planke, S., Berndt, C., Alvarez Zarikian, C.A., Agarwal, A., Andrews, G.D.M., Betlem, P., Bhattacharya, J., Brinkhuis, H., Chatterjee, S., Christopoulou, M., Clementi, V.J., Ferré, E.C., Filina, I.Y., Frieling, J., Guo, P., Harper, D.T., Jones, M.T., Lambart, S., Longman, J., Millett, J.M., Mohn, G., Nakaoka, R., Scherer, R.P., Tegner, C., Varela, N., Wang, M., Xu, W., and Yager, S.L. (2023). Site U1566. In Planke, S., Berndt, C., Alvarez Zarikian, C.A., and the Expedition 396 Scientists, *Mid-Norwegian Margin Magmatism and Paleoclimate Implications. Proceedings of the International Ocean Discovery Program, 396: College Station, TX (International Ocean Discovery Program)*. <https://doi.org/10.14379/iodp.proc.396.104.2023>.

Wotzlaw, J., Bindeman, I., Schaltegger, U., Brooks, C., and Naslund, H. (2012). High-resolution insights into episodes of crystallization, hydrothermal alteration and remelting in the Skaergaard intrusive complex, *Earth Planet. Sc. Lett.*, 355, 199–212.

Reviewer #2 (Remarks to the Author):

I reviewed a previous version of this manuscript and I appreciate the authors responses to my comments, particularly the addition of more information about the age model construction. New high-resolution PETM records, including expression of the POE, are a welcome contribution, and additional age

information about the duration of the POE and between the POE and PETM is especially significant. The problem is that I am ultimately not convinced that the way the double inversion modeling approach is applied and interpreted, and the central conclusions of the manuscript about carbon forcing that result, are valid for the POE. Central to the concept of the double inversion, as applied by Gutjahr et al., 2017, across the PETM, is the understanding that the PETM $\delta^{13}\text{C}$ excursion represents a whole ocean perturbation driven by carbon addition from a source external to the ocean-atmosphere system and that the record of $\delta^{11}\text{B}$ change across the PETM in the North Atlantic, due to its consistency with the existing Pacific record, is a good estimate of the magnitude of global surface ocean acidification. Many PETM records collected over decades substantiate the argument about the whole ocean $\delta^{13}\text{C}$ perturbation, and model-data comparison of simulated pH changes at the locations corresponding to the $\delta^{11}\text{B}$ records substantiate the application of those pH changes as global constraints. I don't think that the evidence for the POE reaches the same bar, so I think too much weight is being given to interpreting the results from an experimental framework that I'm not convinced is valid. The POE has not been conclusively identified in deep-sea records, and despite the authors comparison to features in the Site 1209 and Site 1263 benthic foraminiferal records of the late Paleocene (not shown and no correlation attempted), it's not even clear that the POE is identifiable in the deep sea. This was the basis of my previous major criticism of this paper, and I don't think the revision has adequately addressed this problem. A modeled $<1\text{‰}$ $\delta^{13}\text{C}$ excursion is significant, and a -1.5‰ excursion doesn't compare well to a -0.4‰ benthic excursion, with no quantification of how bioturbation should impact the modeled core-top $\delta^{13}\text{C}$ excursion magnitude (moreover, the issue of experiment duration in recording the core-top excursion magnitude has not been addressed). The authors modeling also imposes a pH change greater than that reconstructed across the PETM and consequently results in a CO_2 increase that is larger than independent estimates for the PETM CO_2 change (not those from boron isotopes, see Tierney et al., 2022).

Reply: Thank you for these great suggestions. We agree that the POE is less well studied compared to the PETM, but it is important to better understand the precursory event that led up to the PETM. In doing so, we now provide revised cGENIE modelling to address the reviewer's concern regarding the validity of the experimental framework. The POE is not recorded in the deep sea for several reasons:

- 1) The rapidity of the POE. The onset of the POE was originally suggested to be less than 1.5 kyr, the total duration less than 2 to 5.5 kyr, and the duration between the onset of the POE and the onset of the PETM is ~ 37.5 kyr by Bowen et al. (2015). In contrast, Babila et al. (2022) suggested that the POD could have been shorter than 1 kyr, possibly century to millennia. Therefore, our new modelling accounts for the rapidity of the POE (we adopted an age model similar to Babila et al. (2022) assuming the onset of the POE is ~ 500 years), which allows for

the assessment of the deep sea (benthic DIC) and the core top $\delta^{13}\text{C}$ responses, in comparison to the surface DIC.

- 2) The small mass of cumulative carbon emitted during the POE. Babila et al. (2022) suggested that “the duration and mass of carbon release was insufficient to alter the deep-ocean reservoir, if we consider the lack of a deep-sea isotopic expression of the POE event at face value.”. The cumulative mass obtained from our revised modelling results support the claim that a small mass of carbon released during the POE is unable to generate a globally significant carbon isotope excursion in the core top sediments (Fig. S10; the core-top carbon isotope excursion magnitude is $\sim 0.2\text{‰}$ to 0.6‰ for the three age models we considered in this study, but much greater for age model 1 ($\sim 1.9\text{‰}$). The fact that the POE is absent in the deep sea (only $\sim 0.4\text{‰}$) is consistent with the shorter-duration of the POE, likely less than two millennia to sub-millennia timescales).
- 3) Issues related to “quantification of how bioturbation should impact the modeled core-top $\delta^{13}\text{C}$ excursion magnitude and how experiment duration affect the core-top excursion magnitude” have been addressed in a new set of experiments turning on and off bioturbation to demonstrate the effects of bioturbation on the CIE magnitudes. Additionally, we tested the effect of experiment duration (from 1700 to 20,000 years) on the CIE magnitudes (Fig. S10). The results suggest that the CIE magnitudes are systematically smaller for experiments with bioturbation on, compared to those without bioturbation. This is similar for core-top CaCO_3 wt%, with smaller degree of dissolution for experiments with bioturbation on. Longer experiment duration allows for a larger CIE magnitude regardless of whether bioturbation is on.

Figure S10. **a-d.** Core-top $\delta^{13}\text{C}$ of carbonate for experiments with bioturbation on (solid lines) and off (dashed lines) varying the experiment duration using four independent age models. **e-h.** Core-top carbonate wt.% for experiments with bioturbation on (solid lines) and off (dashed lines) varying the experiment duration using four independent age models.

Furthermore, we now impose a pH change that is more consistent with the understanding of the POE (i.e., $\Delta\text{pH} \sim -0.1$), as suggested by Babila et al. (2022), and use a global $\delta^{13}\text{C}$ compilation, instead of from a single site $\delta^{13}\text{C}$, along with a number of possible age scenarios to derive the carbon emission pattern during the POE. Our revised model results are consistent with a mainly thermogenic methane source, which is from an external forcing to the Earth's climate system, leading to simultaneous warming and ocean acidification. The rapid recovery to pre-POE $\delta^{13}\text{C}$ condition implies that negative feedback, such as organic carbon burial quickly begins to function to bring the ocean-atmosphere system back to pre-POE conditions. Repeated and larger quantities of carbon emissions from the NAIP and related Earth's system feedbacks later may have pushed the climate into the PETM.

Figure 4. Data assimilation results from our cGENIE Earth system modeling based on the pH- $\delta^{13}\text{C}_{\text{DIC}}$ double inversion of four scenarios based on different assumptions of POE onset duration. (a-d), $\delta^{13}\text{C}_{\text{source}}$ values of the diagnosed carbon source for the four age models (see age model interpretation in the main text). (e-h), Model-diagnosed rates of CO_2 emission for the four age models. (i-l), Cumulative amount of CO_2 emitted for the four age models. The gray shaded area represents 1,500 years.

The authors are using the d11B record from Babila et al. in their inversion, but that paper provides significantly more caveats in the interpretation of d11B as pH across the POE. They write, “the determination of the absolute magnitude of the pH excursion at the POE is limited by both the uncertainty of the initial environmental conditions and the larger uncertainty associated with the laser ablation analyses generated in this study...which makes determining the absolute magnitude of the pH excursion at the POE rather uncertain.” And also, “a full propagation of uncertainty...allows us to determine that the POE pH excursion at the Maryland sites was greater than -0.08, with an upper limit that is poorly constrained but overlaps with the magnitude estimated for the main CIE observed elsewhere.” Two important points I want to emphasize from these quotes: first, the authors here chose to use something close to the poorly constrained upper limit for their pH inversion target (-0.4) in the main results presented in this text but do not provide a justification for this choice. Second, the Babila paper correctly differentiates the pH change observed in the coastal North Atlantic to the pH decline for the CIE observed elsewhere (emphasis mine). I can think of a handful of reasons that the changes in pH near coastal sites and/or more restricted areas (as in this study location) would differ from

the open ocean pH change. Changes in freshwater input, for instance, could account for some amount of local pH change. Hence using a -0.4 pH change in the double inversion seems to me as though the authors are choosing their forcing in order to arrive at a particular result. I appreciate that the authors have performed sensitivity analyses, so I think this entire discussion could be reworked largely using experiments already conducted, even though the authors did not test the minimum pH change estimate of Babila et al. (-0.08). I don't think the "preferred scenario" is well-justified. Moreover, shouldn't the discussion acknowledge and explain contrasts between their results and the modeling results presented by Babila et al., who evaluated a large range of pH changes and POE duration estimates, including much shorter durations than the minimum 1000 years tested here? Also, could the authors provide more discussion of why these age models differ from the age model developed for the Bighorn Basin terrestrial record from Bowen et al. 18? (The authors describe their estimated duration of 7 kyr as similar but slightly longer than" the Bowen estimate, but in reality their estimate is more than double the Bowen estimate, of "fewer than 2,000 years" (Bowen et al., 2018). Also, a more clear acknowledgement that Babila et al also suggested that "it is plausible that a short-lived C emission pulse preceded the main pulse of volcanic emissions" is warranted.

Reply: Thank you for these great suggestions. We now provide new model simulation results assuming the POE onset duration of 500 years (based on Babila et al., 2022), 850 years (based on Bowen et al., 2015), 1,600 years (age option 2), and 7,000 years (age option 1). These new simulations use $\delta^{13}\text{C}$ of dissolved inorganic carbon (DIC) derived from a global compilation of $\delta^{13}\text{C}$ data and annual global mean surface ocean pH (derived from $\delta^{11}\text{B}$ proxy data with $\Delta\text{pH} \sim -0.1$) from the Mid-Atlantic Coastal Plain. The compiled global $\delta^{13}\text{C}$ records across the POE are shown in Fig. S8 (see below), and additional sensitivity experiments are conducted to assess the uncertainty in the alignment between the global $\delta^{13}\text{C}$ and $\delta^{11}\text{B}$ proxy data.

Figure S8. Paired “pH– $\delta^{13}\text{C}$ ” forcings used in our “double inversion” experiments using four independent age models. The $\delta^{13}\text{C}$ data are from a global compilation (blue circles are from the eastern Tethys (this study), green circles are from benthic foraminifera *Subbotina* at South Dover Bridge³⁵, red circles are from benthic foraminifera *Cibicidoids allenii* at South Dover Bridge³⁵, black cross represents bulk carbonate from South Dover Bridge³⁵, yellow upsidedown triangles are from pedogenic carbonate from Wyoming Bighorn Basin³⁶, and blue triangles are bulk carbonate from Forada, Italy³⁷). The pH data are from benthic foraminifera *Cibicidoids allenii* at South Dover Bridge³⁵. The magnitude of pH change used here is a conservative estimate of ~ 0.1 unit decrease from the onset of the POE.

Also, regarding the modeled d13C target: could the authors include a plot in the supplement that shows a high-resolution image of all the d13C records across the POE alone for comparison to the target used in the inversion? Also, were the yellow dots in figure 4 actually the inversion target? Or was the inversion target a smoothed version of this curve?

Reply: A high-resolution image of all the d13C records across the POE from both terrestrial and shallow marine sections is now provided in the supplementary information and also shown here (see Fig. S8 above).

We now make clear that the dots are the raw data from the global compilation, and the black curve is the loess fit curve given by generalized cross validation. The black smoothed curve is the actual inversion target.

Bowen GJ, et al. Two massive, rapid releases of carbon during the onset of the Palaeocene-Eocene thermal maximum. *Nature Geoscience* 8, 44-47 (2015).

Babila TL, et al. Surface ocean warming and acidification driven by rapid carbon release precedes Paleocene-Eocene Thermal Maximum. *Science Advances* 8, eabg1025 (2022).

Crouch EM, Brinkhuis H, Visscher H, Adatte T, Bolle M-P. Late Paleocene-early Eocene dinoflagellate cyst records from the Tethys; further observations on the global distribution of Apectodinium. In: *Special Paper - Geological Society of America*, vol.369 (eds Wing SL, Gingerich PD, Schmitz B, Thomas E) (2003).

Sluijs A, et al. Environmental precursors to rapid light carbon injection at the Palaeocene/Eocene boundary. *Nature* 450, 1218-1221 (2007).

Reference cited: Tierney, J.E., Zhu, J., Li, M., Ridgwell, A., Hakim, G.J., Poulsen, C.J., Whiteford, R.D., Rae, J.W. and Kump, L.R., 2022. Spatial patterns of climate change across the Paleocene–Eocene Thermal Maximum. *Proceedings of the National Academy of Sciences*, 119(42), p.e2205326119.

Line 57 - add reference to Tierney et al., 2022.

Reply: Tierney et al., 2022 is now added to the text "which led to 5-6 °C global warming"

Line 96 – 'The' primary d13Ccarb signal...

Reply: 'The' has now been added.

Line 113 – 'average' rather than 'are averaging'

Reply: The change is made.

Line 109: more about the age model – the 1.2 – 1.9 m peaks show the strongest power? These are well-identified throughout each of the POE and PETM? There is error on the duration between the POE and PETM of +/- 1 precession cycle but no error within the PETM? Presumably the 21-kyr duration estimates are consistent with identification of a single precession cycle, so

please indicate that precession has an assumed 21 kyr duration on Line 112 (instead of ~20 kyr).

Reply: The assumed 21 kyr duration is now indicated.

Line 119 – how can you tell that it is the recovery phase in particular that is truncated? What about alignment based on features in the d13C records? Age model options 1 and 2 seem like extremes – what about 3 and 5 m cycles? What are these?

Reply: Changes in lithology from marlstone to fossiliferous limestone suggest possible truncation during the CIE recovery interval. However, this does not affect our major conclusion on the carbon sources and associated fluxes.

Line 151 – to support the possibility of diagenetic overprinting or freshwater influence, what is the magnitude of the d18O excursion?

Reply: The magnitude of d18O excursion is 2.5‰ (from -3.6‰ to -6.1‰). If this is solely attributed to temperature change, it would indicate a temperature rise of 12.8 °C.

The temperature reconstruction follows the equation of Bemis et al. (1998) and assumptions used in Lauretano et al. (2015):

$$T (\text{°C}) = 16.9 - 4.38(\delta^{18}\text{O}_{\text{carb}} - \delta^{18}\text{O}_{\text{sw}}) + 0.1(\delta^{18}\text{O}_{\text{carb}} - \delta^{18}\text{O}_{\text{sw}})^2$$

Assuming an ice-free sea water value ($\delta^{18}\text{O}_{\text{sw}}$) of -1.2‰.

$$\text{Pre-PETM } T (\text{°C}) = 28.0$$

$$\text{PETM } T (\text{°C}) = 40.8$$

$$\Delta T = 12.8 \text{ °C}$$

This temperature rise is much larger than the range of temperature change across the PETM by Tierney et al. (2022) that suggest global temperature rise is between 5.4 and 5.9 °C at 95% confidence interval. Therefore, we interpret the oxygen isotopes to be possibly impacted by diagenetic overprinting or freshwater influence.

Line 178 – say something about the significance of a reduction in Thaumarchaeota?

Reply: This sentence is now revised to read as: "The lower Cren/(Cren+isoGDGT-0) ratios during the POE and PETM therefore likely reflect a reduction in marine Thaumarchaeota, which may be attributed to warmer surface ocean temperature and lower dissolved oxygen concentration³⁶."

36. Qin W, et al. Confounding effects of oxygen and temperature on the TEX86 signature of marine Thaumarchaeota. *Proceedings of the National Academy of Sciences* 112, 10979-10984 (2015).

Line 255 – perhaps add a sentence here specifying how the carbon emissions rates and sources are re-assessed? As in, substituting the new KZGS d13C record but using existing d11B constraints from Gutjahr?

Reply: Thanks for this suggestion. We now add a new sentence specifying how we substitute the new KZGS d13C record but using existing d11B constraints from Gutjahr et al. (2017). The revised sentence now reads as the following:

“Here, the PETM carbon emissions rates and sources are re-assessed by substituting the new $\delta^{13}\text{C}_{\text{carb}}$ record from the eastern Tethys but using existing $\delta^{11}\text{B}$ constraints from Gutjahr et al.⁹.”

Line 263 – is this really a data assimilation approach?

Reply: We follow the terminology devised in Gutjahr et al. (2017) and use the same approach for our “double inversion” experiment.

Line 274 – the POE change in pH is 0.4 units? As in, larger than the PETM pH change of 0.3?

Reply: We now use the minimum ΔpH of -0.08 unit ($\Delta\text{pH} \approx 0.1$) following the advice of the reviewer and Babila et al., 2022. All modelling results are now revised to reflect this.

Line 279 – this could use a bit more explanation; specifically that you forced the model to follow the d13C excursion in DIC to the minimum but then not through the recovery, such that it was the modeled recovery mechanisms that controlled the mismatch between the modeled d13C of DIC during the recovery from the target curve...

Reply: Thank you for this suggestion. More explanation is now provided to clarify this sentence: “Specifically, the model was forced to follow the $\delta^{13}\text{C}_{\text{DIC}}$ excursion to the minimum in the first ~ 7 kyr but not through the recovery, such that it was the modeled recovery mechanisms that controlled the mismatch between the modeled $\delta^{13}\text{C}_{\text{DIC}}$ during the recovery from the target curve.”

Line 302 – please provide the size of the simulated benthic d13C excursion

Reply: The size of the simulated benthic d13C excursion is provided in supplementary Fig. S10.

Figure S10. **a-d.** Core-top $\delta^{13}\text{C}$ of carbonate for experiments with bioturbation on (solid lines) and off (dashed lines) varying the experiment duration using four independent age models. **e-h.** Core-top carbonate wt.% for experiments with bioturbation on (solid lines) and off (dashed lines) varying the experiment duration using four independent age models.

Line 304 – so why not look at the downcore record?

Reply: We set up the model with bioturbation on and off to test its effect on the magnitude of carbon isotope excursion for the core-top carbonate (see results on bioturbation in Fig. S10). Regarding “downcore record”, the model was only able to simulate core-top CaCO_3 record, therefore we were not able to look at the downcore record.

Line 379 – is this an argument against terrestrial organic carbon burial for the POE? Why would the PETM argument hold for the POE?

Reply: We now remove the discussion on organic carbon burial to focus on the POE model inversion for clarify.

Line 405-406 – are these two transient warming events? Based on the authors discussion, it seems like they are interpreting the entirety of the record as phases of NAIP carbon release.

Reply: We now use “during the POE and PETM” to avoid confusion.

Line 412 – why round up the organic carbon burial estimate; use the same value given above (2500-2700 Pg C)

Reply: We now remove the discussion on organic carbon burial to focus on the POE model inversion for clarify.

Reviewer #1 (Remarks to the Author):

The authors have done an excellent job of addressing previous concerns I had about the presentation and interpretation of their exciting data set. The revised manuscript is much more polished and well presented. I only have a couple of small comments and a few grammar suggestions, otherwise I believe it is now ready for publication.

Reply: Thank you for the suggestions for improving the manuscript, which we have provided detailed responses below.

Response to reviewers: The authors did a great job of addressing my previous comments. One slight error is that they continue to advocate for a potential PETM age of 55.53 Ma from Westerhold et al. (2007) as possible evidence for the POE to be after the current dates we have for the big upswing in NAIP activity. Multiple lines of evidence, including an ash layer within the PETM dated to 55.785 Ma (Charles et al., 2011), rule out a PETM onset age of 55.5 Ma. This doesn't make much difference to the manuscript as the focus has now shifted away from the precise timing of the POE, but I thought it was worth mentioning again.

Reply: We agree with the reviewer that although the focus is not on the precise timing of the POE, the age of the PETM was worth mentioning again. Therefore, we added three references that discuss the age of the PETM (~56 Ma) in Introduction. The three references are the following:

1. Charles AJ, et al. Constraints on the numerical age of the Paleocene-Eocene boundary. *Geochem Geophys Geosyst* 12, Q0AA17 (2011).
2. Röhl U, Westerhold T, Bralower TJ, Zachos JC. On the duration of the Paleocene-Eocene thermal maximum (PETM). *Geochemistry, Geophysics, Geosystems* 8, Q12002 (2007).
3. Zeebe RE, Lourens LJ. Solar System chaos and the Paleocene–Eocene boundary age constrained by geology and astronomy. *Science* 365, 926-929 (2019).

The final piece of the puzzle that I think the manuscript needs prior to acceptance is that there is now a testable hypothesis from their findings. 0.2 to 1.3 Pg C/yr (line 304) is a sizeable flux of carbon to the ocean-atmosphere system, and if the NAIP is the dominant source, this should be discernible in the geological record. With more accurate dating of NAIP rocks, this may confirm the NAIP as a possible source. The discussion and/or conclusion could be improved with a sentence or two along the lines of “our findings predict substantial carbon fluxes driving the POE, and refining geochronological investigations of potential sources such as the NAIP could corroborate

or refute this hypothesis.”

Reply: We now add the suggested sentence in the concluding paragraph, and the new text now reads as the following:

“Furthermore, our findings predict substantial carbon fluxes driving the POE (averaging 0.2 to 1.3 Pg C yr⁻¹), which could be tested by refined geochronological investigations of potential sources such as the NAIP.”

Grammar edits (suggested additions in caps):

Line 40: Add “(NAIP)”

Line 45: Add “confirming THE global...”

Line 50: Add “driven by A largely...”

Line 51: Add “associated with sill intrusionS prior...”

Line 52: Add “involving THE release...”

Line 62: Suggest replacing “twenty millenia” with “20 kyr”

Line 70: Suggest replacing “preceding” with “prior to”

Line 71: Add “PETM ONSET with...”

Line 73: Add “preservation to surface AND SHALLOW WATER records...”

Line 74: Add “Resolving A global POE signal...”

Line 83: Add “that volcanism AND MAGMATISM may also...”

Line 84: Add “the global extent OF THE POE...”

Line 104-105: Describe what the depth scale of the stratigraphic log is relative to when first referred to in the text.

Line 181: Add “weathering AND/or higher terrestrial...”

Reply: All of these suggested changes are made in the revised text.

Lines 260-264: This needs rewording. Mantle inputs of osmium have low values of ¹⁸⁷Os/¹⁸⁸Os due to less rhenium in the source material that has decayed to ¹⁸⁷Os. Low Os isotopes are therefore more unradiogenic, not the other way around.

Reply: The new text that addresses reviewer’s comment is replicated below: “A negative shift in ¹⁸⁷Os/¹⁸⁸Os ratios has been observed prior to the PETM in several sites globally^{32, 73, 78, 79}, lending support to the occurrence of large igneous province activity prior to the PETM.”

Line 345: Specify here whether you are referring to volcanic, thermogenic, or a mixture of these sources from the NAIP here.

Reply: This is now clarified, and the new text reads as the following: “Although this study provides a range of estimates on the carbon source and emission flux during the POE, more precise $\delta^{11}\text{B}$ -based global surface pH records, detailed history of the sill intrusion of the NAIP, sea surface

temperature records from across different latitudes, and better-constrained geochronology of the NAIP activity are clearly needed to reduce the uncertainty of the estimated thermogenic carbon emission fluxes from the NAIP.”